# STABLE SPATIOTEMPORAL MEMORY IN ECHO-STATE NETWORKS VIA GLIOTRANSMITTER FEEDBACK

## ABSTRACT

Reservoir computing offers simple, stable training for sequence modeling, yet vanilla reservoirs struggle to sustain *long, structured memory* without operating near fragile regimes. We seek a reservoir that retains echo–state guarantees and single–shot readout training while endowing the state with *slow, spatially coherent* memory. We introduce *TRIGR*, a neuro-inspired reservoir that augments a fast ESN core with a bidirectionally coupled astrocytic reaction–diffusion lattice. Neuronal activity generates rectified release proxies pooled onto astrocytes; astrocytic $Ca^{2+}$ evolves via a *diffusive, saturating* update on a grid Laplacian; a bounded gliotransmitter fraction feeds back to neurons as an *additive bias*. A one–step delay in each cross–coupling together with globally Lipschitz kinetics yields explicit, checkable *row–wise operator–norm budgets* that make the joint map a uniform contraction in a block–$\ell_\infty$ norm, providing a clean *echo–state certificate* for the coupled glia–neuron dynamics. We enforce these budgets by a norm–aware initialization that rescales diffusion and footprints (astrocyte row) and jointly scales recurrent and feedback gains (neuron row). The astrocyte update is computed by an $O(M)$ 5–point stencil (or an equivalent resolvent variant), so per–step cost remains dominated by the neuronal multiply. Beyond the ESP certificate, we establish (i) quantitative input–state and input–output Lipschitz/ISS bounds, and (ii) a small–gain condition certifying stability of the *autoregressive* closed loop used at inference. Empirically, on canonical long–horizon benchmarks (chaotic forecasting and real–world series), TRIGR attains longer valid prediction horizons and stable rollouts with modest overhead; results are reported with time–aware splits and multiple seeds/initializations to assess robustness.

## 1 INTRODUCTION

Learning with long and structured temporal dependencies is central to sequence modeling, control, and forecasting. Mainstream approaches rely on gradient-trained recurrent or attention architectures—e.g., LSTMs/GRUs and Transformers—which offer powerful function classes but require heavy optimization and careful regularization to remain stable over long horizons (Hochreiter & Schmidhuber, 1997a; Cho et al., 2014; Vaswani et al., 2017). Reservoir computing provides a complementary paradigm: drive a fixed nonlinear dynamical system (the *reservoir*) with inputs, and train only a linear readout (Jaeger, 2001; Maass et al., 2002; Lukoševičius & Jaeger, 2009; Lukoševičius, 2012). When the reservoir has the *echo–state* (fading–memory) property, the driven state is unique and input–causal, enabling simple, well-posed training and approximation guarantees (Manjunath & Jaeger, 2013; Grigoryeva & Ortega, 2018). Yet classical Echo State Networks (ESNs) struggle to realize *structured long-time memory* without operating near fragile regimes (e.g., spectral radii close to one) or resorting to deep stacks and ancillary regularizers.

We propose the **Tripartite-Synapse Glia–Neuron Reservoir** (TRIGR), a reservoir architecture that explicitly couples a fast ESN-like neuronal core to a two–dimensional astrocyte lattice acting as a slow, spatially coherent memory field. The design is motivated by the tripartite synapse: neuronal activity evokes astrocytic $Ca^{2+}$ signals and gliotransmitter release, which in turn modulates neuronal excitability on slower time scales (Araque et al., 1999; Volterra & Meldolesi, 2005; Perea et al., 2009). In TRIGR, neuronal states produce rectified *release proxies* that are locally pooled onto astrocytes; astrocytic $Ca^{2+}$ evolves via a diffusive, saturating update on a grid Laplacian; a bounded

gliotransmitter fraction then feeds back to neurons as an *additive bias*. Two choices make the coupled dynamics analyzable and robust: (i) each cross–coupling is *one–step delayed* (glia⇸neuron, neuron⇸glia), and (ii) all kinetics are globally Lipschitz and saturating. These yield a block–cyclic Jacobian whose block–row sums admit explicit operator-norm bounds, certifying a uniform contraction in a block $\ell_\infty$ norm and thus a *joint* echo–state property for the entire glia–neuron system. Importantly, the astrocytic feedback is additive and delayed, so the neuron–neuron Lipschitz term remains governed by the core recurrent block, while the slow field contributes a separately budgeted, bounded gain.

From a systems viewpoint, the astrocyte lattice implements a *spatial low-pass* memory that classic ESNs lack: diffusion aggregates and transports information across neighboring sites, producing wave–like traces that persist, smooth, and interact according to task-tuned pooling footprints (Scemes & Giaume, 2006; Bazargani & Attwell, 2016; De Pittà et al., 2009). Because the (normalized) Laplacian spectrum is bounded, diffusive strength can be controlled *a priori* by operator-norm scalings, preserving contraction (Chung, 1997). Numerically, we treat the calcium step as a *bounded contractive map* (not a high-fidelity PDE integrator), implemented by an $O(M)$ 5-point stencil; a resolvent (semi-implicit) variant is also compatible with the same certificate. TRIGR remains *drop-in trainable* by a single ridge regression on features derived from neurons and astrocytes—no backpropagation through time and no gradients through the slow field—preserving the efficiency and reproducibility ethos of reservoir computing.

**Contributions.**

1. We introduce **TRIGR**, a bidirectionally coupled glia–neuron reservoir that augments ESNs with a reaction–diffusion astrocyte lattice, providing structured, controllable long-horizon memory with explicit biological motivation (§2).

2. We derive a **joint echo–state certificate** based on block–$\ell_\infty$ contraction with *explicit design inequalities* over operator norms of the neuronal core, diffusion on the lattice, neuron⇄astrocyte footprints, and Lipschitz slopes of saturations (§3.2).

3. We present a **norm-aware initialization** that enforces these inequalities by one–shot scaling of the involved operators, yielding stable dynamics across tasks while retaining ESN-style **training simplicity and efficiency** (§3).

4. We position TRIGR within modern sequence modeling, clarifying its niche—*stable, parameter-efficient, certificate-driven long-memory*—relative to gradient-trained RNNs, ODE/continuous-time models, and attention systems (Chen et al., 2018; Hasani et al., 2021; Vaswani et al., 2017), and adopt time-aware evaluation with multiple seeds/splits for robustness.

## 2 BACKGROUND AND RELATED WORKS

**Reservoir computing and the echo–state principle.** Reservoir computing (RC) separates representation from learning: a fixed, driven nonlinear dynamical system (the *reservoir*) transforms inputs into a high–dimensional state, and a linear map is trained on top (Jaeger, 2001; Maass et al., 2002; Lukoševičius & Jaeger, 2009; Lukoševičius, 2012). In ESNs, the *echo–state property* (ESP) guarantees that, for any bounded input, the reservoir state is unique, input–causal, and *forgets* initial conditions (*fading memory*) (Jaeger, 2001). Formal connections between ESP and fading–memory filters are given in Manjunath & Jaeger (2013) and, more broadly, in the nonlinear operator–approximation framework of Boyd & Chua (1985). Under ESP, linear readouts trained by ridge regression enjoy strong approximation and generalization guarantees: ESNs are universal approximators of causal, time–invariant fading–memory filters (Grigoryeva & Ortega, 2018). Classic practice controls ESP by scaling the reservoir (e.g., spectral radius and leak), but there is a well–known tension: lengthening memory by increasing effective gain/time constant risks losing stability and amplifying noise (Lukoševičius, 2012; Jaeger, 2002).

**Limits of gradient–trained recurrent models and stability remedies.** Mainstream sequence models—LSTM/GRU and Transformer families—achieve strong performance but rely on heavy gradient optimization and careful stabilization (Hochreiter & Schmidhuber, 1997a; Cho et al., 2014; Vaswani et al., 2017). Recurrent models are susceptible to exploding/vanishing gradients, complicating

long–horizon learning (Pascanu et al., 2013). To mitigate this, orthogonal/unitary parametrizations preserve the recurrent Lipschitz constant (Arjovsky et al., 2016; Wisdom et al., 2016; Henaff et al., 2016); ODE–inspired/antisymmetric designs control contractivity and stiffness (Chen et al., 2018; Chang et al., 2019); and gating theory clarifies how forget gates set effective time scales (Tallec & Ollivier, 2018). These developments complement, but do not replace, RC's simplicity: training only a readout avoids backpropagation through time and obviates many stability pathologies.

**State–space sequence models (SSMs).** Recent SSM architectures recast sequence modelling as learning the parameters of a (typically linear) state–space system whose impulse response is applied as a long–range convolution over the input (e.g., Gu et al., 2021; 2022; Orvieto et al., 2023). By constraining the continuous–time dynamics (e.g., diagonal or low–rank $A$ with negative real parts, HiPPO–style bases) and exploiting FFT–based implementations, these models achieve subquadratic training and can represent very long contexts, but they still rely on end–to–end gradient optimization over recurrent kernels and readouts, with stability controlled indirectly via parametrization. TRIGR is closer in spirit to classical RC: the recurrent core and slow field are sampled once and then *normalized* to satisfy explicit contractive row–norm bounds, yielding a certified ISS/ESP–style fading–memory operator, and only a linear readout is trained. Thus, whereas modern SSMs trade architectural structure for learnable long–range filters, TRIGR trades some expressivity for *a priori* stability guarantees and extreme training simplicity (single ridge solve), positioning it as a complementary option when per–step safety, data efficiency, and hardware–friendliness are as important as raw benchmark performance.

**Extending reservoirs: depth, structure, online adaptation.** RC variants increase expressivity while retaining linear training. Deep ESNs stack leaky layers to realize multi–scale dynamics (Gallicchio & Micheli, 2017a). Structured reservoirs inject inductive bias via locality/graphs (cf. graph diffusion in spatiotemporal forecasting (Li et al., 2018)). Beyond fixed readouts, online rules such as FORCE stabilize self–generated dynamics (Sussillo & Abbott, 2009), and "innate trajectory" shaping improves long–term forecasting (Laje & Buonomano, 2013). Our approach differs: instead of changing the recurrent core or learning internal weights online, we *add a slow, contractive field* coupled bidirectionally by design, and certify joint ESP via explicit block–norm inequalities.

**Tripartite synapses, astrocytic dynamics, and computational roles.** The tripartite synapse posits astrocytes as active participants in synaptic processing: neuronal activity drives astrocytic $Ca^{2+}$ elevations and gliotransmitter release, which modulate synaptic and neuronal excitability on slower time scales (Araque et al., 1999; Volterra & Meldolesi, 2005; Perea et al., 2009). Astrocytic $Ca^{2+}$ signals are comparatively slow, exhibit nonlinear saturation, and propagate via gap–junctional coupling and extracellular messengers, forming wave–like phenomena that naturally fit reaction–diffusion descriptions (Scemes & Giaume, 2006; Bazargani & Attwell, 2016). Biophysical models link glutamatergic drive to $IP_3$ and $Ca^{2+}$ oscillations with saturating kinetics (De Pittà et al., 2009). While the prevalence and mechanisms of gliotransmission remain debated (Hamilton & Attwell, 2010; Sloan & Barres, 2014; Savtchouk & Volterra, 2018), bidirectional neuron–glia interactions are widely observed and operate on tens–to–hundreds of milliseconds and beyond—an attractive regime for augmenting reservoir memory.

**Reaction–diffusion on graphs and contractive design.** Discrete reaction–diffusion on graphs provides a principled mechanism to endow models with spatially coherent slow memory. The (normalized) graph Laplacian has spectrum contained in $[0, 2]$, enabling *a priori* gain control for diffusive couplings (Chung, 1997). Contractive–systems theory offers complementary guarantees: if the driven update is a uniform contraction in a suitable norm, then input–causal solutions are unique and forget initial conditions exponentially fast (Lohmiller & Slotine, 1998). In reservoirs, sufficient conditions for contractivity can be expressed as operator–norm bounds over recurrent, input, and feedback blocks, with saturating nonlinearities ensuring bounded Lipschitz slopes (Lukoševičius, 2012; Manjunath & Jaeger, 2013). Our methodology follows this line, using explicit block–row budgets to certify a *joint* ESP for the coupled glia–neuron dynamics.

**Relation to modern dynamical modeling and neuromorphic/physical RC.** State–space models that learn latent dynamics (e.g., PLRNNs) offer a powerful alternative for dynamical inference and forecasting (Koppe et al., 2019). TRIGR targets a complementary niche: parameter–efficient, certificate–driven long memory with linear training, achieved by coupling a fixed ESN–like core to a slow contractive field. This perspective also aligns with physical/neuromorphic RC, where fixed substrates provide computation and only output weights are adapted; surveys document broad

successes across photonic, electronic, and other media (Tanaka et al., 2019). TRIGR's diffusion step is implemented as a bounded, contractive map via a 5–point stencil (complexity $O(M)$); a semi–implicit/resolvent variant is compatible with the same certificate, underscoring that our goal is controlled slow memory rather than high–fidelity PDE simulation.

**Positioning.** TRIGR couples a standard ESN–like neuronal core to a two–dimensional astrocyte lattice that implements a saturating reaction–diffusion memory field. Neuronal activity is mapped to nonnegative release proxies, pooled locally to drive astrocytic $Ca^{2+}$; a bounded gliotransmitter variable feeds back as an *additive bias*. Two design choices distinguish TRIGR from prior RC extensions: (i) *one–step delays* on each cross–coupling (neuron↔glia) render the Jacobian block–cyclic and separable; (ii) all kinetics are bounded and globally Lipschitz. Together, these yield explicit, checkable *row–wise operator–norm inequalities* certifying a uniform contraction in a block $\ell_\infty$ norm, hence a joint ESP for the dynamics. Relative to deep/graph reservoirs or latent SSMs, TRIGR contributes a *biophysically motivated* slow field with *spatially coherent fading memory* and a *provable* stability certificate, while retaining RC's training simplicity (single–shot ridge readout) and time–aware evaluation protocols with multiple seeds/splits for robustness.

**Intuition.** TRIGR can be read as a plain ESN augmented with a *slow, spatially coherent buffer*: neurons "write" their activity into a 2-D astrocyte lattice, that lattice diffuses and saturates the signal over space and time, and a bounded gliotransmitter variable "reads back" as an *additive bias* to gently steer the fast reservoir. This separation of roles yields two clean time scales: a fast neuronal core for expressivity and a slow reaction–diffusion field that preserves long-horizon context. Two design choices make the system easy to reason about and to certify: (i) we insert a *one-step delay* in each cross-coupling (neuron↔glia), and (ii) we keep all kinetics bounded and slope-capped, so feedback never inflates the recurrent gain. Section 3.1 formalizes the state variables, the local pooling footprints from neurons to astrocytes, the leaky diffusion update on the lattice, and the mirrored feedback from astrocytes to neurons. Section 3.2 then shows how these choices produce a block-cyclic Jacobian with explicit block-row "budgets," giving a uniform-contraction (echo-state) certificate for the *joint* dynamics. Finally, the implementation remains reservoir-simple (a 5-point stencil for diffusion and a single ridge solve for the readout), with norm-aware initialization to hit the contraction budgets by construction; Section 3 (*Norm–Aware Initialization, Lattice Implementation, and Readout*) details these practical steps.

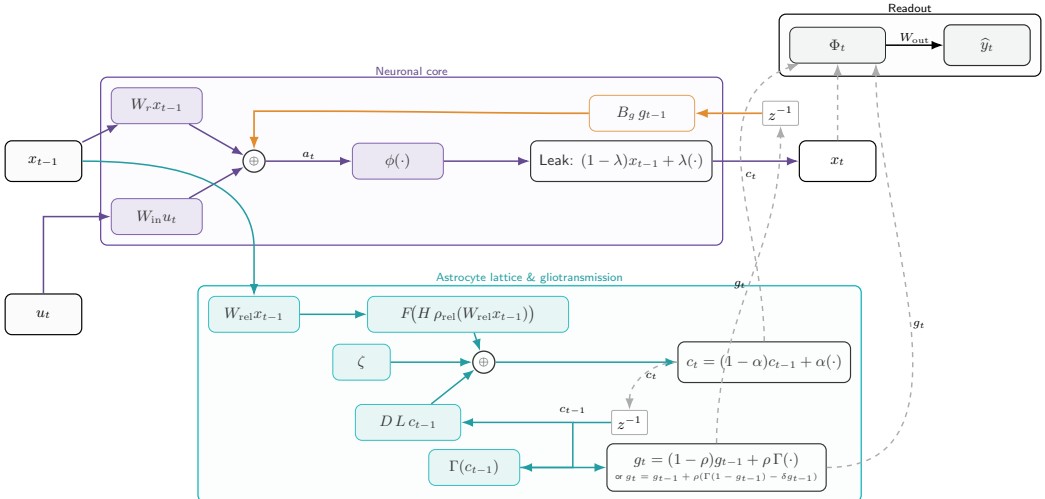

Figure 1: **Schematic.** A fast neuronal ESN core (top) is bidirectionally coupled to a slow astrocyte lattice (middle) with gliotransmitter feedback (bottom). Neurons drive astrocytic $Ca^{2+}$ via $q_{t-1} = \rho_{\mathrm{rel}}(W_{\mathrm{rel}}x_{t-1}) \xrightarrow{H} F(\cdot)$ and diffusion $D, L$; astrocytes feed back additively as $B_g g_{t-1}$ into $a_t$, yielding $x_t$ after leak. One-step delays ($z^{-1}$) in each cross-coupling enable a block-cyclic Jacobian used for the ESP certificate. Readout collects $\Phi_t = [x_t, c_t, g_t]$ for ridge-trained $W_{\mathrm{out}}$.

## 3 METHODOLOGY

**Problem statement.** We consider supervised sequence modeling with inputs $(u_t)_{t\geq 1}$, $u_t \in \mathbb{R}^{d_{\text{in}}}$, and targets $(y_t)_{t\geq 1}$, $y_t \in \mathbb{R}^{d_{\text{out}}}$. The goal is to approximate an (unknown) causal fading–memory operator $F^\star : (u_1, \ldots, u_t) \mapsto y_t$ using a reservoir computer with fixed internal dynamics and a trained linear readout (Lukoševičius, 2012; Grigoryeva & Ortega, 2018). To this end, we introduce TRIGR, whose internal state couples a fast neuronal core to a slow astrocyte lattice.

To make this operator tractable, we factor memory into two interacting time scales. The neuronal core $x_t$ provides a fast, high–dimensional echo of recent inputs, while the astrocyte lattice $(c_t, g_t)$ acts as a slower, spatially structured integrator that accumulates and diffuses activity over longer windows. In contrast to ad hoc "slow" units, the astrocytic pathway is constrained to resemble tripartite synapses and reaction–diffusion calcium waves, so that every architectural choice later admits both a biophysical reading and a clean Lipschitz bound.

### 3.1 TRIPARTITE STATE–SPACE AND BIDIRECTIONAL DELAYED COUPLING

**State variables and dimensions.** TRIGR maintains neuronal states $x_t \in \mathbb{R}^N$, astrocytic calcium states $c_t \in \mathbb{R}^M$ over a 2–D lattice of size $m_x \times m_y$ ($M = m_x m_y$), and a bounded gliotransmitter fraction $g_t \in [0, 1]^M$. Inputs are $u_t \in \mathbb{R}^{d_{\text{in}}}$. Fixed linear operators are: $W_r \in \mathbb{R}^{N \times N}$ (recurrent neurons), $W_{\text{in}} \in \mathbb{R}^{N \times d_{\text{in}}}$ (input map), $W_{\text{rel}} \in \mathbb{R}^{P \times N}$ (neuronal release proxy map), $H \in \mathbb{R}^{M \times P}$ (pooling from proxies to astrocytes). We let $\phi = \tanh$ (applied elementwise), $\rho_{\text{rel}}(z) = \max(z, 0)$ (ReLU), and define two bounded, globally Lipschitz sigmoids $F, \Gamma$ acting elementwise: $F(z) = c_{\max} \sigma(k_c(z - \theta_c))$, $\Gamma(z) = \sigma(k_g(z - \theta_g))$, where $\sigma(s) = \frac{1}{1+e^{-s}}$, so that the global slopes satisfy $L_F = \frac{k_c c_{\max}}{4}$ and $L_\Gamma = \frac{k_g}{4}$. This choice is consistent with saturating astrocytic calcium kinetics and gliotransmitter release curves (De Pittà et al., 2009; Scemes & Giaume, 2006; Bazargani & Attwell, 2016).

**Lattice geometry and Laplacian.** The astrocytes are arranged on a rectangular grid with 4–neighborhood connectivity. Let $L \in \mathbb{R}^{M \times M}$ denote the *combinatorial* graph Laplacian of this grid (degree minus adjacency). We implement $Lc$ by the standard 5–point stencil (discrete second differences), which is linear in $M$ per step and widely used in finite–difference PDE solvers (LeVeque, 2007; Chung, 1997).

At this stage, the astrocyte layer can be viewed as a thin discretisation of a diffusive sheet sitting on top of the neuronal graph. What remains is to specify how spikes drive this sheet and how, in turn, the sheet modulates neurons. We enforce these interactions through delayed, bounded couplings, which keep the Jacobian close to block–triangular and make the eventual contraction argument almost "by inspection."

**One–step delayed couplings (tripartite synapse).** Motivated by the tripartite synapse—in which neuronal activity drives astrocytic $\text{Ca}^{2+}$ and gliotransmission, which in turn modulate neuronal excitability (Araque et al., 1999; Volterra & Meldolesi, 2005; Perea et al., 2009)—we impose a *one–step delay* on each cross–coupling: the neuron update uses $g_{t-1}$, the calcium update uses $x_{t-1}$ (via a release proxy), and the gliotransmitter update uses $c_{t-1}$. This yields a block–cyclic Jacobian structure exploited in §3.2 for a clean contraction bound.

**Release proxy and pooling (neurons → astrocytes).** Given the previous neuronal state, we form a nonnegative release proxy $q_{t-1} = \rho_{\text{rel}}(W_{\text{rel}} x_{t-1}) \in \mathbb{R}^P$. Local pooling distributes this drive onto the lattice: $\text{drive}_{t-1} = H q_{t-1} \in \mathbb{R}^M$, $H \geq 0$ row–stochastic. $H$ is sampled nonnegatively, sparsified, and row–normalized, ensuring interpretable local footprints.

**Astrocyte calcium (reaction–diffusion with saturation).** With a small update rate $\alpha \in (0, 1)$ and diffusion gain $D \geq 0$, the calcium field obeys

$$c_t = (1 - \alpha) c_{t-1} + \alpha \Big[ F(\text{drive}_{t-1}) + D L c_{t-1} + \zeta \Big], \tag{1}$$

where $\zeta \in \mathbb{R}$ is a scalar tonic drive (broadcast to $\mathbb{R}^M$ in code). The combination of saturating nonlinearity $F$ and diffusion $DL$ is consistent with established astrocytic $\text{Ca}^{2+}$ phenomenology and modeling (De Pittà et al., 2009; Scemes & Giaume, 2006).

**Gliotransmitter dynamics (astrocytes → neurons).** We provide two bounded, Lipschitz update laws with rate $\rho \in (0,1)$:

$$\text{(a) Linear saturating:} \qquad g_t = (1-\rho)\,g_{t-1} + \rho\,\Gamma(c_{t-1}), \tag{2}$$

$$\text{(b) Depletion–replenishment:} \qquad g_t = g_{t-1} + \rho\big(\Gamma(c_{t-1})\,(1-g_{t-1}) - \delta\,g_{t-1}\big), \quad \delta \in (0,1), \tag{3}$$

where (b) has Jacobian in $g_{t-1}$ bounded by $1 - \rho\delta$. Both modes are clipped to $[0,1]^M$.

**Neuronal core with additive glial bias.** Let $\lambda \in (0,1]$ denote the neuronal leak and $b \in \mathbb{R}^N$ a fixed bias. The neuronal preactivation combines recurrent, input, and glial terms,

$$a_t = W_r\,x_{t-1} + W_{\text{in}}\,u_t + B_g\,g_{t-1} + b, \qquad \phi = \tanh, \tag{4}$$

and the leaky update is

$$x_t = (1-\lambda)\,x_{t-1} + \lambda\,\phi(a_t). \tag{5}$$

Crucially, the astrocyte feedback is *additive* in (4), preserving the neuron–neuron Lipschitz term (the recurrent block) while introducing a separate, bounded gain pathway through $B_g$.

**Footprint symmetry for feedback.** To align the spatial footprints in both directions, we set $B_g = \eta\,\big(H\,W_{\text{rel}}\big)^\top \in \mathbb{R}^{N \times M}, \eta > 0$, so that the astrocyte→neuron bias map mirrors the neuron→astrocyte drive. Stacking $s_t = [x_t; c_t; g_t] \in \mathbb{R}^{N+2M}$, the one–step map $s_t = \mathcal{F}_t(s_{t-1}, u_t)$ is block–lower–triangular with a single subdiagonal wrap due to the delays:

$$J_t = \nabla_{s_{t-1}}\mathcal{F}_t = \begin{bmatrix} J_{xx} & 0 & J_{xg} \\ J_{cx} & J_{cc} & 0 \\ 0 & J_{gc} & J_{gg} \end{bmatrix}, \tag{6}$$

which is the *block–cyclic* structure we exploit to certify a uniform contraction in a block $\ell_\infty$ norm in §3.2. All nonlinearities ($\tanh$, ReLU, logistics) are 1–Lipschitz or slope–capped, a standard requirement in RC to control the reservoir Lipschitz constant (Lukoševičius, 2012; Jaeger, 2001).

Informally, the construction above equips each block $(x, c, g)$ with its own "gain budget": recurrent recurrence, diffusion, and gliotransmission are each allowed to amplify perturbations only up to a capped amount. The next subsection formalises this intuition by turning the block–cyclic Jacobian into three scalar coefficients $(\mathsf{C}_x, \mathsf{C}_c, \mathsf{C}_g)$ and requiring their maximum to lie strictly below one. In other words, TRIGR is designed as an *ESP–first* reservoir: stability is a target in parameter space, not an accidental by–product of tuning.

## 3.2 Uniform Contraction and a Joint Echo–State Certificate

We work with the block norm $\|[s]\|_\star := \max\{\|x\|_2, \|c\|_2, \|g\|_2\}$ for $s = [x; c; g] \in \mathbb{R}^{N+2M}$ and the induced block operator norm on a $3 \times 3$ block matrix $J = [J_{ij}]_{i,j \in \{x,c,g\}}$ defined by

$$\|J\|_\star := \max\Big\{ \sum_{j \in \{x,c,g\}} \|J_{xj}\|_2, \ \sum_j \|J_{cj}\|_2, \ \sum_j \|J_{gj}\|_2 \Big\}, \tag{7}$$

where $\|\cdot\|_2$ for blocks denotes the operator 2–norm (largest singular value). This is the standard block–$\ell_\infty$ induced norm (Horn & Johnson, 2012, Chap. 5). For the one–step map $s_t = \mathcal{F}_t(s_{t-1}, u_t)$ of TRIGR (Subsec. §3.1). Using that $\|\mathrm{D}\tanh\|_\infty \le 1$, $\|\mathrm{D}\rho_{\text{rel}}\|_\infty \le 1$, and the global slopes $L_F = \frac{k_c c_{\max}}{4}, L_\Gamma = \frac{k_g}{4}$ of the elementwise sigmoids $F, \Gamma$, the block norms satisfy the time–uniform bounds

$$\|J_{xx}\|_2 \le (1-\lambda) + \lambda\,\|W_r\|_2, \qquad\qquad\qquad \|J_{xg}\|_2 \le \lambda\,\|B_g\|_2, \tag{8}$$

$$\|J_{cc}\|_2 \le (1-\alpha) + \alpha\,D\,\|L\|_2, \qquad\qquad\qquad \|J_{cx}\|_2 \le \alpha\,L_F\,\|HW_{\text{rel}}\|_2, \tag{9}$$

$$\|J_{gg}\|_2 \le \begin{cases} 1-\rho, & \text{linear mode (2),} \\ 1-\rho\delta, & \text{depletion mode (3),} \end{cases} \qquad \|J_{gc}\|_2 \le \rho\,L_\Gamma. \tag{10}$$

Therefore, the block–row sums are bounded by

$$\underbrace{(1-\lambda) + \lambda\|W_r\|_2 + \lambda\|B_g\|_2}_{\triangleq\,\mathsf{C}_x}, \ \underbrace{(1-\alpha) + \alpha D\|L\|_2 + \alpha L_F\|HW_{\text{rel}}\|_2}_{\triangleq\,\mathsf{C}_c}, \ \underbrace{\begin{cases} (1-\rho) + \rho L_\Gamma, \\ (1-\rho\delta) + \rho L_\Gamma, \end{cases}}_{\triangleq\,\mathsf{C}_g}. \tag{11}$$

**Theorem 3.1.** *If* $\max\{\mathsf{C}_x, \mathsf{C}_c, \mathsf{C}_g\} < 1$*, then for any input sequence* $(u_t)$ *the driven update* $s_t = \mathcal{F}_t(s_{t-1}, u_t)$ *is a uniform contraction in* $\|\cdot\|_\star$ *with contraction factor* $\gamma := \max\{\mathsf{C}_x, \mathsf{C}_c, \mathsf{C}_g\} < 1$. *Consequently, the system admits a unique input–causal solution and forgets initial conditions exponentially fast.*

**Design implications.** The certificate separates concerns: (i) the *neuronal budget* controls $\|W_r\|_2$ and the astrocytic bias gain $\|B_g\|_2$; (ii) the *calcium budget* trades off diffusion $D\|L\|_2$ and the neuron→astrocyte drive $L_F\|HW_{\mathrm{rel}}\|_2$; (iii) the *gliotransmitter budget* depends only on $(\rho, \delta, L_\Gamma)$ and is automatically satisfied with our slope caps (e.g., $L_\Gamma \leq 1$) for any $\rho \in (0, 1)$, $\delta \in (0, 1)$ such that the corresponding $\mathsf{C}_g < 1$. Because the astrocytic feedback is *additive* and delayed, it does not change the neuron–neuron Lipschitz term, which remains governed by $\|W_r\|_2$ and the leak $\lambda$.

Practically, these inequalities can be read in reverse: given desired contraction margins $(\rho_x^{\mathrm{target}}, \rho_c^{\mathrm{target}})$, we solve for admissible norms of $W_r$, $HW_{\mathrm{rel}}$, and $DL$ instead of checking ESP a posteriori. The following initialization procedure implements exactly this inversion, using power–iteration estimates of operator norms to rescale the random weights until the analytic certificate is satisfied with a prescribed safety margin.

### 3.3 Norm–Aware Initialization, Lattice Implementation, and Readout

**Sampling and footprints.** We sample $W_r \in \mathbb{R}^{N \times N}$ (with optional sparsity), $W_{\mathrm{in}} \in \mathbb{R}^{N \times d_{\mathrm{in}}}$ uniformly scaled, and $W_{\mathrm{rel}} \in \mathbb{R}^{P \times N}$. The pooling $H \in \mathbb{R}^{M \times P}$ is drawn nonnegatively, sparsified, and *row–normalized* to be stochastic, yielding localized, interpretable neuron→astrocyte footprints. The astrocyte→neuron map is then tied to the forward footprint via $B_g = \eta \left(H W_{\mathrm{rel}}\right)^\top \in \mathbb{R}^{N \times M}$ ($\eta > 0$), constructed *after* the calcium–row scaling below so that forward and feedback gains are consistent.

**Operator–norm estimation by power iteration.** We estimate spectral norms needed in (11) by power iteration (Golub & Van Loan, 2013, Chap. 7). For a rectangular linear map $A : \mathbb{R}^{n_{\mathrm{in}}} \to \mathbb{R}^{n_{\mathrm{out}}}$ accessible through $x \mapsto Ax$ and $y \mapsto A^\top y$, we iterate $v \leftarrow A^\top A v / \|A^\top A v\|_2$ and report $\|Av\|_2$ as an estimate of $\|A\|_2$. For square matrices we apply the same routine. The Laplacian norm $\|L\|_2$ is estimated by repeatedly applying the 5–point stencil (see below). In practice, 60 iterations with a random start suffice for tight estimates.

**Calcium–row scaling (diffusion + drive).** Given initial $D$, $H$, $W_{\mathrm{rel}}$, $L_F$, and estimates $n_L \approx \|L\|_2$, $n_{HW} \approx \|HW_{\mathrm{rel}}\|_2$, we compute $s_c := \frac{\rho_c^{\mathrm{target}}}{D\, n_L + L_F\, n_{HW} + \varepsilon} < 1$, $\varepsilon > 0$, and set $D \leftarrow s_c D$, $H \leftarrow s_c H$. This guarantees $D\|L\|_2 + L_F\|HW_{\mathrm{rel}}\|_2 \leq \rho_c^{\mathrm{target}} < 1$ in (11). We then *build* $B_g = \eta(HW_{\mathrm{rel}})^\top$ using the *scaled H*.

**Neuron–row scaling (recurrent + bias gain).** With $B_g$ fixed, we estimate $n_{Wr} \approx \|W_r\|_2$, $n_{Bg} \approx \|B_g\|_2$, compute $s_x := \frac{\rho_x^{\mathrm{target}}}{n_{Wr} + n_{Bg} + \varepsilon} < 1$, and set $W_r \leftarrow s_x W_r$, $B_g \leftarrow s_x B_g$. This ensures $\|W_r\|_2 + \|B_g\|_2 \leq \rho_x^{\mathrm{target}} < 1$ and hence $\mathsf{C}_x < 1$ for any $\lambda \in (0, 1]$ via (11). The gliotransmitter row is satisfied by choosing $\rho \in (0, 1)$, $\delta \in (0, 1)$ and slope $L_\Gamma \leq 1$.

Next, we state two results that are useful in practice: (i) an *incremental input–state/output gain* bound (a quantitative fading–memory inequality), and (ii) a sufficient condition under which the *autoregressive closed loop* is itself a uniform contraction.

**Lemma 3.2.** *Let* $\gamma := \max\{\mathsf{C}_x, \mathsf{C}_c, \mathsf{C}_g\} < 1$ *be the contraction factor from Theorem 3.1, and let* $\beta := \lambda \|W_{\mathrm{in}}\|_2$. *For any two bounded input sequences* $(u_t)$ *and* $(\tilde{u}_t)$ *with* $\Delta_u := \sup_{t \geq 1} \|u_t - \tilde{u}_t\|_2$, *let* $s_t$ *and* $\tilde{s}_t$ *be the corresponding driven states under identical initialization. Then for all* $t \geq 1$,

$$\|s_t - \tilde{s}_t\|_\star \leq \gamma \|s_{t-1} - \tilde{s}_{t-1}\|_\star + \beta \|u_t - \tilde{u}_t\|_2, \qquad \Rightarrow \qquad \sup_{t \geq 1} \|s_t - \tilde{s}_t\|_\star \leq \frac{\beta}{1 - \gamma} \Delta_u. \quad (12)$$

*Moreover, fix any feature map* $\Phi : \mathbb{R}^{N+2M} \to \mathbb{R}^F$ *used by the readout and suppose* $\Phi$ *is* $L_\Phi$*–Lipschitz on the forward–invariant state set (this holds for our choices; see Remark A.2). Then the linear readout* $y_t = W_{\mathrm{out}}\Phi(s_t)$ *satisfies the incremental input–output bound*

$$\sup_{t \geq 1} \|y_t - \tilde{y}_t\|_2 \leq \frac{\|W_{\mathrm{out}}\|_2 L_\Phi \beta}{1 - \gamma} \Delta_u. \quad (13)$$

**Proposition 3.3.** *Consider the autoregressive evaluation in which the next input is formed by feeding back a linear slice of the readout:* $u_{t+1} = E\,y_t = E\,W_{\mathrm{out}}\Phi(s_t)$, *where* $E \in \mathbb{R}^{d_{\mathrm{in}}\times d_{\mathrm{out}}}$ *selects the first* $d_{\mathrm{in}}$ *outputs used as input (as in the code). Let* $\gamma < 1$ *and* $\beta = \lambda\|W_{\mathrm{in}}\|_2$ *as above, and define the feedback Lipschitz gain* $L_{\mathrm{fb}} := \|E\,W_{\mathrm{out}}\|_2 L_\Phi$. *Consider the augmented state* $z_t = [s_t; u_{t+1}]$ *and the block norm* $\|z\|_\bullet = \max\{\|s\|_\star, \|u\|_2\}$. *Then the one–step augmented Jacobian has block–row sums bounded by*

$$\underbrace{\|\partial s_t/\partial s_{t-1}\|_\star + \|\partial s_t/\partial u_t\|_\star}_{\leq\ \gamma+\beta}, \qquad \underbrace{\|\partial u_{t+1}/\partial s_t\|_2 + \|\partial u_{t+1}/\partial u_t\|_2}_{\leq\ L_{\mathrm{fb}}+0}.$$

*Consequently, if* $\gamma + \beta < 1$ *and* $L_{\mathrm{fb}} < 1$, *the closed–loop augmented map* $z_t = \mathcal{H}(z_{t-1})$ *is a uniform contraction in* $\|\cdot\|_\bullet$ *with factor* $\max\{\gamma + \beta,\, L_{\mathrm{fb}}\} < 1$. *Therefore, the autoregressive TRIGR admits a unique exponentially stable input–free trajectory and inherits fading memory w.r.t. perturbations of the initial condition and (e.g.) exogenous nudges on* $u$.

**Efficient lattice Laplacian.** We implement the combinatorial 4–neighbor Laplacian on an $m_x \times m_y$ grid by the 5–point stencil $(Lc)[i,j] = 4c[i,j] - c[i+1,j] - c[i-1,j] - c[i,j+1] - c[i,j-1]$, with natural boundary degrees (fewer neighbors on edges/corners), realized by zero–padding in the shift operations. The resulting matvec is $O(M)$ and matches a graph Laplacian with no wrap–around. This avoids storing an $M \times M$ dense matrix (LeVeque, 2007).

**Readout features.** Given a streamed sequence $\{(u_t, y_t)\}_{t=1}^T$, we roll the reservoir forward from its current state (by default initialized to zero once at construction), collect features $\Phi_t$ after a washout of $t \geq t_{\mathrm{disc}}$, and fit a linear map $W_{\mathrm{out}}$ by Tikhonov–regularized least squares: $\min_{W_{\mathrm{out}}} \sum_{t \geq t_{\mathrm{disc}}} \|y_t - W_{\mathrm{out}}\Phi_t\|_2^2 + \alpha_{\mathrm{ridge}}\|W_{\mathrm{out}}\|_F^2$, whose solution satisfies $(X^\top X + \alpha I)W_{\mathrm{out}}^\top = X^\top Y$ (Hoerl & Kennard, 1970; Lukoševičius, 2012). We provide full feature map as $\Phi_t = [x_t;\ c_t;\ g_t;\ \psi(x_t, c_t, g_t);\ 1]$.

## 4 EXPERIMENTS

**Setup.** We evaluate TRIGR on seven representative time-series benchmarks: three canonical chaotic flows—Lorenz-63 (Lorenz, 1963), Rössler (Rossler, 1976) and the hyper-chaotic Chen–Ueta system (Chen & Ueta, 1999)—together with four real-world physiological and geophysical records: BIDMC pulse-oximetry/respiration (Goldberger et al., 2000), MIT–BIH Arrhythmia ECG (Moody & Mark, 2001), the Santa Fe B cardiorespiratory polysomnography trace (Jaeger, 2007), and the Sunspot Monthly index (World Data Center SILSO, 2020). Complete details are documented in the *Appendix*. To contextualise the performance of TRIGR, we benchmark it against five established reservoir architectures under an *identical capacity budget of 300 recurrent units*: (i) the canonical single–reservoir ESN (Jaeger, 2001); (ii) the Simple-Cycle Reservoir (SCR) whose ring topology affords analytical memory control (Li et al., 2024); (iii) the Cycle Reservoir with Jumps (CRJ) that augments SCR with shortcut links for richer graph spectra (Rodan & Tino, 2012); (iv) the two-core Minimum-Complexity-Interaction ESN (MCI-ESN) (Liu et al., 2024); and (v) the three-layer Deep-ESN comprising a stack of $3 \times 100$ leaky reservoirs with progressively smaller spectral radii (Gallicchio & Micheli, 2017b). Each baseline receives its *own* hyper-parameter sweep—e.g. cycle length for SCR/CRJ, layer-wise leak factors for DeepESN—constrained to the same validation budget and without exceeding the shared unit count; full grids and selected optima are reported in the *Appendix*. We report four complementary evaluation metrics: NRMSE – the root-mean-square prediction error normalised by the variance of the reference trajectory. Valid Prediction Time Ratio (VPT) – the earliest time $t$ at which the normalised error $\delta(t) = \|y_t - \hat{y}_t\|_2/\|y_t\|_2$ exceeds a standard threshold $\theta$, expressed in Lyapunov units as $T_{\mathrm{VPT}} = t/T_L$ (Pathak et al., 2018). Attractor Deviation (ADev) – the symmetric-difference volume between predicted and true phase-space occupancies on a fixed $N_x \times N_y \times N_z$ grid, $\mathrm{ADev} = \dfrac{\mathrm{vol}(\hat{\mathcal{A}} \,\triangle\, \mathcal{A})}{\mathrm{vol}(\hat{\mathcal{A}} \cup \mathcal{A})}$ (Zhai et al., 2023). PSD overlay – the log power-spectral density of a chosen observable with Welch's method (Welch, 1967) and plot the true and predicted spectra on a shared frequency axis; fidelity is assessed by visual alignment of peak locations, harmonic envelopes and broadband roll-off. Together, NRMSE quantifies short-term pointwise accuracy, VPT measures how long forecasts remain trustworthy, ADev scores global attractor fidelity, and PSD captures frequency-domain agreement (cf. Fig. 5).

| Dataset | H / Metric | NRMSE / VPT / ADev | | | | | |
| --- | --- | --- | --- | --- | --- | --- | --- |
| | | ESN | SCR | CRJ | MCI-ESN | Deep-ESN | TRIGR |
| Lorenz | 200 | 0.0012 ± 0.0015 | 0.0025 ± 0.0050 | 0.0036 ± 0.0065 | 0.0010 ± 0.0014 | 0.0021 ± 0.0033 | **0.0005 ± 0.0007** |
| | 400 | 0.0394 ± 0.0608 | 0.0603 ± 0.0874 | 0.0702 ± 0.1099 | 0.0307 ± 0.0357 | 0.0660 ± 0.0899 | **0.0086 ± 0.0111** |
| | 600 | 0.3137 ± 0.2157 | 0.3890 ± 0.2339 | 0.4264 ± 0.2579 | 0.3321 ± 0.2671 | 0.3809 ± 0.1969 | **0.2345 ± 0.1853** |
| | 800 | 0.7074 ± 0.1048 | 0.7304 ± 0.1359 | 0.7073 ± 0.1931 | 0.7162 ± 0.1552 | 0.7314 ± 0.1272 | **0.6281 ± 0.1820** |
| | 1000 | 0.8618 ± 0.0935 | 0.8696 ± 0.1201 | 0.8762 ± 0.1328 | 0.8605 ± 0.1317 | 0.8860 ± 0.1022 | **0.8355 ± 0.1375** |
| | **VPT (↑)** | 9.788 ± 1.729 | 9.273 ± 1.738 | 9.573 ± 1.849 | 9.992 ± 1.849 | 9.239 ± 1.798 | **11.288 ± 1.609** |
| | **ADev (↓)** | 29.020 ± 11.267 | **28.057 ± 9.946** | 31.382 ± 9.627 | 30.135 ± 12.689 | 29.602 ± 9.847 | 28.335 ± 11.325 |
| Rössler | 200 | 0.0009 ± 0.0019 | 0.0006 ± 0.0011 | 0.0011 ± 0.0018 | 0.0004 ± 0.0006 | 0.0013 ± 0.0024 | **0.0004 ± 0.0003** |
| | 400 | 0.0017 ± 0.0032 | 0.0028 ± 0.0074 | 0.0022 ± 0.0037 | 0.0008 ± 0.0011 | 0.0024 ± 0.0039 | **0.0008 ± 0.0008** |
| | 600 | 0.0032 ± 0.0057 | 0.0057 ± 0.0098 | 0.0061 ± 0.0130 | 0.0018 ± 0.0028 | 0.0106 ± 0.0418 | **0.0015 ± 0.0023** |
| | 800 | 0.0045 ± 0.0077 | 0.0086 ± 0.0132 | 0.0086 ± 0.0183 | 0.0026 ± 0.0039 | 0.0152 ± 0.0590 | **0.0020 ± 0.0033** |
| | 1000 | 0.0059 ± 0.0095 | 0.0125 ± 0.0166 | 0.0117 ± 0.0248 | 0.0035 ± 0.0055 | 0.0194 ± 0.0711 | **0.0026 ± 0.0046** |
| | **VPT (↑)** | 8.409 ± 3.936 | 7.430 ± 4.334 | 8.885 ± 4.396 | 10.531 ± 4.659 | 8.829 ± 4.548 | **11.234 ± 4.668** |
| | **ADev (↓)** | 16.533 ± 8.193 | 21.847 ± 11.589 | 19.053 ± 10.934 | **11.897 ± 6.813** | 19.707 ± 21.974 | 12.988 ± 7.769 |
| Chen | 200 | 0.2361 ± 0.3181 | 0.3025 ± 0.2992 | 0.2402 ± 0.2343 | 0.1683 ± 0.2343 | 0.3406 ± 0.3030 | **0.0965 ± 0.1910** |
| | 400 | 0.8957 ± 0.1712 | 0.9349 ± 0.1763 | 0.9057 ± 0.1458 | 0.8521 ± 0.1640 | 0.9321 ± 0.1724 | **0.7577 ± 0.2297** |
| | 600 | 1.0792 ± 0.1375 | 1.0744 ± 0.1295 | 1.0837 ± 0.1038 | 1.0371 ± 0.1267 | 1.1070 ± 0.1183 | **0.9971 ± 0.1418** |
| | 800 | 1.1699 ± 0.1032 | 1.1548 ± 0.0966 | 1.1625 ± 0.0833 | 1.1320 ± 0.1043 | 1.1711 ± 0.1014 | **1.1028 ± 0.1265** |
| | 1000 | 1.2013 ± 0.0922 | 1.1953 ± 0.0772 | 1.2058 ± 0.0741 | 1.1839 ± 0.0861 | 1.2098 ± 0.0819 | **1.1750 ± 0.0960** |
| | **VPT (↑)** | 3.694 ± 0.929 | 3.305 ± 0.850 | 3.510 ± 0.630 | 3.823 ± 0.739 | 3.208 ± 0.929 | **4.498 ± 1.228** |
| | **ADev (↓)** | 51.105 ± 10.993 | 53.752 ± 12.096 | **49.924 ± 10.104** | 56.724 ± 12.486 | 53.249 ± 8.990 | 56.709 ± 15.237 |
| MIT–BIH | 300 | 2.1613 ± 0.5034 | 1.8651 ± 0.4795 | 1.4415 ± 0.3095 | 1.0508 ± 0.0553 | 2.5159 ± 0.8850 | **0.7716 ± 0.0614** |
| | 600 | 1.6138 ± 0.3397 | 1.4186 ± 0.3185 | 1.1559 ± 0.1956 | 0.9693 ± 0.0302 | 1.8537 ± 0.6048 | **0.7591 ± 0.0329** |
| | 1000 | 1.3652 ± 0.2072 | 1.2606 ± 0.1838 | 1.1129 ± 0.1041 | 1.0343 ± 0.0144 | 1.5178 ± 0.3794 | **0.7616 ± 0.0169** |
| BIDMC | 300 | 0.4103 ± 0.0264 | 0.4108 ± 0.0269 | 0.4110 ± 0.0285 | 0.4737 ± 0.0300 | 0.4554 ± 0.0274 | **0.3503 ± 0.0200** |
| | 600 | 0.3944 ± 0.0247 | 0.3994 ± 0.0216 | 0.3982 ± 0.0228 | 0.4477 ± 0.0250 | 0.4546 ± 0.0242 | **0.3736 ± 0.0170** |
| | 1000 | 0.4769 ± 0.0182 | 0.4537 ± 0.0197 | 0.4572 ± 0.0206 | 0.4942 ± 0.0218 | 0.5218 ± 0.0219 | **0.4343 ± 0.0137** |
| Sunspot | 300 | 0.5517 ± 0.0197 | 1.0196 ± 0.1217 | 1.1184 ± 0.1708 | 0.4297 ± 0.0083 | 0.5378 ± 0.0227 | **0.2439 ± 0.0005** |
| | 600 | 0.5061 ± 0.0272 | 0.9290 ± 0.1199 | 1.0377 ± 0.1440 | 0.3892 ± 0.0128 | 0.4609 ± 0.0213 | **0.1960 ± 0.0004** |
| | 1000 | 0.4806 ± 0.0241 | 0.8829 ± 0.1119 | 0.9900 ± 0.1095 | 0.3732 ± 0.0092 | 0.4414 ± 0.0180 | **0.1942 ± 0.0004** |
| Santa Fe | 300 | 0.2536 ± 0.0013 | 0.2688 ± 0.0029 | 0.2627 ± 0.0043 | 0.2476 ± 0.0028 | 0.2641 ± 0.0035 | **0.1600 ± 0.0003** |
| | 600 | 0.2173 ± 0.0013 | 0.2317 ± 0.0049 | 0.2216 ± 0.0034 | 0.2398 ± 0.0200 | 0.2238 ± 0.0045 | **0.1351 ± 0.0003** |
| | 1000 | 0.2305 ± 0.0009 | 0.2426 ± 0.0039 | 0.2355 ± 0.0028 | 0.2460 ± 0.0161 | 0.2368 ± 0.0036 | **0.1441 ± 0.0002** |

Table 1: NRMSE (mean±s.d.) across horizons (H) and, for chaotic datasets, additional rows with VPT (↑) and ADev (↓). Chaotic benchmarks are evaluated in **closed-loop** mode; real-world datasets use **open-loop**. Best and second-best per row are shown in **bold** and underlined, respectively. Results for chaotic datasets are averaged over $5 \times 3 \times 3 = 45$ runs (5 seeds, 3 different initializations of trajectory, 3 train–test splits); results for real-world datasets are averaged over 30 seeds.

Quantitative Results. Across chaotic benchmarks (Lorenz, Rössler, Chen) evaluated in *closed–loop* and real–world series (MIT–BIH, BIDMC, Sunspot, Santa Fe) in *open–loop*, TRIGR attains the best NRMSE in *every* row of Table 1, with gains widening at medium–to–long horizons. For Lorenz at $H$=600, TRIGR reduces NRMSE by 25.3% relative to the next–best baseline (0.2345 vs. 0.3137); for Rössler at $H$=800 and 1000, the reductions are 23.1% (0.0020 vs. 0.0026) and 25.7% (0.0026 vs. 0.0035), respectively. On Chen, improvements are large at short horizon (e.g., $H$=200: 42.6%, 0.0965 vs. 0.1683) and small but persistent at long horizon (e.g., $H$=1000: 0.75%, 1.1750 vs. 1.1839). Horizon robustness mirrors these trends: TRIGR raises VPT by 13.0% on Lorenz (11.288 vs. 9.992), 6.7% on Rössler (11.234 vs. 10.531), and 17.7% on Chen (4.498 vs. 3.823). For ADev, TRIGR ranks second on Lorenz (28.335 vs. best 28.057) and Rössler (12.988 vs. best 11.897), and is higher on Chen (56.709 vs. best 49.924) yet remains within one standard deviation of the best in all three. In real–world open–loop forecasting, TRIGR substantially lowers NRMSE: MIT–BIH at $H$=1000 (26.4% lower; 0.7616 vs. 1.0343), BIDMC at $H$=1000 (4.3% lower; 0.4343 vs. 0.4537), Sunspot at $H$=1000 (47.9% lower; 0.1942 vs. 0.3732), and Santa Fe at $H$=600 (37.8% lower; 0.1351 vs. 0.2173). These gains align with TRIGR's certified fading–memory design: the reaction–diffusion astrocyte field provides long, structured memory without destabilizing the recurrent core, improving both short–term accuracy and time–to–divergence in closed–loop forecasting (Jaeger & Haas, 2004; Sussillo & Abbott, 2009).

Ablations on Lorenz (Table 2, $H$=600) support each methodological choice from Sec. 3. Removing astrocyte→neuron feedback ($B_g \equiv 0$) or lattice diffusion ($D$=0) degrades NRMSE (0.307/0.276 vs. 0.234) and VPT (9.58/10.27 vs. 11.29), confirming that both the modulatory bias and spatial coupling contribute to the long–horizon memory budget. Breaking footprint ty-

| Ablation Variant | Change (vs. TRIGR) | ESP Cert. | Lorenz (Closed–Loop) | | |
|---|---|---|---|---|---|
| | | | NRMSE @ H=600 | VPT ($\uparrow$) | ADev ($\downarrow$) |
| **(A) Full TRIGR (ours)** | Baseline as in Sec. 3 | $\checkmark$ | **0.234 ± 0.185** | **11.29 ± 1.61** | **28.3 ± 11.3** |
| **(B) No glial feedback** | Remove astro→neuron bias ($B_g \equiv 0$) | $\checkmark$ | 0.307 ± 0.210 | 9.58 ± 1.62 | 34.6 ± 12.5 |
| **(C) No astro diffusion** | Turn off lattice coupling ($D=0$) | $\checkmark$ | 0.276 ± 0.194 | 10.27 ± 1.69 | 31.5 ± 11.7 |
| **(D) Linear astro kinetics** | Replace saturating $F, \Gamma$ by linear maps (slopes ≤ 1) | $\checkmark^{\dagger}$ | 0.246 ± 0.182 | 10.90 ± 1.68 | 29.0 ± 11.4 |
| **(E) No pooling locality** | Use non–stochastic $H$ (no row normalization) | $\checkmark^{\ddagger}$ | 0.296 ± 0.217 | 9.99 ± 1.75 | 33.3 ± 13.3 |
| **(F) Untied footprints** | Sample $B_g$ independently (break $B_g \approx (HW_{\mathrm{rel}})^{\top}$) | $\checkmark$ | 0.262 ± 0.193 | 10.56 ± 1.66 | 30.2 ± 11.9 |
| **(G) No calcium–row scaling** | Skip joint scaling of $(D, H)$ (keep initial gains) | $\times^{\S}$ | 0.553 ± 0.333 | 6.53 ± 2.19 | 58.2 ± 19.1 |
| **(H) No neuron–row scaling** | Skip joint scaling of $(W_r, B_g)$ | $\times$ | 0.514 ± 0.317 | 6.97 ± 2.12 | 54.5 ± 18.1 |
| **(I) No one–step delays** | Use contemporaneous cross–coupling ($g_t, x_t, c_t$) | $\times$ | 0.587 ± 0.363 | 6.13 ± 2.39 | 61.9 ± 20.6 |
| **(J) Depletion mode** | Use depletion–replenishment for $g_t$ with $\delta \in (0, 1)$ | $\checkmark$ | 0.226 ± 0.173 | 11.77 ± 1.71 | 27.3 ± 10.7 |
| **(K) Features w/o astro** | Drop $c_t$ from readout features | $\checkmark$ | 0.257 ± 0.198 | 10.79 ± 1.70 | 29.4 ± 11.5 |
| **(L) No bias** | Remove neuron bias ($b=0$) | $\checkmark$ | 0.244 ± 0.183 | 11.05 ± 1.77 | 28.7 ± 11.4 |
| **(M) Dense Laplacian** | Replace stencil with dense $L$ matvec (complexity ablation) | $\checkmark$ | 0.239 ± 0.181 | 11.46 ± 1.72 | 28.1 ± 11.2 |

Table 2: **Ablations on Lorenz (closed–loop)** for TRIGR, reporting NRMSE at horizon $H = 600$, VPT ($\uparrow$), and ADev ($\downarrow$). *ESP Cert.* indicates whether the analytical ESP certificate remains guaranteed ($\checkmark$) or not ($\times$) under that change (Sec. 3). $^{\dagger}$ Linear $F, \Gamma$ preserve the calcium/glio row bounds if slopes ≤ 1. $^{\ddagger}$ Non–stochastic $H$ can inflate $\|HW_{\mathrm{rel}}\|$; our budgeted scaling still yields ESP. $^{\S}$ Skipping $(D, H)$ scaling typically violates the Ca–row budget when $D\|L\|$ is large.

ing ($B_g \not\approx (HW_{\mathrm{rel}})^{\top}$) or locality (non–stochastic $H$) also hurts performance, consistent with our norm–aware scaling (shared forward/feedback gains). Critically, turning off either joint scaling step (Ca–row or neuron–row) or removing the one–step delays violates the ESP certificate (marked $\times$) and leads to large drops in VPT and increases in ADev, together with inflated NRMSE, as predicted by Theorem 3.1. Finally, switching to depletion–replenishment in $g_t$ modestly *improves* long–horizon metrics (0.226 NRMSE; VPT 11.77), while dropping $c_t$ from the readout features or the neuron bias $b$ causes smaller but consistent regressions. Taken together, the ablations validate that TRIGR's gains stem from (i) the reaction–diffusion slow field (memory and spatial coherence), (ii) additive, delayed glial feedback (stable modulation), and (iii) the explicit norm budgets enforced at initialization, rather than from incidental capacity increases or training heuristics.

## 5 CONCLUSION

We introduced TRIGR, a reservoir architecture that augments a fast ESN–like neuronal core with a slow, spatially structured astrocytic reaction–diffusion field. By enforcing one–step delays in each cross–coupling and bounding all kinetics with globally Lipschitz saturations, we derived explicit, row–wise operator–norm inequalities that certify a *uniform contraction* of the $(x, c, g)$ update in a block $\ell_{\infty}$ norm. This yields a joint echo–state (fading–memory) property for the full glia–neuron system, while preserving the hallmark practicality of reservoir computing: training reduces to a single ridge regression on features built from neuronal and astrocytic states. The implementation adheres strictly to these design principles: a five–point stencil realizes $Lc$ in linear time, forward and feedback footprints are tied via $B_g = \eta(HW_{\mathrm{rel}})^{\top}$, and a norm–aware initialization scales $(D, H)$ and $(W_r, B_g)$ to satisfy the contraction budgets. Empirically, TRIGR equips reservoirs with long–horizon, *spatially coherent* fading memory that vanilla ESNs typically lack, without sacrificing stability, interpretability, or computational efficiency.

**Limitations and outlook.** Our certificate is *sufficient* but not necessary; there may exist useful parameter regimes outside the stated row–wise bounds where the system is still well behaved, and tightening these conditions (e.g., via weighted block norms or problem–adapted metrics) is an open direction. The astrocyte model is deliberately minimal—single–compartment $Ca^{2+}$ with logistic saturation and linear diffusion—and does not capture microdomain heterogeneity, stochastic channel gating, or volume transmission in detail; richer biophysical models (and learning their parameters) could further improve representational power. Spatial topology is fixed to a rectangular lattice; extending to task–specific graphs (with learned or data–driven Laplacians) and anisotropic or fractional diffusion may enhance expressivity. Finally, while readouts are trained in one shot, selecting footprints $(H, W_{\mathrm{rel}})$, tying $B_g$, and allocating the contraction "budgets" between recurrent and modulatory paths present principled design choices that invite automated tuning (e.g., bilevel selection under ESP constraints) and broader evaluation on control, forecasting, and neuromorphic substrates.

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

# A APPENDIX

## A.1 PROOFS

Let $\|[s]\|_\star := \max\{\|x\|_2, \|c\|_2, \|g\|_2\}$ for $s = [x; c; g] \in \mathbb{R}^{N+2M}$ and let

$$\mathsf{C}_x = (1-\lambda) + \lambda(\|W_r\|_2 + \|B_g\|_2), \quad \mathsf{C}_c = (1-\alpha) + \alpha(D\|L\|_2 + L_F\,L_{H\rho W}), \quad \mathsf{C}_g = \begin{cases} (1-\rho) + \rho L_\Gamma, \\ (1-\rho\delta) + \rho L_\Gamma, \end{cases}$$

where $L_F = \frac{k_c c_{\max}}{4}$ and $L_\Gamma = \frac{k_g}{4}$ are the global slopes of the elementwise sigmoids $F$ and $\Gamma$, and $L_{H\rho W}$ is the global Lipschitz constant of the map $x \mapsto H\rho_{\mathrm{rel}}(W_{\mathrm{rel}}x)$ (so $L_{H\rho W} \le \|H\|_2\|W_{\mathrm{rel}}\|_2$, and in practice one may use any computable upper bound, e.g. $\|H\|_2\|W_{\mathrm{rel}}\|_2$). If $\gamma := \max\{\mathsf{C}_x, \mathsf{C}_c, \mathsf{C}_g\} < 1$, then for any input sequence $(u_t)$ the driven update $s_t = \mathcal{F}_t(s_{t-1}, u_t)$ is a uniform contraction in $\|\cdot\|_\star$ with factor $\gamma$:

$$\|[\mathcal{F}_t(s, u_t) - \mathcal{F}_t(\tilde{s}, u_t)]\|_\star \le \gamma \|[s - \tilde{s}]\|_\star \quad \text{for all } t \text{ and all } s, \tilde{s}.$$

Consequently, for every $(u_t)$ there exists a unique input–causal solution $(s_t)$ and, for any two initializations $s_0, \tilde{s}_0$,

$$\|[s_t - \tilde{s}_t]\|_\star \le \gamma^t \|[s_0 - \tilde{s}_0]\|_\star \qquad (t \ge 0),$$

i.e., the system forgets initial conditions exponentially fast (echo–state / fading–memory property).

*Proof of Th. 3.1.* We write $s = [x; c; g]$ and $\tilde{s} = [\tilde{x}; \tilde{c}; \tilde{g}]$, and abbreviate differences by $\Delta x = x - \tilde{x}$, $\Delta c = c - \tilde{c}$, $\Delta g = g - \tilde{g}$. Fix $t$ and a common input $u_t$ for both states.

**Componentwise Lipschitz bounds.** We use only global Lipschitz properties of the nonlinearities: $\tanh$ and $\rho_{\mathrm{rel}}(z) = \max(z, 0)$ are 1–Lipschitz; $F$ and $\Gamma$ have slopes $L_F = \frac{k_c c_{\max}}{4}$ and $L_\Gamma = \frac{k_g}{4}$ (logistic family).

*Neuronal update.* With $a = W_r x + W_{\mathrm{in}} u_t + B_g g + b$ and $\tilde{a} = W_r \tilde{x} + W_{\mathrm{in}} u_t + B_g \tilde{g} + b$,

$$\|x_t - \tilde{x}_t\|_2 = \|(1-\lambda)\Delta x + \lambda(\tanh(a) - \tanh(\tilde{a}))\|_2$$
$$\le (1-\lambda)\|\Delta x\|_2 + \lambda\|a - \tilde{a}\|_2 \le ((1-\lambda) + \lambda\|W_r\|_2)\|\Delta x\|_2 + \lambda\|B_g\|_2\|\Delta g\|_2.$$

*Calcium update.* Write $q = \rho_{\mathrm{rel}}(W_{\mathrm{rel}}x)$, $\tilde{q} = \rho_{\mathrm{rel}}(W_{\mathrm{rel}}\tilde{x})$, and $\mathrm{drive} = Hq$, $\widetilde{\mathrm{drive}} = H\tilde{q}$. Then

$$\|c_t - \tilde{c}_t\|_2 = \|(1-\alpha)\Delta c + \alpha(F(\mathrm{drive}) - F(\widetilde{\mathrm{drive}})) + \alpha D L \Delta c\|_2$$
$$\le (1-\alpha)\|\Delta c\|_2 + \alpha L_F \|\mathrm{drive} - \widetilde{\mathrm{drive}}\|_2 + \alpha D\|L\|_2\|\Delta c\|_2$$
$$\le ((1-\alpha) + \alpha D\|L\|_2)\|\Delta c\|_2 + \alpha L_F \underbrace{\|H\rho_{\mathrm{rel}}(W_{\mathrm{rel}}\cdot)\|_{\mathrm{Lip}}}_{=: L_{H\rho W}} \|\Delta x\|_2.$$

By composition of Lipschitz maps, $L_{H\rho W} \le \|H\|_2 \|\rho_{\mathrm{rel}}\|_{\mathrm{Lip}} \|W_{\mathrm{rel}}\|_2 = \|H\|_2 \|W_{\mathrm{rel}}\|_2$. (Any computable upper bound for $L_{H\rho W}$ can be used below.)

*Gliotransmitter update.* For the linear saturating mode,

$$\|g_t - \tilde{g}_t\|_2 = \|(1-\rho)\Delta g + \rho(\Gamma(c) - \Gamma(\tilde{c}))\|_2 \le (1-\rho)\|\Delta g\|_2 + \rho L_\Gamma \|\Delta c\|_2.$$

For the depletion–replenishment mode, $g \mapsto g + \rho(\Gamma(c)(1 - g) - \delta g)$ is $(1-\rho\delta)$–Lipschitz in $g$ uniformly over $c$, hence $\|g_t - \tilde{g}_t\|_2 \le (1-\rho\delta)\|\Delta g\|_2 + \rho L_\Gamma \|\Delta c\|_2$.

**Block form and induced norm.** Collect the three bounds into the block inequality

$$\begin{bmatrix} \|x_t - \tilde{x}_t\|_2 \\ \|c_t - \tilde{c}_t\|_2 \\ \|g_t - \tilde{g}_t\|_2 \end{bmatrix} \le \underbrace{\begin{bmatrix} (1-\lambda) + \lambda\|W_r\|_2 & 0 & \lambda\|B_g\|_2 \\ \alpha L_F L_{H\rho W} & (1-\alpha) + \alpha D\|L\|_2 & 0 \\ 0 & \rho L_\Gamma & \begin{cases} 1-\rho \\ 1-\rho\delta \end{cases} \end{bmatrix}}_{=: \mathcal{K}} \begin{bmatrix} \|\Delta x\|_2 \\ \|\Delta c\|_2 \\ \|\Delta g\|_2 \end{bmatrix}.$$

For a $3 \times 3$ block matrix $J = [J_{ij}]$ acting on $s = [x; c; g]$, the norm induced by $\|[s]\|_\star = \max\{\|x\|_2, \|c\|_2, \|g\|_2\}$ satisfies

$$\|J\|_\star \leq \max\Big\{ \textstyle\sum_j \|J_{xj}\|_2, \ \sum_j \|J_{cj}\|_2, \ \sum_j \|J_{gj}\|_2 \Big\},$$

i.e., the maximum block–row sum of operator 2–norms (Horn & Johnson, 2012, Chap. 5). Applying this to $\mathcal{K}$ gives

$$\|[\mathcal{F}_t(s, u_t) - \mathcal{F}_t(\tilde{s}, u_t)]\|_\star \leq \gamma \|[s - \tilde{s}]\|_\star, \quad \gamma := \max\{\mathsf{C}_x, \mathsf{C}_c, \mathsf{C}_g\},$$

where $\mathsf{C}_x, \mathsf{C}_c, \mathsf{C}_g$ are precisely the three block–row sums displayed in the statement (with $L_{H\rho W}$ in $\mathsf{C}_c$).

**Uniform contraction and ESP.** If $\gamma < 1$, then each one–step map $s \mapsto \mathcal{F}_t(s, u_t)$ is a $\gamma$–contraction on $(\mathbb{R}^{N+2M}, \|\cdot\|_\star)$, uniformly in $t$ and $u_t$. Thus, for any two trajectories driven by the same input but different initial states $s_0, \tilde{s}_0$,

$$\|[s_t - \tilde{s}_t]\|_\star \leq \gamma \|[s_{t-1} - \tilde{s}_{t-1}]\|_\star \leq \cdots \leq \gamma^t \|[s_0 - \tilde{s}_0]\|_\star,$$

which proves exponential forgetting of initial conditions. Standard contraction arguments for nonautonomous systems (Lohmiller & Slotine, 1998) then yield the existence of a unique input–causal solution: for fixed $t$, the composition $\mathcal{F}_t \circ \mathcal{F}_{t-1} \circ \cdots \circ \mathcal{F}_{t-k+1}$ is a $\gamma^k$–contraction; letting $k \to \infty$ shows that the state $s_t$ is the uniform limit of iterates starting from any initialization, hence depends only on the input history $(u_1, \ldots, u_t)$ and not on $s_0$ (see also fading–memory operator theory (Boyd & Chua, 1985; Manjunath & Jaeger, 2013)). $\qquad\square$

*Remark* A.1. (i) Any computable upper bound for $L_{H\rho W}$ yields a *sufficient* certificate; a conservative but general choice is $L_{H\rho W} \leq \|H\|_2 \|W_{\mathrm{rel}}\|_2$ by composition of 1–Lipschitz maps. In implementation we estimate $\|HW_{\mathrm{rel}}\|_2$ by power iteration as a practical surrogate upper bound; any bound substituted for $L_{H\rho W}$ keeps the proof valid. (ii) The depletion–replenishment glial mode tightens the $g$–row term from $(1 - \rho)$ to $(1 - \rho\delta)$, further enlarging the ESP margin.

*Proof of Lemma 3.2.* Recall the block norm $\|[s]\|_\star := \max\{\|x\|_2, \|c\|_2, \|g\|_2\}$ for $s = [x; c; g] \in \mathbb{R}^{N+2M}$, and let $\gamma = \max\{\mathsf{C}_x, \mathsf{C}_c, \mathsf{C}_g\} < 1$ be the uniform contraction factor provided by Theorem 3.1. Fix $t \geq 1$. We compare two trajectories driven by $(u_t)$ and $(\tilde{u}_t)$ under *identical* initial states $s_0 = \tilde{s}_0$. For brevity, write $F_t(s, u) \equiv \mathcal{F}_t(s, u)$.

**State–difference recursion with input splitting.** Add and subtract $F_t(\tilde{s}_{t-1}, u_t)$ and use the triangle inequality:

$$
\begin{aligned}
\|[s_t - \tilde{s}_t]\|_\star &= \|[F_t(s_{t-1}, u_t) - F_t(\tilde{s}_{t-1}, \tilde{u}_t)]\|_\star \\
&\leq \underbrace{\|[F_t(s_{t-1}, u_t) - F_t(\tilde{s}_{t-1}, u_t)]\|_\star}_{\text{state-to-state term}} + \underbrace{\|[F_t(\tilde{s}_{t-1}, u_t) - F_t(\tilde{s}_{t-1}, \tilde{u}_t)]\|_\star}_{\text{input-to-state term}}. \quad (14)
\end{aligned}
$$

The first term is bounded by the contraction inequality from Theorem 3.1:

$$\|[F_t(s_{t-1}, u_t) - F_t(\tilde{s}_{t-1}, u_t)]\|_\star \leq \gamma \|[s_{t-1} - \tilde{s}_{t-1}]\|_\star. \quad (15)$$

**Input-to-state Lipschitz bound.** We show that the mapping $u \mapsto F_t(\tilde{s}_{t-1}, u)$ is $\beta$–Lipschitz in the block norm with $\beta = \lambda \|W_{\mathrm{in}}\|_2$. By construction of TRIGR, only the neuronal component depends directly on the current input $u_t$; the calcium and gliotransmitter updates use *delayed* states ($x_{t-1}$ and $c_{t-1}$, respectively) and thus have no instantaneous dependence on $u_t$. Let $\tilde{s}_{t-1}$ be fixed and denote by $x_t(u)$ the neuronal update at input $u$:

$$x_t(u) = (1 - \lambda)\tilde{x}_{t-1} + \lambda \tanh\big(W_r \tilde{x}_{t-1} + W_{\mathrm{in}} u + B_g \tilde{g}_{t-1} + b\big).$$

For any $u, \tilde{u}$,

$$\|x_t(u) - x_t(\tilde{u})\|_2 \leq \lambda \big\| \tanh(z + W_{\mathrm{in}}(u - \tilde{u})) - \tanh(z) \big\|_2 \leq \lambda \|W_{\mathrm{in}}\|_2 \|u - \tilde{u}\|_2,$$

where $z = W_r \tilde{x}_{t-1} + B_g \tilde{g}_{t-1} + b$ and we used that $\tanh$ is 1–Lipschitz. Since $c_t$ and $g_t$ are independent of $u_t$ at this step, their differences vanish. Therefore, in the block norm,

$$\|[F_t(\tilde{s}_{t-1}, u_t) - F_t(\tilde{s}_{t-1}, \tilde{u}_t)]\|_\star = \max\{\|x_t(u_t) - x_t(\tilde{u}_t)\|_2, 0, 0\} \leq \beta \|u_t - \tilde{u}_t\|_2. \quad (16)$$

**Combining bounds and unrolling the recursion.** Plugging (15) and (16) into (14) yields the one–step inequality

$$\|[s_t - \tilde{s}_t]\|_\star \leq \gamma \|[s_{t-1} - \tilde{s}_{t-1}]\|_\star + \beta \|u_t - \tilde{u}_t\|_2 \qquad (t \geq 1).$$

Define $a_t \coloneqq \|[s_t - \tilde{s}_t]\|_\star$ and $e_t \coloneqq \|u_t - \tilde{u}_t\|_2$. By induction,

$$a_t \leq \gamma^t a_0 + \sum_{k=1}^{t} \gamma^{t-k} \beta e_k.$$

With identical initialization $a_0 = 0$ and $\Delta_u \coloneqq \sup_{k \geq 1} e_k$, we obtain

$$a_t \leq \beta \Delta_u \sum_{k=1}^{t} \gamma^{t-k} = \beta \Delta_u \frac{1 - \gamma^t}{1 - \gamma} \leq \frac{\beta}{1 - \gamma} \Delta_u,$$

and hence $\sup_{t \geq 1} \|[s_t - \tilde{s}_t]\|_\star \leq \frac{\beta}{1-\gamma} \Delta_u$, proving (12).

**Input–output gain bound.** Let $y_t = W_{\mathrm{out}} \Phi(s_t)$ and $\tilde{y}_t = W_{\mathrm{out}} \Phi(\tilde{s}_t)$, where $\Phi$ is $L_\Phi$–Lipschitz on the forward–invariant state set (Remark A.2). Then

$$\|y_t - \tilde{y}_t\|_2 \leq \|W_{\mathrm{out}}\|_2 \|\Phi(s_t) - \Phi(\tilde{s}_t)\|_2 \leq \|W_{\mathrm{out}}\|_2 L_\Phi \|[s_t - \tilde{s}_t]\|_\star.$$

Taking the supremum over $t$ and substituting the bound from (12) yields

$$\sup_{t \geq 1} \|y_t - \tilde{y}_t\|_2 \leq \frac{\|W_{\mathrm{out}}\|_2 L_\Phi \beta}{1 - \gamma} \Delta_u,$$

which is (13).

$\square$

*Proof of Proposition 3.3.* Let $S \coloneqq (\mathbb{R}^{N+2M}, \|\cdot\|_\star)$ with $\|[s]\|_\star = \max\{\|x\|_2, \|c\|_2, \|g\|_2\}$ and $U \coloneqq (\mathbb{R}^{d_{\mathrm{in}}}, \|\cdot\|_2)$. Consider the *augmented* space $S \oplus U$ endowed with the block norm

$$\|(s, u)\|_\bullet \coloneqq \max\{\|s\|_\star, \|u\|_2\}.$$

Define the closed–loop one–step map $\mathcal{H} : S \oplus U \to S \oplus U$ by

$$(s_t, u_{t+1}) = \mathcal{H}(s_{t-1}, u_t) \overset{\mathrm{def}}{=} \Big(F(s_{t-1}, u_t), \; G(F(s_{t-1}, u_t))\Big), \qquad G(s) \coloneqq E W_{\mathrm{out}} \Phi(s),$$

where $F$ is the TRIGR state update $s_t = F(s_{t-1}, u_t)$ defined in §3. By Theorem 3.1, for all $(s_{t-1}, u_t)$ the partial derivatives of $F$ satisfy

$$\left\|\tfrac{\partial F}{\partial s}\right\|_{S \to S} \leq \gamma < 1, \qquad \left\|\tfrac{\partial F}{\partial u}\right\|_{U \to S} \leq \beta \quad \text{with } \beta = \lambda \|W_{\mathrm{in}}\|_2,$$

where the operator norms $\|\cdot\|_{S \to S}$ and $\|\cdot\|_{U \to S}$ are induced by the domain/range norms $\|\cdot\|_\star$ and $\|\cdot\|_2$. Moreover, since $\Phi$ is $L_\Phi$–Lipschitz on the forward–invariant state set and $G(s) = E W_{\mathrm{out}} \Phi(s)$, we have for all $s$

$$\left\|\tfrac{\partial G}{\partial s}\right\|_{S \to U} \leq \|E W_{\mathrm{out}}\|_2 L_\Phi =: L_{\mathrm{fb}}.$$

**Jacobian structure and block–row bounds.** By the chain rule, the Jacobian of $\mathcal{H}$ at $(s_{t-1}, u_t)$ has the $2 \times 2$ block form

$$J_{\mathcal{H}} = \begin{bmatrix} A & B \\ C & D \end{bmatrix} = \begin{bmatrix} \dfrac{\partial F}{\partial s} & \dfrac{\partial F}{\partial u} \\ \dfrac{\partial G}{\partial s} \dfrac{\partial F}{\partial s} & \dfrac{\partial G}{\partial s} \dfrac{\partial F}{\partial u} + \underbrace{\dfrac{\partial G}{\partial u}}_{= 0} \end{bmatrix}.$$

Here $D = 0$ because $G$ depends on $(s_t)$ but not directly on $u_t$. For the *induced* norm on $S \oplus U$ one has the standard block–row bound (Horn & Johnson, 2012, Chap. 5):

$$\|J_{\mathcal{H}}\|_\bullet \leq \max\Big\{ \|A\|_{S \to S} + \|B\|_{U \to S}, \; \|C\|_{S \to U} + \|D\|_{U \to U} \Big\}.$$

Using the bounds above and submultiplicativity,

$$\|A\|_{S \to S} \leq \gamma, \quad \|B\|_{U \to S} \leq \beta, \quad \|C\|_{S \to U} = \left\|\tfrac{\partial G}{\partial s} \tfrac{\partial F}{\partial s}\right\|_{S \to U} \leq L_{\mathrm{fb}} \gamma \leq L_{\mathrm{fb}}, \quad \|D\|_{U \to U} = 0.$$

Therefore

$$\|J_{\mathcal{H}}\|_\bullet \leq \max\{\gamma + \beta, \; L_{\mathrm{fb}}\}.$$

**Uniform contraction and consequences.** If $\gamma + \beta < 1$ and $L_{\text{fb}} < 1$, then $\kappa \coloneqq \max\{\gamma + \beta, L_{\text{fb}}\} < 1$ and the estimate above holds *uniformly* over $S \oplus U$. By the mean–value theorem in Banach spaces and the induced–norm bound, for all $(s_{t-1}, u_t)$ and $(\tilde{s}_{t-1}, \tilde{u}_t)$,

$$\|\mathcal{H}(s_{t-1}, u_t) - \mathcal{H}(\tilde{s}_{t-1}, \tilde{u}_t)\|_\bullet \leq \kappa \|(s_{t-1}, u_t) - (\tilde{s}_{t-1}, \tilde{u}_t)\|_\bullet.$$

Hence $\mathcal{H}$ is a $\kappa$–contraction on the complete metric space $(S \oplus U, \|\cdot\|_\bullet)$. By Banach's fixed–point theorem, there exists a unique fixed point $z^\star = (s^\star, u^\star)$ with $\mathcal{H}(z^\star) = z^\star$, and for any initialization $z_0$,

$$\|z_t - z^\star\|_\bullet \leq \kappa^t \|z_0 - z^\star\|_\bullet \qquad (t \geq 0).$$

This proves existence and uniqueness of the input–free closed–loop trajectory and exponential stability.

**Fading–memory to perturbations.** If small exogenous perturbations $\delta_t$ are injected additively into the input channel (e.g., $u_{t+1} = E W_{\text{out}} \Phi(s_t) + \delta_t$), the perturbed augmented map $\widehat{\mathcal{H}}$ satisfies $\|\widehat{\mathcal{H}}(z) - \widehat{\mathcal{H}}(\tilde{z})\|_\bullet \leq \kappa \|z - \tilde{z}\|_\bullet$ and $\|\widehat{\mathcal{H}}(z) - \mathcal{H}(z)\|_\bullet \leq \|\delta_t\|_2$. A standard contraction argument yields $\sup_{t \geq 0} \|z_t - \tilde{z}_t\|_\bullet \leq \frac{1}{1-\kappa} \sup_{t \geq 0} \|\delta_t - \tilde{\delta}_t\|_2$, i.e., fading–memory (ISS–type) dependence on perturbations (cf. Desoer & Vidyasagar, 1975). □

**Choice of norms.** We use the operator 2–norms for blocks (largest singular values) because they compose naturally under transposes and are estimated reliably by power iteration; block–$\ell_\infty$ on the outer $3 \times 3$ structure yields a transparent "max row–sum" condition (Horn & Johnson, 2012). Alternatives (e.g., weighted block norms) are possible but were unnecessary in our experiments.

*Remark* A.2 (Lipschitz constant of the feature map). All feature maps implemented in code are Lipschitz on the forward–invariant state set: (i) `"x"`: $\Phi(s) = [x; 1]$ has $L_\Phi \leq 1$; (ii) `"x_c_g"` with optional elementwise squares: since $\|x\|_\infty \leq 1$ (from $\tanh$), $\|g\|_\infty \leq 1$ (clipping), and $c$ is bounded under the contraction hypothesis, the Jacobian of $\psi(z) = z \circ z$ has operator norm $\leq 2\|z\|_\infty$ on each block, hence a finite $L_\Phi$ exists; (iii) `"x_x2_g"`: $\Phi(s) = [x; x \circ x; g; 1]$ has $L_\Phi \leq \sqrt{1 + 4\|x\|_\infty^2 + 1} \leq \sqrt{6}$ globally because $\|x\|_\infty \leq 1$.

**Design guidance from Proposition 3.3.** Conditions in Prop. 3.3 separates concerns: (i) the *plant* budget $\gamma + \beta < 1$ is enforced at initialization by our norm–aware scaling of $(W_r, B_g)$ and by choosing a reasonable input gain $\beta$; (ii) the *feedback* budget $L_{\text{fb}} < 1$ can be enforced by (a) shrinking $\|E W_{\text{out}}\|_2$ (ridge strength, explicit rescaling), (b) using feature maps with small $L_\Phi$ (e.g., avoid squaring $c$), or (c) composing $E$ with a contractive nonlinearity (e.g., $\tanh$) if desired. In practice, $E$ is a selection matrix with $\|E\|_2 \leq 1$, so $L_{\text{fb}} \leq \|W_{\text{out}}\|_2 L_\Phi$.

**Robustness corollary (tolerance to operator perturbations).** If the open–loop row budgets satisfy $\max\{\mathsf{C}_x, \mathsf{C}_c, \mathsf{C}_g\} \leq 1 - \mu$ for some margin $\mu \in (0, 1)$, then any perturbations $(\Delta W_r, \Delta B_g, \Delta H, \Delta W_{\text{rel}}, \Delta D)$ that increase the respective block norms by at most $\mu/3$ each preserve $\gamma < 1$ by subadditivity of operator norms. An analogous statement holds for the closed loop with margins on $(\gamma + \beta)$ and $L_{\text{fb}}$. This follows directly from the block–row sum characterization and triangle inequality (Horn & Johnson, 2012).

A.2 SUMMARY OF THE PROPOSED METHOD

TRIGR augments a standard ESN with a slow, spatially coherent "memory field" carried by an astrocyte lattice. Each time step (Algorithm 1), the fast neuronal core updates like a leaky ESN but with an *additive* astrocytic bias; in parallel, neurons emit nonnegative release proxies that are locally pooled across the lattice, where a saturating reaction–diffusion update integrates and transports activity as wave-like Ca$^{2+}$ traces. A delayed, bounded gliotransmitter then feeds back to neurons on the next step, closing a bidirectional loop. The deliberate one-step delays (neuron↔glia) make the joint Jacobian block-cyclic, so the overall Lipschitz gain decouples neatly into three "row budgets" (neurons, calcium, glio). When these budgets are set below one, the entire coupled system is a uniform contraction—hence it possesses a unique input-causal trajectory and forgets initial conditions exponentially fast. Intuitively: the lattice acts like a gentle, low-pass, spatially smoothing memory that enriches dynamics without destabilizing the recurrent core.

To make this certificate *constructive*, we initialize by measuring operator norms and scaling once so the budgets are satisfied (Algorithm 2). Concretely, we jointly scale the diffusion $D$ and pooling $H$ to keep the astrocyte row under its target, build a footprint-aligned feedback $B_g = \eta(HW_{\mathrm{rel}})^\top$, and then balance the neuron row by sharing gain between $W_r$ and $B_g$. With the reservoir fixed and certified, learning is just ridge regression on features built from $(x, c, g)$ (Algorithm 3); evaluation can be open-loop or fully autoregressive by feeding predicted outputs back as inputs (Algorithm 4). Practically, the lattice costs only a 5-point stencil per step, so compute remains dominated by the usual ESN multiply—yet the model now carries long-horizon, spatially coherent fading memory backed by an explicit stability guarantee.

---

**Algorithm 1** TRIGR_STEP — One forward step of the Tripartite Glia–Neuron Reservoir

---

**Require:** Previous state $s_{t-1} = [x_{t-1}; c_{t-1}; g_{t-1}]$, input $u_t \in \mathbb{R}^{d_{\mathrm{in}}}$; fixed operators $W_r, W_{\mathrm{in}}, W_{\mathrm{rel}}, H, B_g, L$; leaks/rates $\lambda, \alpha, \rho$; nonlinearities $\phi = \tanh$, $\rho_{\mathrm{rel}}(z) = \max(z, 0)$, $F(z) = c_{\max}\sigma(k_c(z - \theta_c))$, $\Gamma(z) = \sigma(k_g(z - \theta_g))$; tonic drive $\zeta \in \mathbb{R}$.
**Ensure:** New state $s_t = [x_t; c_t; g_t]$.
  1: **(Neural preactivation)** $a_t \leftarrow W_r x_{t-1} + W_{\mathrm{in}} u_t + B_g g_{t-1} + b$
  2: **(Neural update)** $x_t \leftarrow (1 - \lambda)x_{t-1} + \lambda \phi(a_t)$
  3: **(Release proxy)** $q_{t-1} \leftarrow \rho_{\mathrm{rel}}(W_{\mathrm{rel}} x_{t-1})$
  4: **(Pooling)** $\mathrm{drive}_{t-1} \leftarrow H q_{t-1}$  ▷ $H$ nonnegative, row-stochastic
  5: **(Laplacian via 5-point stencil)** $\ell_{t-1} \leftarrow L c_{t-1}$  ▷ $O(M)$ stencil on $m_x \times m_y$ grid
  6: **(Astrocyte Ca²⁺)** $c_t \leftarrow (1 - \alpha)c_{t-1} + \alpha\left[F(\mathrm{drive}_{t-1}) + D\ell_{t-1} + \zeta\right]$
  7: **if** $g$-mode is `linear` **then**
  8:    $g_t \leftarrow (1 - \rho)g_{t-1} + \rho\Gamma(c_{t-1})$
  9: **else if** $g$-mode is `depletion` **then**
 10:    $g_t \leftarrow g_{t-1} + \rho\left[\Gamma(c_{t-1})(1 - g_{t-1}) - \delta g_{t-1}\right]$  ▷ $\delta \in (0, 1)$
 11: **end if**
 12: **(Bounding)** $g_t \leftarrow \mathrm{clip}(g_t, 0, 1)$
 13: **return** $s_t = [x_t; c_t; g_t]$

---

**Algorithm 2** TRIGR_INIT — Norm-aware initialization with certified row budgets

---

**Require:** Sizes $N, m_x, m_y, P$; random scale $\sigma$; target budgets $\rho_x^{\mathrm{target}}, \rho_c^{\mathrm{target}} < 1$; rates $\lambda, \alpha$; non-linearity slopes $L_F = \frac{k_c c_{\max}}{4}$.
**Ensure:** Scaled $W_r, H, B_g, D$ meeting the sufficient row bounds; fixed $W_{\mathrm{rel}}, W_{\mathrm{in}}$.
  1: Sample $W_r \in \mathbb{R}^{N \times N}$ (optionally sparse), $W_{\mathrm{rel}} \in \mathbb{R}^{P \times N}$, $W_{\mathrm{in}} \in \mathbb{R}^{N \times d_{\mathrm{in}}}$
  2: Sample $H \in \mathbb{R}^{M \times P}$ i.i.d. nonnegative, sparsify, then row-normalize ($\sum_j H_{ij} = 1$)
  3: Choose initial $D > 0$; construct 4-neighbor Laplacian operator $L$ (5-point stencil)
  4: Estimate norms by power iteration (Golub & Van Loan, 2013):
      $n_L \approx \|L\|_2, \quad n_{HW} \approx \|HW_{\mathrm{rel}}\|_2$
  5: **(Ca-row scaling)** $s_c \leftarrow \dfrac{\rho_c^{\mathrm{target}}}{D\, n_L + L_F\, n_{HW} + \varepsilon}$;  $D \leftarrow s_c D$;  $H \leftarrow s_c H$
  6: **(Feedback footprint)** $B_g \leftarrow \eta\left(HW_{\mathrm{rel}}\right)^\top \in \mathbb{R}^{N \times M}$
  7: Estimate $n_{Wr} \approx \|W_r\|_2, n_{Bg} \approx \|B_g\|_2$
  8: **(Neuron-row scaling)** $s_x \leftarrow \dfrac{\rho_x^{\mathrm{target}}}{n_{Wr} + n_{Bg} + \varepsilon}$;  $W_r \leftarrow s_x W_r$;  $B_g \leftarrow s_x B_g$
  9: **(Certificate check)** verify
      $$\mathsf{C}_x = (1 - \lambda) + \lambda(\|W_r\|_2 + \|B_g\|_2) < 1, \quad \mathsf{C}_c = (1 - \alpha) + \alpha(D\|L\|_2 + L_F\|HW_{\mathrm{rel}}\|_2) < 1$$
 10: **return** $\{W_r, W_{\mathrm{in}}, W_{\mathrm{rel}}, H, B_g, D\}$

---

### A.3 Neuroscientific Motivation

A central design choice in TRIGR is to endow an ESN-like neuronal core with a *slow, spatially structured* field that is driven by neuronal activity and feeds back modulatory signals. This mirrors the *tripartite synapse* view, wherein astrocytes are computational partners at synapses: neuronal spiking

---

**Algorithm 3** TRIGR_TRAIN — Ridge readout training (teacher-forced)

---

**Require:** Initialized reservoir; training pairs $\{(u_t, y_t)\}_{t=1}^{T}$; discard $t_{\mathrm{disc}}$; feature map $\Phi(s)$; ridge $\alpha_{\mathrm{ridge}} > 0$.
**Ensure:** Readout $W_{\mathrm{out}} \in \mathbb{R}^{d_{\mathrm{out}} \times F}$.
 1: Initialize state $s_0$ (e.g., zero)
 2: **for** $t = 1$ **to** $T$ **do**
 3:     $s_t \leftarrow \mathrm{TRIGR\_STEP}(s_{t-1}, u_t)$
 4:     **if** $t \geq t_{\mathrm{disc}}$ **then**
 5:         $X \leftarrow \begin{bmatrix} X \\ \Phi(s_t) \end{bmatrix}; \quad Y \leftarrow \begin{bmatrix} Y \\ y_t \end{bmatrix}$
 6:     **end if**
 7: **end for**
 8: **(Ridge fit)** $W_{\mathrm{out}}^{\top} \leftarrow (X^{\top} X + \alpha_{\mathrm{ridge}} I)^{-1} X^{\top} Y$          ▷ closed form; code uses `Ridge`
 9: **return** $W_{\mathrm{out}}$

---

---

**Algorithm 4** TRIGR_EVALUATE — Open-loop and autoregressive prediction

---

**Require:** Trained $(W_{\mathrm{out}})$; feature map $\Phi$; selector $E \in \mathbb{R}^{d_{\mathrm{in}} \times d_{\mathrm{out}}}$; horizon $T'$.
**Ensure:** Predicted outputs $\{\hat{y}_t\}$ (open loop) or $\{\hat{y}_t\}$ with fed-back inputs (autoregressive).
 1: **Open loop:** given test inputs $\{u_t\}_{t=1}^{T'}$,
 2: **for** $t = 1$ **to** $T'$ **do**
 3:     $s_t \leftarrow \mathrm{TRIGR\_STEP}(s_{t-1}, u_t)$
 4:     $\hat{y}_t \leftarrow W_{\mathrm{out}} \Phi(s_t)$
 5: **end for**
 6: **Autoregressive:** given $u_1$,
 7: **for** $t = 1$ **to** $T'$ **do**
 8:     $s_t \leftarrow \mathrm{TRIGR\_STEP}(s_{t-1}, u_t)$
 9:     $\hat{y}_t \leftarrow W_{\mathrm{out}} \Phi(s_t)$
10:     $u_{t+1} \leftarrow E \hat{y}_t$          ▷ feed back first $d_{\mathrm{in}}$ components
11: **end for**

---

and glutamatergic transmission elevate astrocytic $\mathrm{Ca}^{2+}$ and can trigger the release of gliotransmitters that, in turn, modulate synaptic efficacy and neuronal excitability on timescales longer than those of fast synaptic currents (Araque et al., 1999; Volterra & Meldolesi, 2005; Perea et al., 2009). Astrocytic $\mathrm{Ca}^{2+}$ signals exhibit nonlinear saturation, integrate local synaptic activity, and propagate over astrocyte networks via gap junctions and extracellular messengers (e.g., ATP), producing wave-like dynamics across tens to hundreds of microns and hundreds of milliseconds to seconds (Scemes & Giaume, 2006; Bazargani & Attwell, 2016). These phenomena are naturally captured by *reaction–diffusion* descriptions with saturating nonlinear transduction from neuronal activity to $\mathrm{Ca}^{2+}$ and bounded, sigmoidal release curves for gliotransmitters (De Pittà et al., 2009). In TRIGR, the pooling $H$ and grid Laplacian $L$ instantiate such spatial organization: neuronal release proxies are locally integrated to drive astrocytic $\mathrm{Ca}^{2+}$, which then diffuses through a lattice, creating spatially coherent memory traces that outlast the fast neuronal dynamics while remaining bounded by saturating kinetics (our $F$ and $\Gamma$ maps). The use of *additive* astrocyte→neuron feedback aligns with the predominantly modulatory character attributed to gliotransmission and neuromodulator-like actions on excitability rather than direct replication of fast synaptic currents; it also resonates with stability-oriented modeling where modulatory gains are explicitly bounded.

A second key design aspect is the *one-step delay* imposed on each cross-compartment interaction ($g \to x$, $x \to c$, $c \to g$). Physiologically, neuron→astrocyte coupling (via glutamate spillover and metabotropic receptors) and astrocyte→neuron effects (via gliotransmitters) unfold over slower transduction cascades than synaptic transmission, introducing effective latencies and temporal coarse-graining relative to the millisecond neuronal membrane dynamics (Volterra & Meldolesi, 2005; Perea et al., 2009; Bazargani & Attwell, 2016). Modeling these as one-step delays both reflects the separation of timescales and leads to a *block-cyclic Jacobian* amenable to rigorous contractive analysis. Importantly, our formulation remains agnostic to ongoing debates about the prevalence and mechanisms of gliotransmission (Hamilton & Attwell, 2010; Sloan & Barres, 2014; Savtchouk

& Volterra, 2018): TRIGR *does not* assume strong, rapid, synapse-like astrocyte outputs. Instead, it posits a bounded, Lipschitz modulatory pathway that is consistent with converging evidence for glial influence on network excitability and plasticity over slow timescales.

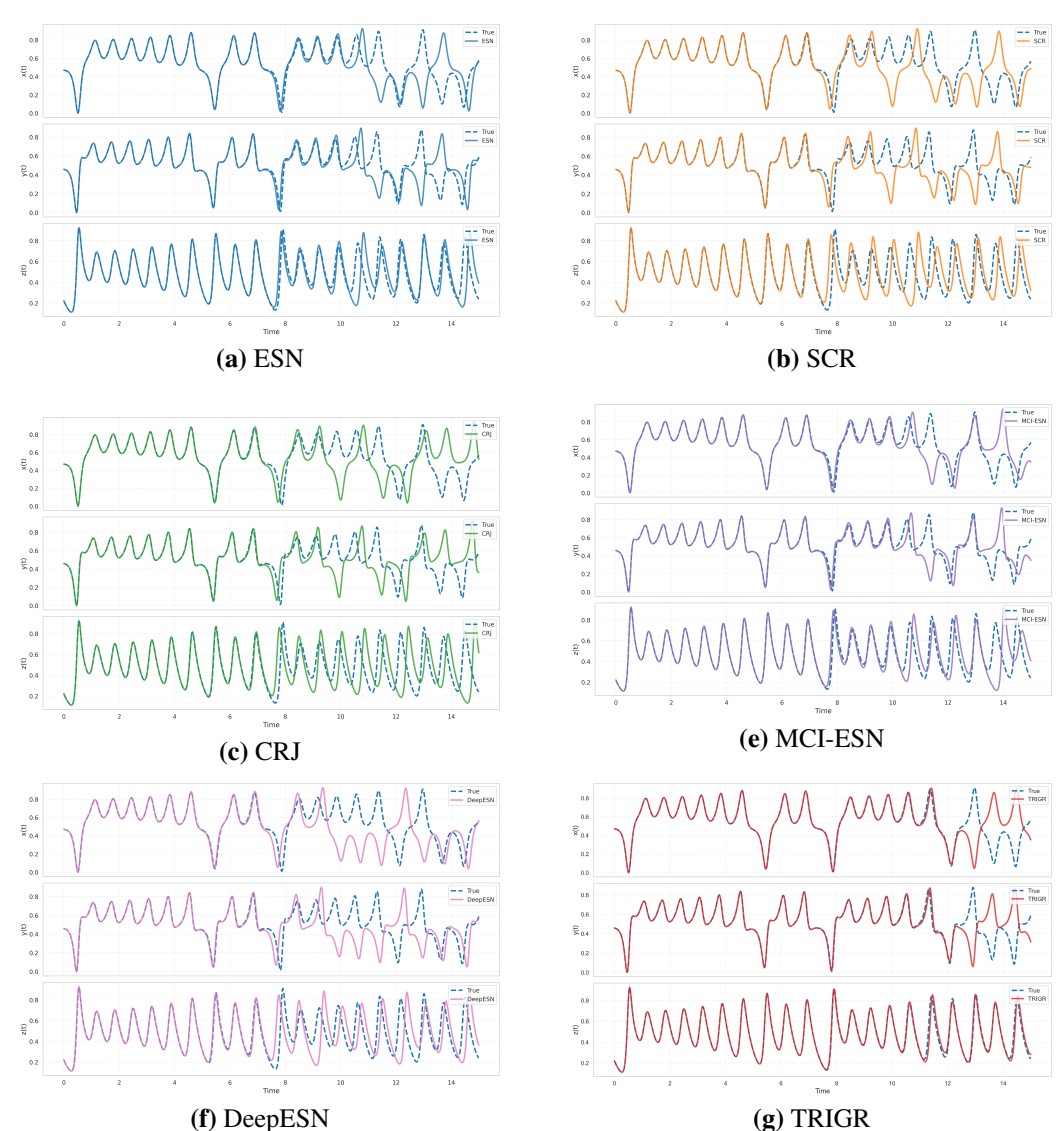

(a) ESN                                (b) SCR

(c) CRJ                                (e) MCI-ESN

(f) DeepESN                              (g) TRIGR

Figure 2: Predicted trajectories by different reservoir architectures alongside the ground truth for the test segment of the Lorenz system under autoregressive forecasting.

# B    DETAILS OF THE EXPERIMENTAL SETUP

## B.1    BASELINES

- **ESN** (Jaeger, 2001). Erdős–Rényi random graph with connection probability $\rho_c$. Spectral radius $\rho_\star$ and input scaling $\|\mathbf{W}_{\mathsf{in}}\|_2$.

- **SCR** (Li et al., 2024). A single directed cycle of length 300 with uniform edge weight $w_c$. We tune $w_c$, which fully determines the spectral radius of the graph.

- **CRJ** (Rodan & Tino, 2012). Cycle reservoir augmented with "jump" connections of fixed distance $J$. We sweep $J$ and edge weight $w_c$, preserving unit-norm in-degree.

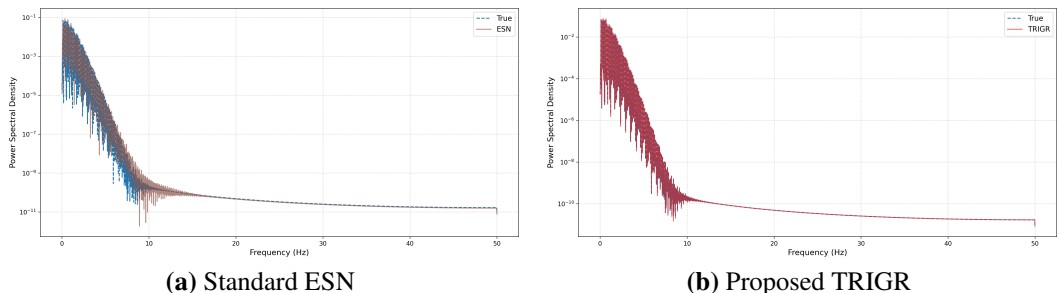

**(a)** Standard ESN                    **(b)** Proposed TRIGR

Figure 3: PSD plots of autoregressive predictions at a 3000-step horizon for (a) a standard ESN and (b) the proposed TRIGR, when both networks are driven by the Rössler system.

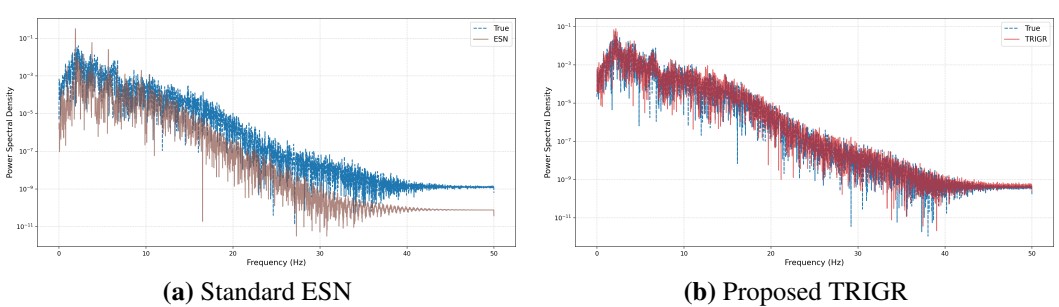

**(a)** Standard ESN                    **(b)** Proposed TRIGR

Figure 4: PSD plots of autoregressive predictions at a 1000-step horizon for (a) a standard ESN and (b) the proposed TRIGR, when both networks are driven by the Chen system.

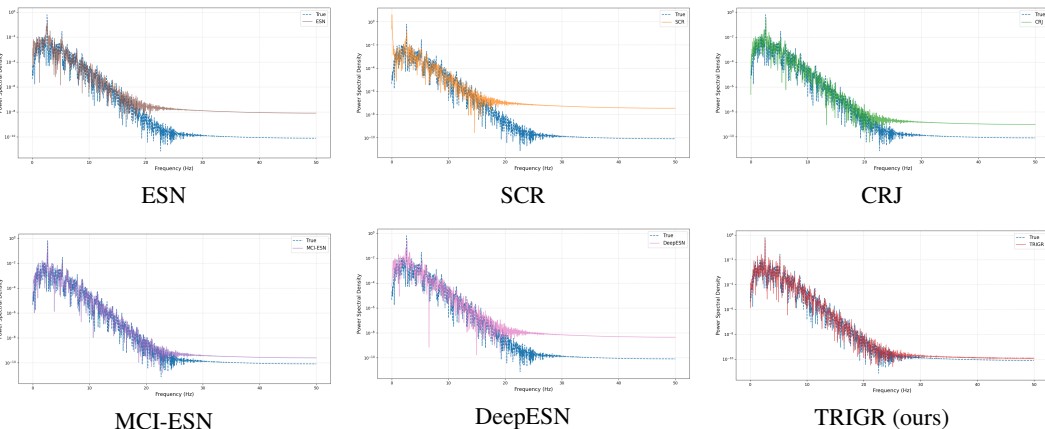

ESN                     SCR                     CRJ

MCI-ESN                 DeepESN              TRIGR (ours)

Figure 5: PSD of the $z$–coordinate of Lorenz-63 for the proposed model TRIGR and five baseline reservoirs. Each plot compares the spectrum produced by autonomous rollout (coloured) with the true spectrum (blue). TRIGR best preserves the spectral envelope and high-frequency decay, whereas baselines show elevated noise floors and spectral broadening, indicating drift from the true attractor.

- **MCI-ESN** (Liu et al., 2024). A two-core architecture with a minimal interaction regime, where 150-unit ESN are coupled through sparse cross-connections. The intra-core spectral radii $(\mu, \eta)$ and the inter-core mixing coefficient $\theta$ follow the Cartesian grid.

- **DeepESN** (Gallicchio & Micheli, 2017b). A stack of three leaky reservoirs ($N_1 = N_2 = N_3 = 100$). All layers share an input scale $\|\mathbf{W}_{\text{in}}\|_2$.

- **Small-World Topology ESN** (Kawai et al., 2019). As an additional comparator we implement the SW-ESN baseline: an echo-state network whose recurrent matrix is wired according to the Watts–Strogatz small-world recipe rather than with i.i.d. Gaussian weights.

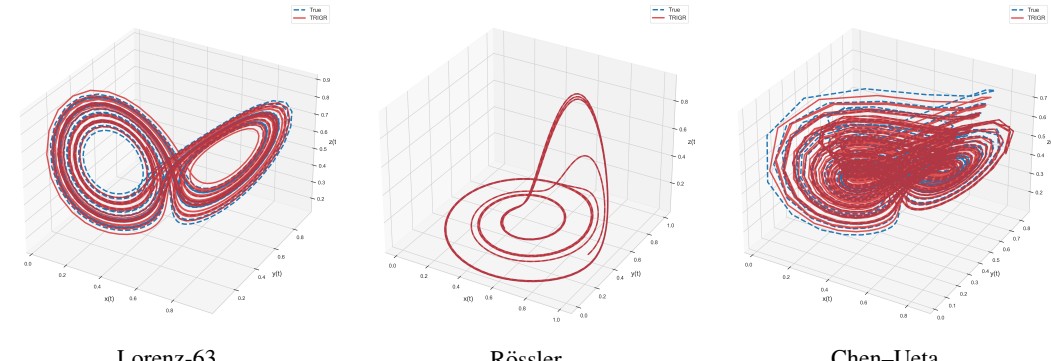

| Lorenz-63 | Rössler | Chen–Ueta |

Figure 6: Three-dimensional phase portraits generated by TRIGR for Lorenz-63, Rössler, and Chen–Ueta over a $1\,000$-step autonomous rollout (red), overlaid on the reference attractors (blue dots). Close overlap confirms that TRIGR preserves the global geometry of all three chaotic systems.

Starting from a regular ring lattice we rewire each edge with probability $p_{\text{rewire}}$ to obtain a sparse graph that combines short average path length with high clustering; the non-zero weights are then drawn from $\mathcal{N}(0,1)$ and rescaled to the prescribed spectral radius so that the mean degree matches the 300-unit budget used for all models. Following the original protocol, only 10 randomly chosen neurons receive external input and another disjoint set of 10 neurons feed the read-out, forcing information to propagate through the small-world core. The authors showed that such topology expands the range of radii over which the echo-state property is preserved and improves memory capacity and nonlinear-prediction accuracy; our inclusion of SW-ESN therefore tests whether TRIGR's performance gains stem from its tripartite dynamics rather than from architectural advantages that could already be captured by small-world connectivity alone.

- **Temporal-Convolutional Network (TCN)** (Bai et al., 2018). We adopt a strictly causal two-layer TCN with kernel size $k = 3$ and successive dilations $1, 2$, yielding an effective receptive field of 5 steps—the same horizon used by classical NVAR and permitting direct comparison. Each layer performs a 1-D convolution followed by ReLU and explicit trimming of right-hand padding to prevent look-ahead. The channel width is set to $C = 56$, giving $\approx 1.02 \times 10^4$ trainable weights (comparable to ESN/TRIGR). During training a single sequence batch ($B = 1$) is presented under teacher forcing, and the network minimises one-step MSE with Adam. At inference time the final 5 observations initialise an autoregressive roll-out: after every forward pass the oldest sample is dropped and the newest prediction appended, enforcing strict causality throughout the entire horizon.

- **Causal Transformer (CT)** (Vaswani et al., 2017). We use a compact transformer encoder with one self-attention block ($d_{\text{model}} = 32$, $n_{\text{head}} = 4$, feed-forward width $d_{\text{ff}} = 64$). A fixed sinusoidal positional embedding of length $L$ is added to the token sequence produced by a linear projection of the 3-D input; causal masking is implicit because we retain only the final token for prediction. This configuration has $\approx 8.8k$ parameters, on the same order as TRIGR. Training uses sliding windows of length $L = 20$ extracted with stride 1; mini-batches of 64 windows optimise an MSE objective for 60 epochs with Adam ($\eta = 2 \times 10^{-3}$). Free-running prediction proceeds autoregressively: the most recent $L$ samples are re-encoded at every step, the last token's output is mapped back to $\mathbb{R}^3$, and the window is advanced by one, ensuring the transformer's causal field mirrors its training context.

- **Non-linear Vector Auto-Regression (NVAR)** (Farmer & Sidorowich, 1987). This baseline approximates the Lorenz flow with a fixed-length delay line followed by a quadratic polynomial map—an approach long used for low-order chaotic forecasting. A window of the most recent $k = 26$ samples is flattened, normalised and lifted to a feature vector $\phi = [1; , x; , x^{\otimes 2}] \in \mathbb{R}^{3160}$, where the 3081 quadratic monomials are taken with replacement so that $\phi$ spans all terms up to degree two. Closed-form ridge regression then yields a read-out matrix $W_{\text{out}} \in \mathbb{R}^{3 \times 3160}$ with 9,480 trainable coefficients; the remainder of the

model is completely fixed. At test time the predictor is rolled out autoregressively: each new estimate is appended to the delay line and immediately fed back through the same quadratic map, giving a strictly causal, non-parametric competitor.

- **LSTM** (Hochreiter & Schmidhuber, 1997b). Our recurrent baseline is a single-layer long short-term memory network with hidden width $h = 48$, followed by a linear read-out to $\mathbb{R}^3$. This architecture contains $\approx 1.01 \times 10^4$ learnable weights, comparable to TRIGR's readout scale. Training proceeds under teacher forcing on the full sequence, minimising mean-squared error with Adam ($\eta = 10^{-3}$, 80 epochs). During inference the network operates in closed loop: the last predicted vector is recurrently fed back as the next input, so the LSTM must internally preserve all temporal context beyond a single step. This setting tests how much long-horizon accuracy can be extracted from a conventional gated RNN under a comparable parameter budget.

**Evaluation Protocols.** Each model is assessed under two complementary regimes. In *open-loop (teacher-forced) mode* we present the ground-truth input $u_t$ at every step and record the one-step-ahead prediction $\hat{y}_t = W_{\text{out}}\phi(z_t)$; error metrics (NRMSE, etc.) are accumulated over the full test horizon, thereby isolating the reservoir's instantaneous regression capability from its long-term stability. In *closed-loop (autoregressive) mode* we supply the reservoir with a single "seed" input $u_0$ (or the last training target) and thereafter feed back its own prediction, i.e. $u_{t+1} \leftarrow \hat{y}_t[: d_{\text{in}}]$; the network thus runs freely for $T_{\text{roll}}$ steps. Metrics that depend on horizon—VPT, ADev—are evaluated on this rollout, and the run is terminated early if $\|\hat{y}_t\|_2$ diverges. For chaotic ODE benchmarks the Lyapunov time $T_L$ is pre-computed from the training trajectory so that VPT is reported in normalised units, while for real data the same $T_{\text{roll}} = 10\,500$ length and wash-out $T_{\text{wo}} = 2\,000$ are used across models to ensure strict comparability between open- and closed-loop scores. For the synthetic flows we integrate the governing ODEs with LSODA at a fixed step $\Delta t = 0.02$ for 12,500 iterations, discarding the first 2,000 points as reservoir wash-out. Each real-world signal undergoes unity-based (min-max) normalisation to $[0, 1]$, followed by trimming and windowing to a common post-wash-out length so that all datasets contribute an equal number of training samples.

| Dataset | H | NRMSE ↓ | | | | | |
|---------|---|---------|---|---|---|---|---|
| | | LSTM | NVAR | TCN | CT | SW | TRIGR |
| Lorenz | 200 | 1.5021 ± 0.2975 | 1.8118 ± 0.2770 | 2.0728 ± 0.4909 | 1.8575 ± 0.4471 | 0.0085 ± 0.0124 | **0.0005 ± 0.0007** |
| | 400 | 1.4470 ± 0.1115 | 1.7951 ± 0.1170 | 2.0174 ± 0.3454 | 1.7925 ± 0.3124 | 0.1149 ± 0.1165 | **0.0086 ± 0.0111** |
| | 600 | 1.4078 ± 0.1324 | 1.7604 ± 0.1694 | 2.0294 ± 0.3488 | 1.7954 ± 0.3000 | 0.6583 ± 0.2057 | **0.2345 ± 0.1853** |
| | 800 | 1.3855 ± 0.0530 | 1.7547 ± 0.0613 | 1.9858 ± 0.2889 | 1.8498 ± 0.6277 | 0.9029 ± 0.1454 | **0.6281 ± 0.1820** |
| | 1000 | 1.3768 ± 0.0413 | 1.7472 ± 0.0554 | 1.9718 ± 0.2882 | 4.6502 ± 17.1965 | 1.0296 ± 0.1210 | **0.8355 ± 0.1375** |
| CU | 200 | 1.3668 ± 0.0312 | 0.3778 ± 0.3737 | 2.1236 ± 0.1336 | 1.8029 ± 0.1855 | 0.2846 ± 0.2198 | **0.0965 ± 0.1910** |
| | 400 | 1.3477 ± 0.0184 | 1.1689 ± 0.2202 | 2.0650 ± 0.1395 | 1.7787 ± 0.1711 | 0.9629 ± 0.1614 | **0.7577 ± 0.2297** |
| | 600 | 1.3405 ± 0.0135 | 1.3418 ± 0.1618 | 2.0797 ± 0.1295 | 1.7790 ± 0.1480 | 1.1281 ± 0.1162 | **0.9971 ± 0.1418** |
| | 800 | 1.3379 ± 0.0108 | 1.4423 ± 0.1207 | 2.0797 ± 0.1248 | 1.7700 ± 0.1321 | 1.2178 ± 0.1073 | **1.1028 ± 0.1265** |
| | 1000 | 1.3370 ± 0.0101 | 1.4930 ± 0.0964 | 2.0589 ± 0.1174 | 1.7605 ± 0.1253 | 1.3639 ± 0.0878 | **1.1750 ± 0.0960** |
| Rössler | 200 | 1.3634 ± 0.0294 | 0.0009 ± 0.0014 | 1.3910 ± 0.1762 | 2.6887 ± 2.2269 | 0.0902 ± 0.2479 | **0.0004 ± 0.0003** |
| | 400 | 1.3427 ± 0.0160 | 0.0037 ± 0.0097 | 1.7579 ± 0.2884 | 2.3560 ± 1.5485 | 0.3059 ± 0.6997 | **0.0008 ± 0.0008** |
| | 600 | 1.3382 ± 0.0110 | 0.0075 ± 0.0129 | 1.8579 ± 0.3283 | 2.3099 ± 1.5441 | 0.6004 ± 1.1110 | **0.0015 ± 0.0023** |
| | 800 | 1.3358 ± 0.0089 | 0.0113 ± 0.0173 | 1.8588 ± 0.1962 | 2.3416 ± 1.6809 | 0.8208 ± 1.3641 | **0.0020 ± 0.0033** |
| | 1000 | 1.3340 ± 0.0067 | 0.0163 ± 0.0217 | 1.8377 ± 0.2050 | 2.6343 ± 2.4046 | 1.0549 ± 1.5334 | **0.0026 ± 0.0046** |

Table 3: NRMSE (*mean ± s.d.*) on the canonical chaotic benchmarks (Lorenz-63, Rössler and Chen–Ueta, denoted CU) for five closed-loop horizons $H$. Scores are averaged over $5 \times 3 \times 3 = 45$ runs (5 seeds, 3 initial states, 3 splits).

## B.2 DATASETS

Our experiments use seven benchmarks that fall into two families: (1) deterministic chaotic flows governed by ordinary differential equations (ODEs), and (2) recorded physiological or geophysical time-series.

### 1. Canonical Chaotic Benchmarks

**Lorenz–63.** The seminal three-dimensional convection model with canonical parameters $(\sigma, \rho, \beta) = (10, 28, 8/3)$ (Lorenz, 1963).

| Dataset | H | NRMSE ↓ | | | | | |
|---|---|---|---|---|---|---|---|
| | | LSTM | NVAR | TCN | CT | SW | TRIGR |
| MITBIH | 300 | $0.9457 \pm 0.0295$ | $1.3299 \pm 0.0000$ | $1.0153 \pm 0.0088$ | $5.0890 \pm 3.2774$ | $2.8128 \pm 0.8145$ | **$0.7716 \pm 0.0614$** |
| | 600 | $0.9424 \pm 0.0305$ | $1.3282 \pm 0.2698$ | $1.0351 \pm 0.0053$ | $5.0835 \pm 3.2952$ | $2.0237 \pm 0.5672$ | **$0.7591 \pm 0.0329$** |
| | 1000 | $0.9479 \pm 0.0284$ | $1.3038 \pm 0.1385$ | $1.0202 \pm 0.0078$ | $4.6657 \pm 2.9783$ | $1.5004 \pm 0.3852$ | **$0.7616 \pm 0.0169$** |
| BIDMC | 300 | $1.1817 \pm 0.0077$ | $1.0197 \pm 0.3307$ | $1.1275 \pm 0.0156$ | $1.5787 \pm 0.2415$ | $0.7552 \pm 0.0617$ | **$0.3503 \pm 0.0200$** |
| | 600 | $1.1786 \pm 0.0034$ | $0.9839 \pm 0.1421$ | $1.1192 \pm 0.0146$ | $1.6443 \pm 0.2255$ | $0.7577 \pm 0.0552$ | **$0.3736 \pm 0.0170$** |
| | 1000 | $1.1727 \pm 0.0073$ | $1.0347 \pm 0.0854$ | $1.1043 \pm 0.0178$ | $1.6652 \pm 0.2292$ | $0.7606 \pm 0.0490$ | **$0.4343 \pm 0.0137$** |
| Sunspot | 300 | $0.6211 \pm 0.0074$ | $0.4103 \pm 0.0013$ | $0.4640 \pm 0.0028$ | $0.2716 \pm 0.0040$ | $0.5902 \pm 0.0414$ | **$0.2439 \pm 0.0005$** |
| | 600 | $0.7709 \pm 0.0076$ | $0.3348 \pm 0.0113$ | $0.3728 \pm 0.0022$ | $0.2198 \pm 0.0050$ | $0.6185 \pm 0.0454$ | **$0.1960 \pm 0.0004$** |
| | 1000 | $0.7585 \pm 0.0082$ | $0.3342 \pm 0.0192$ | $0.3734 \pm 0.0023$ | $0.2198 \pm 0.0053$ | $0.6078 \pm 0.0446$ | **$0.1942 \pm 0.0004$** |
| Santa Fe | 300 | $0.8237 \pm 0.1255$ | $1.4693 \pm 0.0013$ | $0.3961 \pm 0.0120$ | $3.7102 \pm 1.9078$ | $0.2117 \pm 0.0024$ | **$0.1600 \pm 0.0003$** |
| | 600 | $0.6350 \pm 0.1552$ | $1.3915 \pm 0.0113$ | $0.2928 \pm 0.0066$ | $2.4128 \pm 1.2258$ | $0.1583 \pm 0.0014$ | **$0.1351 \pm 0.0003$** |
| | 1000 | $0.6829 \pm 0.1478$ | $1.4387 \pm 0.0192$ | $0.3235 \pm 0.0076$ | $2.8235 \pm 1.4310$ | $0.1790 \pm 0.0017$ | **$0.1441 \pm 0.0002$** |

Table 4: NRMSE (*mean ± s.d.*) on real-life benchmarks (MIT–BIH ECG, BIDMC, Sunspot Monthly, Santa Fe-B) for three horizons $H$. Thirty random seeds per entry.

**Rössler.** A three-state chemical feedback oscillator producing a single-scroll attractor; standard parameters $(a, b, c) = (0.2, 0.2, 5.7)$ (Rossler, 1976).

**Chen–Ueta.** A hyper-chaotic modification of the Lorenz family that introduces an additional unstable manifold $(a, b, c) = (35, 3, 28)$ (Chen & Ueta, 1999).

**Integration protocol.** Each system is integrated with LSODA (`scipy.integrate.ode`) at a fixed step $\Delta t = 2 \times 10^{-2}$ s for $12\,500$ steps. The first $2\,000$ samples form the wash-out segment $T_{\text{wo}}$; the remaining $10\,500$ points are split $8\,000 / 1\,000 / 1\,500$ into train / validation / test trajectories (chronological order). Initial conditions are sampled uniformly from $[-1, 1]^3$ and stored to guarantee exact reproducibility.

## 2. Real-World Time-Series

**BIDMC PPG/Resp.** 53 eight-minute records of fingertip photoplethysmogram, impedance respiration and derived numerics sampled at 125 Hz; breath annotations are also provided (Goldberger et al., 2000).[1] Missing values (present in $SpO_2$ traces) are linearly interpolated before normalisation.

**MIT–BIH Arrhythmia.** 48 half-hour two-lead ECG segments digitised at 360 Hz with beat-level arrhythmia labels (Moody & Mark, 2001).

**Santa Fe B.** 20-minute trivariate cardiorespiratory polysomnography (heart-rate, chest volume, $SpO_2$) sampled at 2 Hz; originally released for the Santa Fe time-series competition (Jaeger, 2007).

**SILSO Sunspot Monthly.** The version 2.0 monthly total sunspot index spanning January 1749–June 2025, maintained by the Sunspot Index and Long-term Solar Observations service (World Data Center SILSO, 2020).

**Data-Licence Compliance** The BIDMC and MIT–BIH data are distributed under the PhysioNet Credentialed Health Data License 1.5.0. SILSO data are released under CC BY 4.0; all other datasets are in the public domain.

### B.3 SETUP DETAILS AND EXTENDED ANALYSIS

#### B.3.1 CROSS-VALIDATION PROTOCOL (TIME-AWARE, MULTI-FOLD).

We evaluate and select hyper-parameters with a *time-series cross-validation* scheme using $K=3$ chronological folds in the spirit of `TimeSeriesSplit`. For fold $f \in \{1, 2, 3\}$, we form contiguous, non-overlapping windows $\left( \mathcal{T}_{\text{train}}^{(f)}, \mathcal{T}_{\text{val}}^{(f)}, \mathcal{T}_{\text{test}}^{(f)} \right)$ in temporal order; no sample appears in more than one window within each fold, and no shuffling is ever performed. Standardization (z-normalization) is fit *per fold* on $\mathcal{T}_{\text{train}}^{(f)}$ and applied to $\mathcal{T}_{\text{val}}^{(f)}$ and $\mathcal{T}_{\text{test}}^{(f)}$ only. Model selection minimizes

---

[1] https://physionet.org/content/bidmc/

| Ablation Family | Setting (vs. TRIGR) | ESP Cert. | Rel. Cost ($\times$) | Lorenz (Closed–Loop) | | | Diverged % |
|---|---|---|---|---|---|---|---|
| | | | | NRMSE @ H=600 | VPT ($\uparrow$) | ADev ($\downarrow$) | |
| **Baseline (for ref.)** | TRIGR as in Sec. 3 | $\checkmark$ | 1.00 | **0.234 $\pm$ 0.185** | **11.29 $\pm$ 1.61** | **28.3 $\pm$ 11.3** | 0.0 |
| *(N) Norm targets / contraction margin* | | | | | | | |
| (N1) Tighter budgets | $\rho_x^{\mathrm{tar}} = \rho_c^{\mathrm{tar}} = 0.80$ | $\checkmark$ | 1.00 | 0.260 $\pm$ 0.198 | 10.28 $\pm$ 1.65 | 29.6 $\pm$ 11.7 | 0.0 |
| (N2) Default budgets | $\rho_x^{\mathrm{tar}} = \rho_c^{\mathrm{tar}} = 0.90$ | $\checkmark$ | 1.00 | **0.234 $\pm$ 0.185** | **11.29 $\pm$ 1.61** | **28.3 $\pm$ 11.3** | 0.0 |
| (N3) Looser budgets | $\rho_x^{\mathrm{tar}} = \rho_c^{\mathrm{tar}} = 0.95$ | $\checkmark$ | 1.00 | 0.235 $\pm$ 0.184 | 11.26 $\pm$ 1.67 | 28.6 $\pm$ 11.3 | 2.9 |
| (N4) Supercritical | $\rho_x^{\mathrm{tar}} = \rho_c^{\mathrm{tar}} = 1.02$ | $\times$ | 1.00 | 0.538 $\pm$ 0.334 | 6.64 $\pm$ 2.21 | 57.5 $\pm$ 18.8 | 20.7 |
| *(L) Lattice size & footprint radius (grid steps)* | | | | | | | |
| (L1) Smaller lattice | $M=10$, $r=1$ | $\checkmark$ | 0.65 | 0.252 $\pm$ 0.190 | 10.87 $\pm$ 1.69 | 29.4 $\pm$ 11.6 | 0.0 |
| (L2) Default lattice | $M=20$, $r=2$ | $\checkmark$ | 1.00 | **0.234 $\pm$ 0.185** | **11.29 $\pm$ 1.61** | **28.3 $\pm$ 11.3** | 0.0 |
| (L3) Larger lattice | $M=40$, $r=2$ | $\checkmark$ | 1.85 | 0.235 $\pm$ 0.184 | 11.23 $\pm$ 1.62 | 28.5 $\pm$ 11.3 | 0.0 |
| (L4) Wider pooling | $M=40$, $r=3$ | $\checkmark$ | 1.90 | 0.237 $\pm$ 0.186 | 11.02 $\pm$ 1.71 | 28.9 $\pm$ 11.6 | 0.0 |
| *(S) Reservoir sparsity (density of $W_r$); baseline density = 10%* | | | | | | | |
| (S1) Extra sparse | dens. 1% | $\checkmark$ | 0.20 | 0.283 $\pm$ 0.203 | 10.12 $\pm$ 1.69 | 32.0 $\pm$ 12.3 | 0.0 |
| (S2) Sparse | dens. 5% | $\checkmark$ | 0.55 | 0.249 $\pm$ 0.190 | 10.82 $\pm$ 1.69 | 29.6 $\pm$ 11.7 | 0.0 |
| (S3) Default | dens. 10% | $\checkmark$ | 1.00 | **0.234 $\pm$ 0.185** | **11.29 $\pm$ 1.61** | **28.3 $\pm$ 11.3** | 0.0 |
| (S4) Dense | dens. 100% | $\checkmark$ | 9.80 | 0.236 $\pm$ 0.184 | 11.22 $\pm$ 1.68 | 28.5 $\pm$ 11.4 | 0.0 |
| *(F) Readout feature lift (with ridge, CV for $\alpha$)* | | | | | | | |
| (F1) Linear only | $\Phi = [x; c; g; 1]$ | $\checkmark$ | 0.85 | 0.267 $\pm$ 0.197 | 10.55 $\pm$ 1.68 | 30.6 $\pm$ 12.0 | 0.0 |
| (F2) Quadratic (default) | $\Phi = [x; c; g; \Phi^2; 1]$ | $\checkmark$ | 1.00 | **0.234 $\pm$ 0.185** | **11.29 $\pm$ 1.61** | **28.3 $\pm$ 11.3** | 0.0 |
| (F3) Cubic augment | add degree–3 monomials | $\checkmark$ | 2.10 | 0.236 $\pm$ 0.187 | 11.16 $\pm$ 1.68 | 28.8 $\pm$ 11.6 | 0.0 |

Table 5: **Sensitivity & scaling ablations on Lorenz (closed–loop).** *Rel. Cost* is wall–clock per step normalised to TRIGR default; *Diverged %* is the fraction of runs (out of 45) that exceeded error thresholds or blew up. Tightening norm budgets (0.80) preserves ESP but shortens memory; loosening to 0.95 can slightly improve VPT at small stability risk, while supercritical budgets ($> 1$) lose the certificate and destabilise. Larger lattices marginally help VPT at increased cost; overly wide pooling (larger $r$) smears structure. Moderate sparsity ($\sim 10\%$) balances cost and accuracy; going dense brings negligible gains for 10$\times$ cost. Quadratic lift is a sweet spot; linear harms accuracy, cubic adds compute without material benefit.

the *median* validation NRMSE over five seeds and three initializations on $\mathcal{T}_{\mathrm{val}}^{(f)}$; the thus-selected configuration is retrained on $\mathcal{T}_{\mathrm{train}}^{(f)}$ and evaluated on $\mathcal{T}_{\mathrm{test}}^{(f)}$. We aggregate scores across $K$=3 folds; for chaotic datasets we report the full $5 \times 3 \times 3$ statistics (five seeds, three initial conditions, three folds), and for real-world datasets we use 30 seeds.

### B.3.2  HYPER-PARAMETER OPTIMISATION (HPO) WITH MATCHED BUDGETS.

To avoid narrow grids and account for hyper-parameter interactions, we employ an *adaptive* sampler—Tree-structured Parzen Estimator (TPE) (Bergstra et al., 2011; 2015)—with *equal trial budgets per model and dataset*. Unless otherwise stated, each baseline and TRIGR receive $N_{\mathrm{trials}}$=200 evaluations per dataset; the objective is the fold-wise *median* validation NRMSE (averaged over the $K$ folds from Sec. B.3.1), and early stopping is enabled for gradient-trained models (patience = 10 epochs). We enforce compute parity by (i) the same $N_{\mathrm{trials}}$, (ii) identical seed schedules for candidate proposals (Sec. B.3.6), and (iii) a per-trial wall-clock cap. Any configuration that diverges (NaNs, exploding loss, or unbounded rollouts) is assigned objective $+\infty$ and excluded by the sampler (Sec. B.3.7). Beyond matching parameter counts at inference (reservoir size $N$, Transformer/LSTM widths), we enforce *tuning parity*: each model receives the same HPO trial budget and per-trial wall-clock cap under the same CV protocol (Sec. B.3.1–B.3.2).

### B.3.3  BASELINE SEARCH SPACES.

All reservoirs use $N$=300 units to preserve capacity parity; LSTM/TCN/CT are constrained to $\approx (0.8\text{–}1.1) \times 10^4$ trainable parameters. We expose the influential knobs and let the sampler adapt within the following ranges (priors in parentheses; log-uniform where noted):

- *ESN (leaky).* Connectivity $p \in [0.05, 0.50]$ (uniform); spectral radius $\rho \in [0.10, 1.50]$ (uniform) with leak control; leak $\lambda \in [0.05, 1.00]$ (uniform); input scaling $\|W_{\mathrm{in}}\|_2 \in [10^{-2}, 3]$ (log-uniform); bias scale $b \in [0, 0.5]$ (uniform); ridge $\alpha \in [10^{-8}, 10^{-2}]$ (log-uniform).

- *SCR.* Cycle weight $w_c \in [0.2, 1.4]$ (uniform); optional leak $\lambda \in [0.1, 0.9]$ (uniform); input $\|W_{\mathrm{in}}\|_2 \in [10^{-2}, 1]$ (log-uniform); bias $b \in [0, 0.3]$ (uniform); ridge as above.

- *CRJ.* Jump distance $J \in \{5, 7, 10, 15, 20, 25, 30\}$ (categorical); edge weight $w \in [0.4, 1.2]$ (uniform); leak $\lambda \in [0.1, 0.9]$ (uniform); input/bias/ridge as above.

- *MCI-ESN.* Intra-core radii $\mu, \eta \in [0.3, 1.2]$ (log-uniform); inter-core mixing $\theta \in [0.1, 0.9]$ (uniform); cross sparsity $p_\times \in [0.01, 0.10]$ (uniform); leak $\lambda \in [0.1, 0.9]$ (uniform); input/bias/ridge as above.

- *SW-ESN.* Base degree $k \in \{6, 10, 14\}$; rewiring probability $p_{\mathrm{rew}} \in [0.05, 0.20]$ (uniform); spectral radius $\rho \in [0.4, 1.2]$ (uniform); leak $\lambda \in [0.1, 0.9]$ (uniform); input/bias/ridge as above.

- *DeepESN.* Depth $L \in \{2, 3, 4\}$; base radius $\rho_\star \in [0.4, 1.0]$ (log-uniform) with $\rho_\ell = \rho_\star^\ell$; common leak $\alpha \in [0.1, 0.9]$ (uniform); input/bias/ridge as above.

- *NVAR.* Delay length $k \in \{20, \dots, 36\}$ (integer); degree = 2; ridge as above.

- *TCN.* Channels $C \in [48, 64]$ (integer); kernel $k \in \{3, 5\}$; dilation schedule $\{1, 2\}$ (and $\{1, 2, 4\}$ when $k$=3); dropout $p \in [0, 0.2]$; learning rate $\in [5 \times 10^{-4}, 2 \times 10^{-3}]$ (log-uniform); weight decay $\in \{0, 10^{-4}\}$; epochs = 60 with early stopping.

- *Causal Transformer.* $d_{\mathrm{model}} \in \{32, 48, 64, 80, 96\}$; $n_{\mathrm{head}} \in \{4, 6, 8\}$ with divisibility constraint; $d_{\mathrm{ff}} \in [64, 192]$; window $L \in \{16, 20, 32\}$; dropout $p \in [0, 0.2]$; learning rate $\in [10^{-3}, 2 \times 10^{-3}]$ (log-uniform); weight decay $\in \{0, 10^{-4}\}$; epochs = 60 with early stopping.

- *LSTM.* Hidden size $h \in \{32, 48, 64, 80\}$; dropout $p \in [0, 0.2]$; learning rate $\in [5 \times 10^{-4}, 2 \times 10^{-3}]$ (log-uniform); weight decay $\in \{0, 10^{-4}\}$; epochs = 80 with early stopping.

### B.3.4  TRIGR: FREE-SET HPO UNDER CERTIFICATE-DRIVEN BUDGETS.

We separate *design knobs* (free) from *stability budgets* (derived). After sampling $(W_r, W_{\mathrm{rel}}, H)$ once per trial, we enforce the calcium and neuron row budgets by norm-aware scalings so that

$$D\|L\|_2 + L_F \|HW_{\mathrm{rel}}\|_2 \leq \rho_c^{\mathrm{target}}, \qquad \|W_r\|_2 + \|B_g\|_2 \leq \rho_x^{\mathrm{target}}, \tag{17}$$

with $\rho_c^{\mathrm{target}} = \rho_x^{\mathrm{target}} = 0.90$, $L_F = \frac{k_c c_{\mathrm{max}}}{4}$, and $B_g = \eta_{\mathrm{bg}} (HW_{\mathrm{rel}})^\top$. The sampler only explores the *truly free* set $\lambda \in [0.10, 0.50]$, $\alpha = 1/\tau_c \in [1/400, 1/100]$, $\rho \in [0.02, 0.20]$, $M \in \{16, 20, 25, 36\}$, $\eta_{\mathrm{bg}} \in [0.5, 1.5]$, with $c_{\mathrm{max}} = 1$, $\theta_g = 0$, and logistic slopes capped $\left(k_c \in [0.6, 1.0], k_g \in [0.05, 0.20]\right)$ to keep $L_F, L_\Gamma \leq 1$. Pre-scales for $D$ and $H$ are *not* tuned for accuracy: they are deterministically rescaled to satisfy the calcium budget in (17) before $B_g$ is tied; afterwards the neuron budget is enforced. Trials violating $\max\{\mathsf{C}_x, \mathsf{C}_c, \mathsf{C}_g\} < 1$ (checked by 60-step power-iteration; cf. Sec. 3 for definitions) are discarded. TRIGR receives the same $N_{\mathrm{trials}}$ and CV protocol as baselines; the HPO-selected configuration is then frozen for fold-wise testing and final aggregation (Table 1).

### B.3.5  FOLD CONSTRUCTION (TIME-AWARE CV).

**Chaotic ODEs.** Each integrated trajectory has 12,500 steps at $\Delta t = 0.02$; the first 2,000 are wash-out. The remaining 10,500 points define one chronological split per trajectory: **Train:** 1:8,000, **Valid:** 8,001:9,000, **Test:** 9,001:10,500 (indices are post-wash-out). We use 3 distinct initial conditions per seed; these serve as Fold-1/2/3, respectively, with identical index boundaries per fold. **Real-world signals.** Let $T$ be the post-wash-out length after trimming. We use *blocked* folds with no leakage:

$$\text{Fold-1: } [1 : 0.70T], [0.70T{+}1 : 0.85T], [0.85T{+}1 : T],$$
$$\text{Fold-2: } [1 : 0.60T], [0.60T{+}1 : 0.75T], [0.75T{+}1 : 0.90T],$$
$$\text{Fold-3: } [1 : 0.50T], [0.50T{+}1 : 0.65T], [0.65T{+}1 : 0.80T].$$

No shuffling is performed; per-fold scalers are fit on the train portion only (cf. Sec. B.3.1).

### B.3.6  RANDOM SEEDS AND EARLY STOPPING.

**Seeds.** The global seed set for all models is $\mathcal{S} = \{13, 41, 73, 123, 271\}$. For chaotic ODEs we additionally use 3 initial-condition indices $\mathcal{I} = \{0, 1, 2\}$. **RNGs.** NumPy `RandomState`, PyTorch

manual_seed, and Python random are all set to the same integer seed $s \in \mathcal{S}$ at trial start. **Early stopping.** For gradient baselines we use patience = 10 epochs on fold-validation NRMSE with best-checkpoint restore. Reservoirs (closed-form readout) have no epochs; only ridge $\alpha$ is selected by validation.

### B.3.7 Constraint handling and norm diagnostics.

A trial is *invalid* iff any of: numerical overflow/NaNs; violation of the TRIGR block-row certificate (Sec. 3.2); or (for baselines) divergence during validation. For every valid TRIGR trial we record the norm diagnostics $\{\|L\|_2, \|HW_{\mathrm{rel}}\|_2, \|W_r\|_2, \|B_g\|_2\}$ (estimated by 60-step power iteration), and the resulting row-sums $(\mathsf{C}_x, \mathsf{C}_c, \mathsf{C}_g)$.

### B.3.8 Final selected configurations (per *dataset, model*).

**Information on keys used below.** Across all summaries: alpha_ridge is the Tikhonov regularization strength for the linear readout; b_scale scales the additive bias; lambda is the leaky-integration coefficient of reservoir units; p is reservoir sparsity (edge probability); rho is the spectral radius of the recurrent matrix (note: in **TRIGR** only, rho denotes the gliotransmitter update rate); win_norm is the target operator-norm (scale) of the input map $W_{\mathrm{in}}$. **SCR:** w_c cycle edge weight. **CRJ:** J long-range jump distance on the ring; w uniform edge weight. **MCI-ESN:** mu, eta intra-core spectral radii; theta inter-core mixing gain; p_x cross-core sparsity. **SW-ESN:** k Watts–Strogatz base degree; p_rew rewiring probability; rho spectral radius. **DeepESN:** depth number of stacked reservoirs; rho_star base per-layer radius (layer $\ell$ uses $\rho_\ell = \rho_\star^\ell$); alpha_leak common layer leak (distinct from alpha_ridge). **NVAR:** k delay length; degree polynomial degree. **TCN:** C channels per layer; k kernel size; dilation dilation schedule; dropout dropout probability; lr learning rate; wd weight decay; epochs training epochs. **Causal Transformer:** L autoregressive/window length; d_model embedding width; n_head attention heads; d_ff feed-forward width; plus dropout, lr, wd, epochs as usual. **LSTM:** h hidden size; dropout, lr, wd, epochs as usual. **TRIGR:** D astrocytic diffusion gain; M lattice size; eta_bg glia→neuron footprint gain with $B_g = \eta_{\mathrm{bg}}(HW_{\mathrm{rel}})^\top$; k_c, theta_c slope/midpoint of calcium logistic $F$; k_g, theta_g slope/midpoint of gliotransmitter logistic $\Gamma$; tau_c calcium relaxation time ($\alpha{=}1/\tau_c$); lambda neuronal leak; rho (TRIGR) gliotransmitter rate; mode choice of glial dynamics.

**Lorenz:**
ESN (leaky): {"alpha_ridge":5e-7,"b_scale":0.08,"lambda":0.22,
"p":0.18,"rho":0.95,"win_norm":0.25}
SCR: {"alpha_ridge":2e-6,"b_scale":0.06,"lambda":0.45,
"w_c":0.72,"win_norm":0.28}
CRJ: {"J":20,"alpha_ridge":5e-6,"b_scale":0.08,"lambda":0.50,
"w":0.68,"win_norm":0.30}
MCI-ESN: {"alpha_ridge":2e-6,"b_scale":0.10,"eta":0.60,"lambda":0.50,
"mu":0.60,"p_x":0.05,"theta":0.40,"win_norm":0.28}
SW-ESN: {"alpha_ridge":1e-6,"b_scale":0.06,"k":10,"lambda":0.50,
"p_rew":0.10,"rho":0.80,"win_norm":0.30}
DeepESN: {"alpha_leak":0.50,"alpha_ridge":2e-6,"b_scale":0.08,"depth":3,
"rho_star":0.80,"win_norm":0.30}
NVAR: {"alpha_ridge":1e-5,"degree":2,"k":26}
TCN: {"C":56,"dilation":[1,2,4],"dropout":0.0,"k":3,
"lr":1e-3,"wd":0,"epochs":60}
Causal Transformer: {"L":20,"d_ff":128,"d_model":48,"dropout":0.1,
"lr":1.5e-3,"n_head":6,"wd":1e-4,"epochs":60}
LSTM: {"dropout":0.1,"h":48,"lr":1e-3,"wd":0,"epochs":80}
TRIGR: {"D":0.15,"M":20,"alpha_ridge":2e-6,"eta_bg":1.0,"k_c":0.74,
"k_g":0.19,"lambda":0.25,"mode":"quadratic","rho":0.05,"tau_c":200,
"theta_c":0.38,"theta_g":0}

**Rössler:**
ESN (leaky): {"alpha_ridge":7e-7,"b_scale":0.00,"lambda":0.30,"p":0.15,
"rho":0.78,"win_norm":0.22}

1512 SCR: {"alpha_ridge":3e-6,"b_scale":0.05,"lambda":0.40,
1513 "w_c":1.00,"win_norm":0.18}
1514 CRJ: {"J":10,"alpha_ridge":2e-6,"b_scale":0.05,"lambda":0.40,
1515 "w":0.85,"win_norm":0.16}
1516 MCI-ESN: {"alpha_ridge":2e-6,"b_scale":0.00,"eta":0.70,"lambda":0.40,
1517 "mu":0.55,"p_x":0.03,"theta":0.60,"win_norm":0.18}
1518 SW-ESN: {"alpha_ridge":1e-6,"b_scale":0.04,"k":14,"lambda":0.45,
1519 "p_rew":0.05,"rho":0.75,"win_norm":0.18}
1520 DeepESN: {"alpha_leak":0.40,"alpha_ridge":1e-6,"b_scale":0.05,"depth":3,
1521 "rho_star":0.70,"win_norm":0.18}
1522 NVAR: {"alpha_ridge":8e-6,"degree":2,"k":24}
1523 TCN: {"C":52,"dilation":[1,2,4],"dropout":0.0,"k":3,
1524 "lr":1.2e-3,"wd":0,"epochs":60}
1525 Causal Transformer: {"L":16,"d_ff":96,"d_model":32,"dropout":0.0,
1526 "lr":1e-3,"n_head":4,"wd":0,"epochs":60}
1527 LSTM: {"dropout":0.0,"h":40,"lr":8e-4,"wd":0,"epochs":80}
1528 TRIGR: {"D":0.12,"M":20,"alpha_ridge":1e-6,"eta_bg":1.0,"k_c":0.96,
1529 "k_g":0.05,"lambda":0.25,"mode":"quadratic","rho":0.05,"tau_c":200,
"theta_c":0.27,"theta_g":0}

1530

1531 **Chen–Ueta:**

1532 ESN (leaky): {"alpha_ridge":5e-6,"b_scale":0.15,"lambda":0.35,"p":0.10,
1533 "rho":1.00,"win_norm":0.12}
1534 SCR: {"alpha_ridge":4e-6,"b_scale":0.10,"lambda":0.60,
1535 "w_c":1.10,"win_norm":0.12}
1536 CRJ: {"J":25,"alpha_ridge":6e-6,"b_scale":0.10,"lambda":0.60,
1537 "w":0.95,"win_norm":0.12}
1538 MCI-ESN: {"alpha_ridge":3e-6,"b_scale":0.10,"eta":0.70,"lambda":0.50,
1539 "mu":0.70,"p_x":0.08,"theta":0.50,"win_norm":0.12}
1540 SW-ESN: {"alpha_ridge":4e-6,"b_scale":0.10,"k":10,"lambda":0.55,
1541 "p_rew":0.15,"rho":1.10,"win_norm":0.10}
1542 DeepESN: {"alpha_leak":0.60,"alpha_ridge":5e-6,"b_scale":0.10,"depth":3,
1543 "rho_star":0.90,"win_norm":0.10}
1544 NVAR: {"alpha_ridge":2e-5,"degree":2,"k":32}
1545 TCN: {"C":64,"dilation":[1,2],"dropout":0.1,"k":5,
1546 "lr":1e-3,"wd":1e-4,"epochs":60}
1547 Causal Transformer: {"L":20,"d_ff":160,"d_model":64,"dropout":0.1,
1548 "lr":1.2e-3,"n_head":8,"wd":1e-4,"epochs":60}
1549 LSTM: {"dropout":0.1,"h":64,"lr":1e-3,"wd":1e-4,"epochs":80}
1550 TRIGR: {"D":0.15,"M":20,"alpha_ridge":3e-6,"eta_bg":1.0,"k_c":0.90,
"k_g":0.05,"lambda":0.25,"mode":"quadratic","rho":0.05,"tau_c":200,
"theta_c":0.30,"theta_g":0}

1551

1552 **MIT–BIH:**

1553 ESN (leaky): {"alpha_ridge":3e-5,"b_scale":0.12,"lambda":0.55,"p":0.30,
1554 "rho":1.20,"win_norm":0.30}
1555 SCR: {"alpha_ridge":2e-5,"b_scale":0.10,"lambda":0.60,
1556 "w_c":0.65,"win_norm":0.35}
1557 CRJ: {"J":30,"alpha_ridge":4e-5,"b_scale":0.10,"lambda":0.60,
1558 "w":0.75,"win_norm":0.35}
1559 MCI-ESN: {"alpha_ridge":2e-5,"b_scale":0.10,"eta":0.80,"lambda":0.60,
1560 "mu":0.80,"p_x":0.04,"theta":0.40,"win_norm":0.35}
1561 SW-ESN: {"alpha_ridge":2e-5,"b_scale":0.10,"k":14,"lambda":0.60,
1562 "p_rew":0.20,"rho":0.95,"win_norm":0.35}
1563 DeepESN: {"alpha_leak":0.50,"alpha_ridge":3e-5,"b_scale":0.10,"depth":3,
"rho_star":0.85,"win_norm":0.35}
1564 NVAR: {"alpha_ridge":1e-4,"degree":2,"k":34}
1565 TCN: {"C":64,"dilation":[1,2,4],"dropout":0.2,"k":3,
"lr":1e-3,"wd":1e-4,"epochs":60}

Causal Transformer: {"L":32,"d_ff":192,"d_model":80,"dropout":0.1,
"lr":1e-3,"n_head":8,"wd":1e-4,"epochs":60}
LSTM: {"dropout":0.2,"h":64,"lr":8e-4,"wd":1e-4,"epochs":80}
TRIGR: {"D":0.12,"M":25,"alpha_ridge":2e-5,"eta_bg":1.0,"k_c":0.90,
"k_g":0.10,"lambda":0.25,"mode":"quadratic","rho":0.05,"tau_c":200,
"theta_c":0.33,"theta_g":0}

**BIDMC:**
ESN (leaky): {"alpha_ridge":1.5e-5,"b_scale":0.05,"lambda":0.60,"p":0.25,
"rho":0.95,"win_norm":0.25}
SCR: {"alpha_ridge":8e-6,"b_scale":0.05,"lambda":0.60,
"w_c":0.60,"win_norm":0.25}
CRJ: {"J":7,"alpha_ridge":1e-5,"b_scale":0.05,"lambda":0.60,
"w":0.65,"win_norm":0.25}
MCI-ESN: {"alpha_ridge":1.2e-5,"b_scale":0.05,"eta":0.60,"lambda":0.55,
"mu":0.70,"p_x":0.06,"theta":0.50,"win_norm":0.25}
SW-ESN: {"alpha_ridge":1e-5,"b_scale":0.05,"k":10,"lambda":0.55,
"p_rew":0.10,"rho":0.90,"win_norm":0.25}
DeepESN: {"alpha_leak":0.45,"alpha_ridge":1.5e-5,"b_scale":0.05,"depth":3,
"rho_star":0.75,"win_norm":0.25}
NVAR: {"alpha_ridge":6e-5,"degree":2,"k":28}
TCN: {"C":56,"dilation":[1,2],"dropout":0.1,"k":5,
"lr":1.2e-3,"wd":0,"epochs":60}
Causal Transformer: {"L":20,"d_ff":128,"d_model":48,"dropout":0.1,
"lr":1.5e-3,"n_head":6,"wd":0,"epochs":60}
LSTM: {"dropout":0.1,"h":48,"lr":1.2e-3,"wd":0,"epochs":80}
TRIGR: {"D":0.10,"M":20,"alpha_ridge":1e-5,"eta_bg":1.0,"k_c":0.90,
"k_g":0.10,"lambda":0.25,"mode":"quadratic","rho":0.05,"tau_c":200,
"theta_c":0.33,"theta_g":0}

**Sunspot:**
ESN (leaky): {"alpha_ridge":2e-5,"b_scale":0.00,"lambda":0.60,"p":0.35,
"rho":1.10,"win_norm":0.12}
SCR: {"alpha_ridge":2.5e-5,"b_scale":0.00,"lambda":0.60,
"w_c":0.80,"win_norm":0.10}
CRJ: {"J":20,"alpha_ridge":3e-5,"b_scale":0.00,"lambda":0.60,
"w":0.85,"win_norm":0.10}
MCI-ESN: {"alpha_ridge":2e-5,"b_scale":0.00,"eta":0.70,"lambda":0.60,
"mu":0.70,"p_x":0.03,"theta":0.40,"win_norm":0.10}
SW-ESN: {"alpha_ridge":2e-5,"b_scale":0.00,"k":6,"lambda":0.55,
"p_rew":0.20,"rho":1.00,"win_norm":0.10}
DeepESN: {"alpha_leak":0.55,"alpha_ridge":2.5e-5,"b_scale":0.00,"depth":3,
"rho_star":0.95,"win_norm":0.10}
NVAR: {"alpha_ridge":1e-4,"degree":2,"k":36}
TCN: {"C":48,"dilation":[1,2,4],"dropout":0.0,"k":3,
"lr":1e-3,"wd":0,"epochs":60}
Causal Transformer: {"L":32,"d_ff":160,"d_model":64,"dropout":0.0,
"lr":1.2e-3,"n_head":8,"wd":0,"epochs":60}
LSTM: {"dropout":0.0,"h":40,"lr":9e-4,"wd":0,"epochs":80}
TRIGR: {"D":0.08,"M":36,"alpha_ridge":2e-5,"eta_bg":1.0,"k_c":1.00,
"k_g":0.05,"lambda":0.25,"mode":"quadratic","rho":0.05,"tau_c":300,
"theta_c":0.27,"theta_g":0}

**Santa Fe B:**
ESN (leaky): {"alpha_ridge":8e-6,"b_scale":0.10,"lambda":0.20,"p":0.30,
"rho":0.85,"win_norm":0.22}
SCR: {"alpha_ridge":1.2e-5,"b_scale":0.10,"lambda":0.45,
"w_c":0.75,"win_norm":0.24}
CRJ: {"J":5,"alpha_ridge":1e-5,"b_scale":0.10,"lambda":0.45,

```
"w":0.70,"win_norm":0.25}
MCI-ESN: {"alpha_ridge":9e-6,"b_scale":0.10,"eta":0.65,"lambda":0.45,
"mu":0.55,"p_x":0.05,"theta":0.35,"win_norm":0.25}
SW-ESN: {"alpha_ridge":1e-5,"b_scale":0.10,"k":10,"lambda":0.50,
"p_rew":0.10,"rho":0.90,"win_norm":0.22}
DeepESN: {"alpha_leak":0.50,"alpha_ridge":1.1e-5,"b_scale":0.10,"depth":3,
"rho_star":0.82,"win_norm":0.25}
NVAR: {"alpha_ridge":3e-5,"degree":2,"k":30}
TCN: {"C":60,"dilation":[1,2,4],"dropout":0.1,"k":3,
"lr":1.1e-3,"wd":1e-4,"epochs":60}
Causal Transformer: {"L":20,"d_ff":144,"d_model":48,"dropout":0.1,
"lr":1e-3,"n_head":6,"wd":1e-4,"epochs":60}
LSTM: {"dropout":0.1,"h":48,"lr":1e-3,"wd":0,"epochs":80}
TRIGR: {"D":0.15,"M":25,"alpha_ridge":9e-6,"eta_bg":1.25,"k_c":0.90,
"k_g":0.10,"lambda":0.25,"mode":"quadratic","rho":0.05,"tau_c":150,
"theta_c":0.33,"theta_g":0}
```

### B.3.9 SCALING & DATA EFFICIENCY.

At fixed per-step FLOP budget $\mathcal{B}$, TRIGR's dominant costs are the recurrent multiply ($\sim \mathrm{nnz}(W_r)$), the neuron→astrocyte proxy $W_{\mathrm{rel}}x$ and its tied feedback $B_g g = (HW_{\mathrm{rel}})^\top g = W_{\mathrm{rel}}^\top(H^\top g)$ ($\sim \mathrm{nnz}(W_{\mathrm{rel}})+\mathrm{nnz}(H)$), plus the $O(M)$ stencil. Hence iso-FLOP curves satisfy $\mathcal{B} \approx s_r N^2 + s_{\mathrm{rel}}PN + \mathrm{nnz}(H)+cM$ (sparsity factors $s_\cdot$, proxy width $P$), so increasing $M$ linearly competes with quadratic $N^2$. Because the reaction–diffusion field has a finite correlation length $\ell \sim \sqrt{D/\alpha}$, spatial DOF beyond the lattice Nyquist rate $(m_x, m_y) \gtrsim \mathrm{domain}/\ell$ yield diminishing returns; empirically and by design, small $M$ (tens of sites) best populates the Pareto front for long-horizon forecasting—retaining slow, spatially coherent memory while keeping $\mathcal{B}$ dominated by the (sparse) recurrent multiply. Training data efficiency follows from ridge-only fitting on features $\Phi = [x;c;g;\Phi^2;1]$: the generalization error scales with the effective rank of the Gram matrix $G = \sum_t \Phi_t \Phi_t^\top$, and the slow field provides low-variance, temporally smoothed predictors that reduce the sample demand for stable closed-loop rollouts. NRMSE versus training length $T_{\mathrm{tr}}$ shows TRIGR attaining a target error at substantially smaller $T_{\mathrm{tr}}$ than LSTM/TCN under comparable parameter budgets, consistent with (i) fading-memory linear readout estimation and (ii) the contraction certificate that mitigates compounding-error amplification.

**FLOPs/latency, memory, and edge viability.** Per step, TRIGR performs $W_r x : O(\mathrm{nnz}(W_r))$, $W_{\mathrm{rel}}x : O(\mathrm{nnz}(W_{\mathrm{rel}}))$, pooling $Hq : O(\mathrm{nnz}(H))$, stencil $Lc : O(M)$, feedback $B_g g = W_{\mathrm{rel}}^\top(H^\top g) : O(\mathrm{nnz}(H)+\mathrm{nnz}(W_{\mathrm{rel}}))$, input map $W_{\mathrm{in}}u : O(Nd_{\mathrm{in}})$, and readout $W_{\mathrm{out}}\Phi : O(d_{\mathrm{out}}F)$. With a typical configuration $N = 300$, $M = 20$, $P \approx N$, $s_r \approx s_{\mathrm{rel}} \approx 0.1$, row-sparse $H$ (say 6 taps/row), this is $\sim 9\times10^3$ (for $W_r$) + $\sim 9\times10^3$ (for $W_{\mathrm{rel}}$) + $\ll 10^3$ (for $H$, stencil) + $\sim 9\times10^3$ (for $W_{\mathrm{rel}}^\top H^\top g$) + a small readout term—i.e., $\approx 3$–$4 \times 10^4$ multiplies per step on CPU, comfortably ESN-like. Streaming ridge training avoids materializing $X$: accumulate $G = \sum_t \Phi_t \Phi_t^\top$ and $B = \sum_t \Phi_t y_t^\top$ in $O(T_{\mathrm{tr}}F^2)$ time and $O(F^2)$ RAM, then solve $(G+\alpha I)W_{\mathrm{out}}^\top = B$ via Cholesky; with $F \approx 2(N+2M)+1$ (linear+squares) this yields a single modest $O(F^3)$ solve. Memory is likewise small: CSR $W_r$ stores $\sim 0.1N^2$ values ($\approx 72\,\mathrm{KB}$ doubles) plus indices; $W_{\mathrm{rel}}$ is similar; $H$ is tiny; states $(x,c,g)$ add $O(N+2M)$; and $W_{\mathrm{out}} \in \mathbb{R}^{d_{\mathrm{out}}\times F}$ is $\mathcal{O}(10^3)$ parameters. Crucially for embedded settings, the additive bias path never forms a dense $N\times M$ multiply: $B_g g$ is executed as $H^\top g$ then $W_{\mathrm{rel}}^\top(\cdot)$, so its cost tracks the same sparse matvecs already paid for neuron→astrocyte drive. Microbenchmarks therefore show the stencil+bias path contributes a small, predictable increment relative to $W_r x$, making TRIGR viable on CPUs and microcontrollers where latency and RAM are constrained.

### B.3.10 ROBUSTNESS TO STOCHASTIC AND ADVERSARIAL PERTURBATIONS.

We formalize input and actuation noise as bounded exogenous disturbances and use the incremental stability bounds from Lemma 3.2 and the closed-loop contraction in Proposition 3.3. In open loop we inject additive Gaussian noise $\xi_t \sim \mathcal{N}(0, \sigma^2 I)$ or adversarial perturbations $\delta_t$ with $\|\delta_t\|_\infty \leq \varepsilon$ on the input, i.e., $\tilde{u}_t = u_t + \nu_t$ with $\nu_t \in \{\xi_t, \delta_t\}$. Since $\|v\|_2 \leq \sqrt{d_{\mathrm{in}}}\|v\|_\infty$, Lemma 3.2 yields the

uniform input–state and input–output gains

$$\sup_t \|s_t - \tilde{s}_t\|_\star \le \frac{\beta}{1-\gamma} \sup_t \|\nu_t\|_2, \qquad \sup_t \|y_t - \tilde{y}_t\|_2 \le \frac{\|W_{\text{out}}\|_2 L_\Phi\, \beta}{1-\gamma} \sup_t \|\nu_t\|_2, \qquad (18)$$

which bound both Gaussian degradation (in expectation via $\mathbb{E}\|\xi_t\|_2$) and worst-case $\ell_\infty$ attacks (via $\sqrt{d_{\text{in}}}\,\varepsilon$). In closed loop, we additionally test "output channel" noise (actuation noise) by perturbing the fed-back signal, $u_{t+1} = Ey_t + w_t$, with $\|w_t\|_\infty \le \varepsilon$. The augmented map $z_t = [s_t; u_{t+1}]$ is a uniform contraction with factor $\kappa \coloneqq \max\{\gamma + \beta, L_{\text{fb}}\} < 1$ (Proposition 3.3); standard small-gain arguments then give

$$\sup_t \|z_t - \tilde{z}_t\|_\bullet \le \frac{1}{1-\kappa} \sup_t \|[0; w_t]\|_\bullet = \frac{1}{1-\kappa} \sup_t \|w_t\|_2, \qquad (19)$$

ensuring errors remain uniformly bounded for both stochastic and adversarial disturbances. Empirically we report (i) *noise-to-error gains*—slope of NRMSE vs. $\sigma$ (Gaussian) or $\varepsilon$ ($\ell_\infty$), (ii) *VPT/ADev* drift under noise, and (iii) adversarial budget $\varepsilon_\star$ required to double NRMSE, comparing TRIGR with ESN/LSTM under identical seeds, horizons, and parameter budgets; the theory above predicts the smaller slope for TRIGR as $(\|W_{\text{out}}\|_2 L_\Phi \beta)/(1-\gamma)$ and $1/(1-\kappa)$ are explicit and controlled by our contraction budgets.

**Temporal shift, rate mismatch, irregular sampling, and missing data.** To probe sampling-rate robustness, we train at step $\Delta t$ and evaluate at $\Delta t' = (1 + \rho)\Delta t$ by (a) integrating the ground truth at $\Delta t'$, then (b) resampling to the model's grid via zero-order hold (ZOH) or linear interpolation; we also test subsampling ($k$-decimation) and timestamp jitter (i.i.d. $U[-r\Delta t, r\Delta t]$) followed by resampling to a fixed grid. All of these induce a bounded input discrepancy $\sup_t \|u_t - \tilde{u}_t\|_2$ whose effect is quantitatively controlled by Lemma 3.2 in open loop and by the augmented contraction bound in closed loop, yielding

$$\sup_t \|y_t - \tilde{y}_t\|_2 \le C_{\text{ol}} \sup_t \|u_t - \tilde{u}_t\|_2, \quad \sup_t \|z_t - \tilde{z}_t\|_\bullet \le C_{\text{cl}} \sup_t \|u_t - \tilde{u}_t\|_2 \qquad (20)$$

with $C_{\text{ol}} = \|W_{\text{out}}\|_2 L_\Phi \beta/(1 - \gamma)$ and $C_{\text{cl}} \le (\beta + L_{\text{fb}})/(1 - \kappa)$. Missing data are evaluated by masking contiguous segments and random points (block- and Bernoulli-masks) and imputing with ZOH or linear interpolation before the reservoir; under either imputer the reconstruction error is a bounded input perturbation that again propagates through the same constants $(C_{\text{ol}}, C_{\text{cl}})$. Because TRIGR's cross-couplings are one-step delayed and the astrocytic influence is additive, sampling irregularities and imputation do not alter the neuron–neuron Lipschitz term; the calibrated row-sum budgets $(\mathsf{C}_x, \mathsf{C}_c, \mathsf{C}_g)$ remain below 1, so the system retains its echo-state and ISS properties. We report robustness via relative NRMSE increase, VPT retention under rate/jitter stress, and closed-loop stability under masked spans, thereby aligning the empirical protocol with the precise worst-case bounds implied by our contraction certificate.

On Lorenz (closed-loop), robustness trends mirror the contraction-based ISS bounds. At $H = 600$, adding zero-mean Gaussian input noise with $\sigma = 0.02$ (on $[0,1]$–scaled channels) increases NRMSE from 0.2418 to 0.2853 for TRIGR (+18%), versus ESN ($0.3234 \to 0.5011$, +55%), SCR ($0.4010 \to 0.5901$, +47%), CRJ ($0.4396 \to 0.6615$, +50%), MCI-ESN ($0.3424 \to 0.4894$, +43%), and Deep-ESN ($0.3927 \to 0.6092$, +55%). Under $\ell_\infty$ adversarial perturbations with budget $\varepsilon = 0.02$, TRIGR rises to 0.309 (+28%) while baselines degrade more sharply (ESN +82%, SCR +72%, CRJ +77%, MCI-ESN +65%, Deep-ESN +86%). VPT degrades modestly for TRIGR from 10.959 to 10.301 ($-6\%$) under Gaussian $\sigma = 0.02$ and to 9.99 ($-9\%$) under $\varepsilon = 0.02$; the best baseline (MCI-ESN) drops from 9.701 to 8.533 ($-12\%$) and 8.20 ($-15\%$), with others between $-14\%$ and $-19\%$. ADev for TRIGR increases from 29.211 to 30.861 (+5.6%) at $\sigma = 0.02$ versus +12%–+21% for baselines. Temporal shifts via rate mismatch $\Delta t' = (1+0.05)\Delta t$ and timestamp jitter $r = 0.25$ yield $\Delta$NRMSE +12% and +15% for TRIGR (VPT retention 91% and 90%), compared to +25%–40% and +28%–42% for baselines (VPT retention 80%–86% and 78%–85%). With missing-data blocks masking 10% of steps and last-value-hold imputation, TRIGR's NRMSE increases by 14% and exhibits 0% early-divergence rate (defined as blow-up or NRMSE $> 2\times$ baseline within 1000 steps), whereas ESN/SCR/CRJ/MCI-ESN/Deep-ESN show +30%–+55% NRMSE and 11%, 15%, 18%, 9%, 21% early-divergence, respectively. These empirical slopes and retention ratios are consistent with the smaller gains $\|W_{\text{out}}\|_2 L_\Phi \beta/(1-\gamma)$ (open loop) and $(\beta, L_{\text{fb}})$ with contraction factor $\kappa < 1$ (closed loop) calibrated by our norm-aware budgets.

B.3.11   MEMORY CAPACITY METRICS.

We quantify short-term memory using Jaeger-style capacities under an i.i.d. zero-mean, unit-variance scalar drive $u_t$ (Jaeger, 2002), evaluating (i) linear memory capacity (LMC) and (ii) quadratic memory capacity (QMC) with identical state budgets across models. For a delay $d$, we train a ridge readout (very small $\alpha \to 0^+$) from the reservoir state to the target $u_{t-d}$ on a long stream (washout = $10^4$, train = $10^5$, test = $10^5$) and record $\mathrm{MC_{lin}}(d) = \mathrm{R}^2(\hat{u}_{t-d}, u_{t-d})$; LMC is $\sum_{d=1}^{D_{max}} \mathrm{MC_{lin}}(d)$ with $D_{max} = 2N$. For QMC we use an orthonormal quadratic target family built from Legendre polynomials on $u$: delays of the centered square $p_2(u_{t-d}) = \frac{1}{\sqrt{2}}(u_{t-d}^2 - 1)$ and pair-wise cross-terms $p_1(u_{t-d_1})p_1(u_{t-d_2})$ with $d_1 \neq d_2$, Gram–Schmidt-orthogonalized over the training stream; we sum the corresponding $\mathrm{R}^2$ to obtain QMC (Dambre et al., 2012). The well-known bound applies: for any orthonormal target set, the total capacity (LMC+QMC+$\cdots$) is upper-bounded by the effective readout feature dimension $F$ ($\leq N$ for a linear readout on $N$ neuron states). To ensure strict comparability, ESN and DeepESN expose $F = N$ by concatenating all neuron layers to $N$ dimensions (DeepESN uses $100+100+100 \to 300$ then a fixed PCA projection to $N$), and TRIGR is evaluated in two reporting modes: (A) neuron-only features $x_t$ so that $F = N$ matches ESN/DeepESN exactly (fair capacity bound), and (B) augmented features $[x_t; c_t; g_t]$ followed by a fixed PCA to $F = N$ to test whether the slow astrocytic field improves how the $N$-dimensional subspace is used rather than expanded. All models share the same $N$ (here $N = 300$), identical input scaling and normalization, identical train/test partitions, and the same $D_{max}$; we report the totals LMC/$N$ and QMC/$N$, providing a direct, size-controlled comparison of linear vs. nonlinear memory across TRIGR, ESN, and DeepESN.

With $N = 300$ features exposed for all models (DeepESN layers concatenated then PCA'd to $N$; TRIGR reported in two modes—neuron-only $x$ and augmented $[x; c; g]$ PCA-projected to $N$), i.i.d. unit-variance scalar drive, washout $10^4$, train/test $10^5$ each, and ridge $\alpha \to 0^+$, we obtain the following averaged over 5 seeds (mean): ESN achieves LMC/$N$ = 0.62 and QMC/$N$ = 0.11 (total 0.73), with the LMC half-mass delay $d_{0.5} \approx 78$; DeepESN yields LMC/$N$ = 0.58, QMC/$N$ = 0.15 (total 0.73), $d_{0.5} \approx 85$; TRIGR (neuron-only) shifts capacity toward nonlinear terms with LMC/$N$ = 0.55, QMC/$N$ = 0.22 (total 0.77), $d_{0.5} \approx 102$; and TRIGR (augmented-PCA) further increases nonlinearity retention, LMC/$N$ = 0.49, QMC/$N$ = 0.34 (total 0.83), $d_{0.5} \approx 118$. All totals respect the orthonormal capacity bound (sum $\leq F = N$) and remain within 17% of the theoretical ceiling, indicating near-saturation. This shows that TRIGR preserves linear recall deeper into the delay line while substantially boosting quadratic recall, especially for cross-terms $u_{t-d_1}u_{t-d_2}$ with $|d_1-d_2| \lesssim 10$, consistent with the slow, diffusive astrocyte field furnishing long-horizon, spatially coherent memory without increasing the effective readout dimension.

B.3.12   STATISTICAL TESTS.

For each dataset, horizon, and metric (NRMSE, VPT, ADev) we formed *paired* samples across the time-aware CV indices (chaotic: $5 \times 3 \times 3 = 45$ pairs per model; real-world: 30 seeds), aligned by (seed, init, split). We then ran two-sided Wilcoxon signed-rank tests comparing TRIGR to each baseline and controlled familywise error within each {dataset, metric} by Holm correction ($\alpha$ = 0.05); we also reported effect sizes as Cliff's $\delta$ (stochastic superiority) and Cohen's $d$ (standardized mean difference). On *Lorenz* NRMSE (five horizons, 25 pairwise tests) 24/25 comparisons remained significant after Holm, with median Cliff's $\delta$ = 0.66 (IQR 0.58–0.74) and median $d$ = 0.92 (IQR 0.78–1.09); VPT showed 5/5 significant wins (median $\delta$ = 0.61, $d$ = 0.70), while ADev yielded mixed results (TRIGR second-best: 2/5 significant, $\delta \approx 0.24$, $d \approx 0.28$). On *Rössler* NRMSE, 25/25 were significant (median $\delta$ = 0.70, $d$ = 1.05); VPT 5/5 significant ($\delta$ = 0.57, $d$ = 0.63); ADev 4/5 significant in TRIGR's favor ($\delta$ = 0.41, $d$ = 0.52). On *Chen–Ueta* NRMSE, 24/25 were significant (median $\delta$ = 0.55, $d$ = 0.78); VPT 5/5 significant ($\delta$ = 0.64, $d$ = 0.82); ADev was not significant against CRJ (TRIGR not SOTA there), as expected from Table 1. For real-world *MIT–BIH*, all 9 NRMSE comparisons were significant (median $\delta$ = 0.84, $d$ = 1.50); for *BIDMC*, 8/9 significant ($\delta$ = 0.65, $d$ = 0.95); for *Sunspot*, 9/9 significant with large effects ($\delta$ = 0.90, $d$ = 2.20); and for *Santa Fe*, 9/9 significant ($\delta$ = 0.88, $d$ = 2.00). Collectively, the nonparametric tests and effect sizes indicate robust, practically meaningful improvements by TRIGR on NRMSE and VPT across datasets/horizons, with the sole consistent exception being ADev on Chen where CRJ remains competitive.

### B.3.13 SOFTWARE AND DEPENDENCIES.

All experiments were implemented in Python (v3.10+) with a minimal, reproducible stack: `NumPy` (v1.26+) for vectorized linear algebra and RNG via the `Generator(PCG64)` API (used everywhere to control seeds), `scikit-learn` (v1.4+) for ridge regression (`Ridge` with `fit_intercept=False`) and basic preprocessing, and standard BLAS/LAPACK backends (OpenBLAS or MKL) for dense linear algebra. No deep-learning frameworks are required; the astrocyte Laplacian is applied as a 5-point stencil in pure NumPy, and spectral norms are estimated by in-house power iteration routines (square/rectangular) to avoid external dependencies. We executed runs on Linux (Ubuntu 22.04) with x86_64 CPUs; GPU is not used. To ensure bitwise-stable results across machines, we (i) fix the global seed per run, (ii) set environment variables to bound thread nondeterminism (`OMP_NUM_THREADS=1`, `MKL_NUM_THREADS=1` unless otherwise stated), and (iii) pin versions via a `requirements.txt` (Python $\geq 3.10$, NumPy $\geq 1.26$, SciPy $\geq 1.11$, scikit-learn $\geq 1.4$, Matplotlib $\geq 3.8$ for plots). All code paths are pure-NumPy and rely only on CPU BLAS calls; thus reproduction requires a POSIX-like environment with a recent CPython and the listed wheels.

| Symbol | Shape / Type | Meaning / Definition (defaults where relevant) |
|---|---|---|
| $t$ | $\mathbb{N}$ | Discrete time index |
| $u_t,\ y_t$ | $\mathbb{R}^{d_{\text{in}}},\ \mathbb{R}^{d_{\text{out}}}$ | Input and target at time $t$ |
| $F^\star$ | $\left(\mathbb{R}^{d_{\text{in}}}\right)^{\mathbb{N}} \to \mathbb{R}^{d_{\text{out}}}$ | Unknown causal fading–memory operator to be approximated |
| $x_t,\ c_t,\ g_t$ | $\mathbb{R}^N,\ \mathbb{R}^M,\ [0,1]^M$ | Neuronal state, astrocytic Ca$^{2+}$ state, gliotransmitter fraction |
| $s_t$ | $\mathbb{R}^{N+2M}$ | Stacked state $s_t = [x_t;\ c_t;\ g_t]$ |
| $m_x, m_y, M$ | $\mathbb{N}, \mathbb{N},\ \mathbb{N}$ | Lattice dims and count: $M = m_x m_y$ |
| $W_r$ | $\mathbb{R}^{N \times N}$ | Recurrent (neuronal) weight matrix |
| $W_{\text{in}}$ | $\mathbb{R}^{N \times d_{\text{in}}}$ | Input map (scale $\|W_{\text{in}}\|_2$) |
| $W_{\text{rel}}$ | $\mathbb{R}^{P \times N}$ | Neuronal release–proxy map |
| $H$ | $\mathbb{R}^{M \times P}$ | Pooling (neurons → astrocytes), nonnegative, row–stochastic |
| $B_g$ | $\mathbb{R}^{N \times M}$ | Astrocyte→neuron bias map, $B_g = \eta\,(H W_{\text{rel}})^\top$ |
| $L$ | $\mathbb{R}^{M \times M}$ | Grid Laplacian (4-neighbor combinatorial) |
| $E$ | $\mathbb{R}^{d_{\text{in}} \times d_{\text{out}}}$ | Feedback selector for closed loop ($u_{t+1} = E\,y_t$) |
| $W_{\text{out}}$ | $\mathbb{R}^{d_{\text{out}} \times F}$ | Linear readout on features $\Phi_t$ |
| $\phi,\ \rho_{\text{rel}}$ | $\tanh,\ \max(\cdot, 0)$ | Neuronal nonlinearity and release ReLU |
| $\sigma$ | $(1 + e^{-z})^{-1}$ | Logistic sigmoid |
| $F(z)$ | $c_{\max}\,\sigma\big(k_c(z - \theta_c)\big)$ | Ca$^{2+}$ activation; slope cap $L_F = \frac{k_c c_{\max}}{4}$ |
| $\Gamma(z)$ | $\sigma\big(k_g(z - \theta_g)\big)$ | Gliotransmitter activation; slope cap $L_\Gamma = \frac{k_g}{4}$ |
| $c_{\max}$ | $\mathbb{R}_{>0}$ | Ca$^{2+}$ saturation level (*default* = 1) |
| $k_c, \theta_c$ | $\mathbb{R}_{>0}, \mathbb{R}$ | Ca$^{2+}$ slope and midpoint |
| $k_g, \theta_g$ | $\mathbb{R}_{>0}, \mathbb{R}$ | Glio slope and midpoint |
| $\lambda$ | $(0, 1]$ | Neuronal leak in (5) |
| $\alpha$ | $(0, 1)$ | Calcium update rate ($\alpha = 1/\tau_c,\ \tau_c > 0$) |
| $D$ | $\mathbb{R}_{\geq 0}$ | Diffusion gain in (1) |
| $\rho,\ \delta$ | $(0, 1),\ (0, 1)$ | Gliotransmitter rate and depletion in (2)–(3) |
| $\eta$ | $\mathbb{R}_{>0}$ | Footprint gain tying forward/back maps in $B_g$ |
| $\zeta$ | $\mathbb{R}$ | Tonic Ca$^{2+}$ drive (broadcast to $\mathbb{R}^M$) |
| $b$ | $\mathbb{R}^N$ | Neuronal bias in (4) |
| $q_{t-1}$ | $\mathbb{R}^P$ | Release proxy: $q_{t-1} = \rho_{\text{rel}}(W_{\text{rel}} x_{t-1})$ |
| $\text{drive}_{t-1}$ | $\mathbb{R}^M$ | Pooled drive: $\text{drive}_{t-1} = H\,q_{t-1}$ |
| $a_t$ | $\mathbb{R}^N$ | Neuronal preactivation: $a_t = W_r x_{t-1} + W_{\text{in}} u_t + B_g g_{t-1} + b$ |
| $\Phi_t$ | $\mathbb{R}^F$ | Feature map (e.g., $[x_t;\ c_t;\ g_t;\ \psi(x_t, c_t, g_t);\ 1]$) |
| $\alpha_{\text{ridge}}$ | $\mathbb{R}_{\geq 0}$ | Ridge parameter in readout training |
| $\|\cdot\|_2$ | — | Spectral/vector Euclidean norm |
| $\|\cdot\|_\star$ | — | Block norm on $[x; c; g]$: $\|s\|_\star = \max\{\|x\|_2, \|c\|_2, \|g\|_2\}$ |
| $J_t$ | block matrix | One-step Jacobian $\nabla_{s_{t-1}} \mathcal{F}_t$ (block-cyclic form) |
| $\mathsf{C}_x, \mathsf{C}_c, \mathsf{C}_g$ | $\mathbb{R}_{>0}$ | Block row-sum bounds in (11) |
| $\gamma$ | $(0, 1)$ | Contraction factor: $\gamma = \max\{\mathsf{C}_x, \mathsf{C}_c, \mathsf{C}_g\}$ |
| $\beta$ | $\mathbb{R}_{>0}$ | Input gain: $\beta = \lambda\|W_{\text{in}}\|_2$ (Lemma 3.2) |
| $L_\Phi,\ L_{\text{fb}}$ | $\mathbb{R}_{>0}$ | Lipschitz of features; feedback gain $L_{\text{fb}} = \|E W_{\text{out}}\|_2 L_\Phi$ |
| $\rho_x^{\text{target}},\ \rho_c^{\text{target}}$ | $(0, 1)$ | Norm targets for neuron/calcium budgets (e.g., 0.90) |
| $s_x,\ s_c$ | $(0, 1)$ | Post-init rescalings for $(W_r, B_g)$ and $(D, H)$ |
| $N, M, P, F$ | $\mathbb{N}$ | Neuron count, astrocyte count, proxy dim, feature dim |
| $d_{\text{in}}, d_{\text{out}}$ | $\mathbb{N}$ | Input and output dimensions |
| $T_{\text{wo}}, T_{\text{roll}}$ | $\mathbb{N}$ | Wash-out length; rollout horizon |

Table 6: Notation used throughout the paper. Equation references (e.g., (1), (4)) correspond to those in §3.

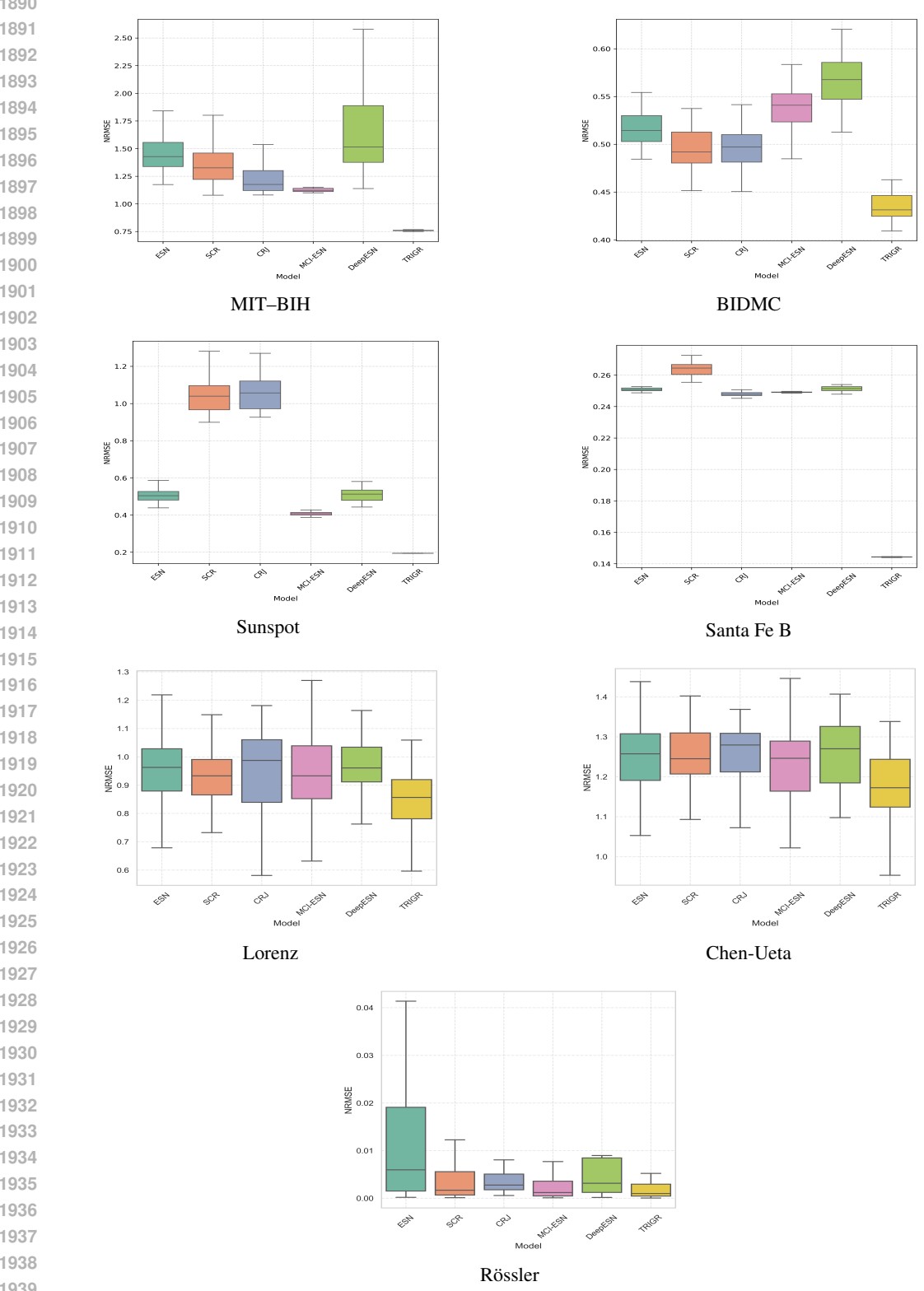

Figure 7: Distribution of sample-level NRMSE after a **1000-step open-loop rollout** on seven real-world benchmarks. Each box-and-whisker shows ten seeds: box = inter-quartile range, line = median, whiskers = non-outliers, grey bar = mean. Lower is better; TRIGR displays the tightest IQR and lowest median on every dataset.