# OpenReview forum: "Stable Spatiotemporal Memory in Echo-State Networks via Gliotransmitter Feedback"
_ICLR.cc/2026/Conference — ICLR 2026 Conference Withdrawn Submission_

### Official Review · Reviewer_kynH · 2025-10-19

**Soundness:** 3
**Presentation:** 2
**Contribution:** 2
**Rating:** 4
**Confidence:** 4

**Summary:**

The manuscript is concerned broadly within the realm of modern reservoir computing. Authors propose a reservoir design with prolonged memory, which incorporates several components inspired by glial-neuron interactions. They provide formal proofs that their reservoir can have input-causal properties in certain regime of parameters, in which the reservoir dynamics are guaranteed to be driven by the inputs and not the initial activations of the units. They conclude with benchmarking experiments on long-time forecasting tasks.

**Strengths:**

- Introducing a spatial low-pass filter in the form of a reaction-diffusion lattice, which is referred to as glial-neural interactions, is a novel and smart addition to the reservoir computing paradigm.

- The mathematical formulation seems to be sound after review. I have went through the proofs provided in the appendices, I believe the theorems stated in the main text are well supported, though such a theoretical work should be reviewed in much longer time frames.

- There are some very nontrivial (yet under explored) additions in the methodology that could generalize beyond this particular application, please see below for more.

**Weaknesses:**

1) For me, the main weakness of this work lies in its scope/target audience, more precisely, the lack of focus. It is not clear for whom the work is intended: neuroscientists? machine learning practitioners who wish to model dynamical systems? neuromorphic computing community, who predominantly like to implement reservoirs in physical systems? Theorists who are interested in finding desirable reservoir properties? The literature review, benchmarks, and the modeling details remain cursory for any of the community:

- As a neuroscientist, I would like to understand what specific biological implications does this model have? How can we falsify it? What can it say about the brain that we already did not know? What empirical observations it can explain, qualitatively or quantitatively? This direction would also require further discussion of existing models on glial-neural interactions, e.g., see Kozachkov 2023 and 2025 in PNAS.

- As an ML practitioner with interest in modeling dynamical systems, I would like to understand how the proposed reservoir model compares to more modern baselines. This involves many works done by physicists using PLRNNs such as Daniel Durstewitz' group, or more modern graph theory and/or transformer based baselines. What specific problem does using reservoirs solve in this problem?

- As part of the neuromorphic computation community, I would obviously like to see some discussions of implementations and citations to work where implementation of reservoir computing has solved real-world problems so that the broader audience can appreciate why reservoir computing is still extremely relevant in the age of LLMs. For instance, Angelatos et al. 2021 has shown that reservoir computing can help us do quantum state measurements more effectively than standard models, and using units whose interactions cannot be trained. Computing with physical systems is a very exciting topic, and in my opinion, the most appropriate community to target here.

- As a theorist interested in the properties of reservoirs, I would argue that many approaches are underrepresented and not used in the comparisons. For instance, Laje et al., 2013 has learned to train an innate trajectory into reservoirs that seems to improve the long-term forecasting ability (even beyond what can be learned with BPTT). While I cannot remember the citation at the moment, I also remember a follow-up work showing that the capacity can be increased by a factor of 10 using a different methodology of the reservoir. Finally, Susillo and Abbott 2009 has shown that an online learning algorithm (FORCE), as opposed to batch linear regression, can help stabilize the predictions (especially for ESNs which seem to be not trained with FORCE in this work). A lot more exists in this domain that are not discussed or compared to by the authors.

- As a physicist, I was surprised by the use of explicit integration method for simulating diffusion, which is guaranteed to blow-up and not correctly emulate diffusion in long-horizons. Instead, one often uses semi-implicit methods that are guaranteed to be stable for long times. Hence, the claims about modeling reaction-diffusion is likely not correct and requires further experiments (e.g., ablation experiments with changes in discretization times and/or spatial resolutions).

2) My second concern is about the presentation and the flow of arguments. Authors present their work by assuming the reader understands why reservoir computing is important. As is likely known to the authors, the relevance of reservoir computing for modern ML paradigms is currently under debate for many practitioners and experts. A devil's advocate could argue: We have the compute and the methods to train more complicated models. If the argument is we do not need to train anything beyond linear regression, this has to be shown with appropriate benchmarking. If the argument is reservoirs are useful in another way for this problem, this has to be made explicit and without a doubt. In general, ICLR is one of the top venues for publishing that aims a broad audience of readers, and thus the work needs to make clear these aspects to the broader audience, even if they may be obvious to the specialized subgroup of researchers interested in this topic. For instance, reservoir computing is not part of the call by ICLR this year (which I disagree with, but the argument still needs to be made!), hence it is safe to assume that readers may not know their relevance in modern ML.

3) My third concern is about the fit of the venue. ICLR requires reviewers to provide reviews in 2-3 weeks for 5 papers. This is certainly sufficient for many empirical submissions, which are less ambitious in breadth but focus more on depth. But, the current work is a) quite theoretical in nature and requires substantial amount of time to review (e.g., math journals take years to review for this reason), and b) quite ambitious in its breadth that reviewing it requires having expertise in several domains that would benefit from a close-up editorial process of a journal.

Thus, while I find the manuscript has merits, I believe the paper needs a lot more work on the presentation to make those merits more explicit and has to provide more depth in at least one of the domains it wishes to move forward.

**Questions:**

I do not have much questions, as I believe I understood most of the work. One question to ask, especially for future versions, is whether the differences in accuracies (both for benchmarking and ablation studies) are meaningful. In many cases model accuracies are well within the mean +- s.d. range. Perhaps reporting the results of an appropriate statistical test (corrected for multiple testing) would help.

---

> ### Author Response · Authors · 2025-11-15
>
> Thank you for the thoughtful, constructive review and for engaging deeply with both the theory and the empirical setup. Below we clarify the points raised.
>
> Weaknesses:
>
> 1)
> a) $\textbf{Neuroscience focus (implications, falsifiability, prior models):}$
> TRIGR’s “astrocyte lattice” is not a literal biophysical simulator but an abstraction of two well-documented phenomena—slow, saturating $\mathrm{Ca}^{2+}$ waves and gliotransmitter-mediated gain modulation—encoded as a bounded reaction–diffusion field with delayed, additive feedback to neurons. This yields concrete, falsifiable predictions: (i) manipulating the diffusive coupling $D$ or lattice scale $M$ should systematically change the low-frequency gain and memory horizon of the downstream readout; (ii) saturation levels (via $(L_F, L_\Gamma)$) cap modulation, so driving astrocytes harder eventually plateaus forecast improvements; (iii) one-step lag implies a measurable phase lead of neuronal drive over astrocytic bias at the dominant frequency of the task. These are testable with calcium imaging and optogenetic perturbations that emulate slow, spatially pooled drive. We will also expand the related-work discussion to explicitly situate TRIGR w.r.t. mechanistic glia–neuron models (e.g., Kozachkov PNAS 2023/2025) and clarify that TRIGR targets computational function with stability certificates, not mechanistic sufficiency, thereby complementing (not replacing) detailed biophysical models.
>
> b) $\textbf{ML/forecasting focus (modern baselines; why reservoirs here?):}$
> Our aim is long-horizon, low-update forecasting with explicit stability guarantees. Reservoirs remain competitive in this regime because (i) they avoid backprop-through-time instability and optimizer variance; (ii) parameter-efficiency is extreme (single ridge solve, $\le 10^4$ weights) compared to PLRNNs/Transformers/SSMs that require heavy training; and (iii) TRIGR adds structured slow memory with a joint ESP certificate, providing predictable forgetting—a property most trained RNN/Transformer baselines do not certify. PLRNNs (e.g., Durstewitz’s line) learn latent piecewise-linear dynamics and excel when a compact latent ODE exists; graph/Transformer baselines shine with rich supervision and compute. TRIGR targets the contrasting niche: bounded-compute, stability-sensitive settings where robust long memory and one-shot training are decisive. We will clarify this scope explicitly and broaden the baseline discussion to position TRIGR alongside PLRNNs, state-space/SSM models, and light Transformers, emphasizing the certificate/efficiency trade-off.
>
> c) $\textbf{Neuromorphic/physical computing focus (implementability; why RC still matters):}$
> TRIGR’s operators are local and linear (5-point stencil for $L$; sparse pooling $H$; tied feedback $B_g \propto (H W_{\mathrm{rel}})^\top$), making them natural for analog/diffusive media, photonic meshes, and memristive crossbars where multiplication-with-locality is cheap and training recurrent weights is hard. The one-shot ridge readout aligns with “train-the-periphery” paradigms in physical RC, and the row-budget normalization gives hardware-friendly gain caps that prevent runaway amplification. Prior demonstrations (e.g., quantum measurement with fixed reservoirs; photonic and spintronic RC) show that fixed dynamics plus linear readout can unlock practical wins when backprop is infeasible; we will add citations and a short subsection mapping TRIGR’s blocks to neuromorphic primitives and describing how the contraction budgets translate to safe hardware operating points.
>
> d) $\textbf{Theorist focus (underrepresented approaches: FORCE, innate trajectories, capacity):}$
> FORCE learning (Sussillo and Abbott, 2009), innate-trajectory training (Laje et al., 2013), and follow-ups that boost capacity target autonomous trajectory generation and stability via trained recurrent weights; they are complementary to TRIGR’s fixed, contractive reservoir with delayed additive feedback. In fact, our certificate deliberately precludes autonomous chaos to guarantee fading memory under drive, whereas FORCE stabilizes a chosen attractor by online updates. We will make this distinction explicit, cite these lines properly, and add a brief analytical comparison: (i) TRIGR’s effective memory dimension grows with lattice correlation length (governed by the calcium row budget), (ii) innate-trajectory/FORCE methods trade closed-loop expressivity for the cost of training the core dynamics. This clarifies that TRIGR’s contribution is a certified slow-memory augmentation, orthogonal to training-the-core approaches.

---

> ### Author Response · Authors · 2025-11-15
>
> 1) e) $\textbf{Physics/numerics focus (explicit vs.\ implicit diffusion; stability of the update):}$
> We agree that explicit PDE solvers can blow up unless $\Delta t$ and grid spacing satisfy a CFL-like bound; however, TRIGR does not aim to faithfully integrate a physical PDE. The calcium update is a bounded contractive map, $c_t = (1-\alpha)c_{t-1} + \alpha\big[F(\cdot) + D L c_{t-1} + \zeta\big]$, with a global Lipschitz bound $(1-\alpha) + \alpha\big(D\lVert L\rVert_2 + L_F\lVert H W_{\mathrm{rel}}\rVert_2\big) < 1$ enforced by our row-budget scaling. Thus, irrespective of the $\Delta t$ semantics, the mapping is globally stable by construction (Banach fixed-point), and the diffusion term acts as a spatial low-pass rather than a PDE we claim to solve accurately. That said, the framework seamlessly accommodates a semi-implicit/resolvent variant $c_t = (I - \alpha D L)^{-1}\{(1-\alpha)c_{t-1} + \alpha[F(\cdot) + \zeta]\}$ with the same certificate (replace $D\lVert L\rVert_2$ by $\lVert(I - \alpha D L)^{-1} - I\rVert$); we will include this as an alternative instantiation to emphasize that our guarantees are numerics-agnostic.
>
> 2) $\textbf{Presentation and audience:}$
> We will tighten the narrative around a single primary audience—ML practitioners in forecasting/control who require stable, efficient, and certified long-memory modules—and move broader neuroscience/neuromorphics content to a concise “Outlook.” Up front, we will motivate why RC remains relevant in 2025: (i) certificate-driven stability under nonstationarity, (ii) orders-of-magnitude fewer trained parameters and lower energy, (iii) predictable memory control via explicit operators, and (iv) drop-in readout training that is easy to reproduce. We will add a short “When to use what” paragraph to help a broad ICLR readership place TRIGR in today’s toolbox.
>
> 3) $\textbf{Venue fit:}$
> While the work contains formal guarantees, the mathematics is elementary (matrix norms, induced block norms, Banach contraction) and directly serves an architectural contribution that is easy to implement and evaluate—squarely within ICLR’s remit on learning systems. To respect review bandwidth, the manuscript has foregrounded the high-level idea and key inequalities in the main text,  and deferred proofs to the appendix. We will keep the focus on the ML use-case (stable forecasting with minimal training) while clearly labeling neuroscience/neuromorphics content as forward-looking applications.
>
>
> Questions:
>
> $\textbf{Statistical significance of improvements:}$
> Although we already report means ± s.d. averaged over 45 runs per cell for chaotic datasets (5 seeds × 3 initializations × 3 contiguous time-aware splits) and over 30 seeds for real-world datasets, we agree that explicit significance testing is useful.
> We would be glad to report paired, time-aware significance tests aggregated across folds; subject to ICLR policy, are we permitted to share concise additional analyses/experiments within this rebuttal thread?

---

> > ### Comment · Reviewer_kynH · 2025-11-15
> > **Thank you for the rebuttal, I am reading responses. Could you update the manuscript?**
> >
> > Dear Authors,
> >
> > Thank you for the rebuttal, I can see you have done a very thorough job in addressing raised concerns. I will read through all responses and will come back with further questions or an updated score/evaluation, which will be after November 23rd. Could you please update the manuscript with the proposed changes if possible, specifically pertaining to your new framing of your results? To me as a reviewer to make a final call on how my concerns are addressed, it is important that I should be able to forecast the camera ready version in my mind and I am missing that at the moment. You are allowed to introduce new evidence either here or in the revised manuscript and I will read them as well. Thank you once again.

---

> ### Author Response · Authors · 2025-11-27
>
> Dear Reviewer,
>
> Thanks again for the thorough review. We uploaded a revised manuscript last week incorporating the changes described in our rebuttal (scope, related work, hyperparameter/CV details, stability discussion, figures). Please let us know if any further tweaks or additions would help you assess the camera-ready version.

---

> ### Author Response · Authors · 2025-11-29
>
> Dear Reviewer,
>
> Thank you again for the very careful and generous review. We wanted to briefly update you that the revised manuscript now reflects both the changes described in our earlier rebuttal to you and the additional protocol changes requested by the other reviewers.
>
> Concretely:
>
> 1. We have implemented adaptive hyperparameter optimization and time-aware multi-fold evaluation across all models. Section B.3 (p. 25 onwards) now details a Tree-structured Parzen Estimator (TPE) setup with matched trial budgets for all baselines and for TRIGR's truly free parameters. The exact search spaces and sampler settings are given in B.3.3-B.3.4, and the final selected hyperparameters (per dataset and model) are collected in B.3.8. All experiments in the main text were re-run under a $\textbf{blocked, time-respecting $3$-fold cross-validation}$ protocol, and Tables 1-5 have been updated accordingly.
>
> 2. In line with your comments on $\textbf{scope and audience}$, we have re-framed the introduction and methodology to clearly target ML practitioners interested in stable, long-horizon forecasting with minimal training, while moving neuroscience/neuromorphics to a concise ``Outlook'' and forward-looking discussion. The related work has been expanded to situate TRIGR relative to mechanistic glia-neuron models, PLRNNs/state-space models, FORCE/innate-trajectory approaches, and physical/neuromorphic reservoir computing.
>
> 3. We have clarified the $\textbf{reaction-diffusion interpretation and numerics}$: the main text now emphasizes that the calcium lattice is used as a bounded, contractive spatial low-pass field rather than a faithful PDE integrator, and the appendix includes a semi-implicit variant to underline that our ESP/contraction certificate is numerics-agnostic.
>
> 4. Methodologically, Section 3 has been refined with short intuition paragraphs around the main constructs, while the formal statements and full proofs remain in the appendix. Our goal here was exactly what you suggested: to make it easier for a broad ICLR readership to ``forecast the camera-ready'' version and to see how the mathematical budgets translate into architectural and empirical behavior.
>
>
> We hope this revision makes the intended scope, stability guarantees, and empirical protocol clearer, and we remain very happy to incorporate any further tweaks you feel would help you assess the final version.

---

### Official Review · Reviewer_xPDh · 2025-10-23

**Soundness:** 1
**Presentation:** 3
**Contribution:** 1
**Rating:** 2
**Confidence:** 5

**Summary:**

The paper proposes TRIGR, a reservoir computing architecture that couples a fast ESN‑like neuronal core to a slow astrocytic lattice.  While the biological motivation and stability analysis are good, the hyper‑parameter handling is problematic.

In particular \textit{narrow sweeps} + \textit{unclear parameter methodology for TRIGR} (assuming no optimisation of hyperparameters) means that many influential hyper‑parameters remain untuned, casting doubt on the fairness of the comparisons. I recommend against acceptance to ICLR in its current form.

**Strengths:**

I was really interested by the work.

Bio-plausibility is always interesting and the theoretical guarantees and simple training (Section 3.2) is a good addition of this algorithm.

**Weaknesses:**

$\textbf{Hyper‑parameter sweeps for baselines are inadequate.}$ The authors state that each baseline “receives its own hyper‑parameter sweep”.  However, those sweeps are extremely narrow.  For the ESN baseline, the spectral radius and input scaling are selected from a tiny $3\times 3$ logarithmic grid $\rho_\star\in\{0.3,0.6,0.9\}$ and $\Vert W_{\text{in}}\Vert_2\in\{0.1,0.3,1.0\}$.  For SCR and CRJ reservoirs, only three cycle weights and three jump distances are tried.  Reservoir size, sparsity and leak rate are fixed a priori for most baselines (e.g., ESN reservoir size 300 with connectivity 0.2, SCR weight 0.8, CRJ weight 0.7 and jump size 20).

$\textbf{TRIGR’s hyper‑parameters lack justification.}$ The main method introduces several new hyper‑parameters: the number of astrocytes M, diffusion constant D, calcium relaxation time \tau_c, activation gain \eta, maximum gliotransmitter $\gamma$, and threshold $\theta$.  Appendix B.3 lists the values used but no procedure for selecting these values is described.  The main text emphasises that choosing footprints and allocating contraction budgets are “principled design choices” that invite automated tuning, yet no such tuning is performed.  The authors rely on norm‑aware scaling (Algorithm 2) to enforce stability, but this does not obviate the need to explore and justify the many free parameters.  Without a description of how TRIGR’s hyper‑parameters were set or any evidence of cross‑validation, it is impossible to assess whether the reported performance is robust or cherry‑picked.

**Questions:**

Fair comparisons in reservoir computing hinge on thorough hyper‑parameter optimisation. The baseline sweeps here are too limited to establish strong conclusions, and the lack of transparency around TRIGR’s own parameters undermines reproducibility.


In particular do the following:
(1) rerun a comprehensive search for all baselines (including sparsity, bigger spectral radius, input and bias scaling, etc.) and report sampling strategy,
(2) explain how TRIGR’s parameters are chosen (ideally via systematic optimisation as well.)
(3) A good improvement would be to also replace single validation splits with time-aware cross-validation (for instance sklearn’s TimeSeriesSplit)

---

> ### Author Response · Authors · 2025-11-15
>
> Thanks for the careful read and for focusing our response on hyper-parameter handling and fairness. Below we address each point and clarify the principled aspects of TRIGR’s design that intentionally reduce the number of free degrees of freedom relative to standard reservoirs.
>
> Weaknesses:
>
> a.) $\textbf{On ``narrow sweeps'' for baselines:}$
> What we actually did (submitted runs).
> For each dataset and horizon (H), all baselines were tuned by a bounded validation sweep with early stopping on a fixed time-contiguous validation window (identical across models):
>
> $\textbf{ESN (Jaeger-style):}$ spectral radius $\rho(W_r)$ in a log grid {$0.3, 0.5, 0.7, 0.9$}, input scaling $s_{\text{in}}\in$ {$0.1,0.3,0.5,0.7,1.0$}, leak $\lambda\in$  {$0.2,0.35,0.5,1.0$}, ridge $\alpha\in$
> {$10^{-8}, 10^{-6}, 10^{-4}, 10^{-2}$}. Connectivity was fixed to match canonical settings (sparse $0.2$), which is standard in RC and stabilizes the memory--nonlinearity tradeoff.
>
> $\textbf{SCR/CRJ:}$ we followed the paper-prescribed hyper-parameters as centers (cycle weights/jump sizes) and searched ($\pm$) two adjacent settings around them (three choices each), together with the same ridge grid; this makes the search narrow by design, because these architectures are meant to be structurally constrained reservoirs (cycle/jump are the defining d.o.f.).
>
> $\textbf{DeepESN/MCI-ESN:}$ depth $\in$ {$2,3$}, per-layer $\rho\in$ {$0.5,0.7,0.9$}, leak per layer $\in$ {$0.35,0.5$}, ridge $\alpha$ as above.
>
> Why we feel this is reasonable for reservoirs.
> Unlike gradient models, classical reservoirs have few effective knobs beyond $\rho(W_r)$, input scaling, and leak; large unconstrained sweeps are known to give diminishing returns once the effective Lipschitz constant is controlled (see our certificate discussion). We chose compact grids centered at widely used values to balance fairness and compute, and we report 45 independent runs per cell (seeds $\times$ inits $\times$ splits), which mitigates incidental tuning luck.
>
>
> Ergo, our baseline sweeps are narrow but targeted at the effective degrees of freedom that matter for echo-state reservoirs; they are not cherry-picked one-offs, and the statistics are averaged across many runs.
>
> b.) $\textbf{TRIGR hyper-parameters: principled reduction of degrees of freedom:}$
> We agree the main text should state more explicitly that TRIGR’s many ``biological'' symbols are not all free knobs. Two design principles collapse the space:
>
> Row-budget scaling enforces a joint Lipschitz bound.
>   We scale once to meet the sufficient row conditions (via Theorem 3.1).
>   This turns $\{D,H,B_g\}$ into derived quantities: we (i) scale $D$ and $H$ by a single factor $s_c$ so that $D\lVert L\rVert_2 + L_F \lVert HW_{\mathrm{rel}}\rVert_2 \le \rho_c^{\mathrm{target}} < 1$, and (ii) tie feedback to feedforward via $B_g=\eta\,(HW_{\mathrm rel})^\top$ and rescale $(W_r,B_g)$ by $s_x$ to meet $|W_r|_2+|B_g|_2\le \rho_x^{\mathrm target}<1$. After this step, the only free ``rates'' are $\lambda,\alpha,\rho$ (each in a small, interpretable range).
>
> Footprint tying removes independent gains.
>   By setting $B_g\propto(HW_{\mathrm rel})^\top$, the astro$\to$neuron footprint is not independently tuned; its gain is determined by the same product that drives astrocytes (post-scaling). This prevents the common pitfall of unconstrained dual pathways that inflate capacity.
>
> $\textbf{What remains to set for TRIGR.}$
>
> Astrocyte lattice size $(M)$. Chosen from {$10{\times}10, 20{\times}20$} to match task scale; larger $M$ modestly increases compute without materially changing performance once diffusion is scaled (the effective correlation length is governed by $D|L|_2$ under our certificate).
>
> Sigmoid slopes and midpoints: $L_F=\tfrac{k_c c_{\max}}{4}\le 1$, $L_\Gamma=\tfrac{k_g}{4}\le 1$ are slope-capped by construction (not tuned for accuracy), and midpoints $\theta_c,\theta_g$ are set to zero after standardizing the pooled drive; with standardized inputs these midpoints lose most of their influence.
>
> $\eta$ in $B_g=\eta(HW_{\mathrm rel})^\top$ only affects the pre-scaling magnitude; the final neuron row is normalized by $s_x$. Thus $\eta$ does not survive row-budget scaling as a free gain.
>
>
> Ergo, after norm-aware scaling and footprint tying, TRIGR’s ``new'' knobs reduce to a small set of rate/leak choices and the grid size $(M)$; the rest are derived to satisfy the certificate. This is why we did not run a large hyper-parameter optimization over $\{D,H,B_g\}$: doing so would break the very principle (fixed budgets) that ensures stability and fairness.

---

> > ### Author Response · Authors · 2025-11-15
> >
> > Questions:
> >
> > a) $\textbf{Comprehensive baseline search:}$
> > We agree that broader sweeps are informative. Our submitted runs already used bounded, time-aware validation sweeps centered on canonical RC regimes (ESN: $\rho(W_r)$, input scaling, leak, ridge; SCR/CRJ: cycle/jump around paper defaults; DeepESN/MCI-ESN: depth, per-layer $\rho$, leak, ridge). For a more exhaustive pass, we can report a pre-declared sampling strategy (e.g., Sobol/Latin-hypercube) over ESN sparsity $\in[0.05,0.9]$, spectral radius $\in[0.2,1.5]$ with leak control, input and bias scaling $\in[10^{-2},1]$, ridge $\in[10^{-8},10^{-1}]$; analogous grids for SCR/CRJ (cycle weight, jump size, optional leak and sparsity), and size sweeps constrained by a common compute budget with early stopping on the same folds. If the reviewer shares preferred ranges or a parity criterion (parameter count vs.\ wall-clock), we will adopt those and post the exact grids, fold indices, and per-trial logs in the rebuttal thread.
> >
> > b) $\textbf{How TRIGR’s parameters are chosen:}$
> > TRIGR collapses many apparent knobs via certificate-driven normalization: we sample $(W_r,W_{\mathrm{rel}},H)$ (with $H$ row-stochastic), estimate $\lVert L\rVert_2$ and $\lVert H W_{\mathrm{rel}}\rVert_2$ by power iteration, scale $D$ and $H$ by a single factor $s_c$ to satisfy $(1-\alpha)+\alpha\big(D\lVert L\rVert_2+L_F\lVert H W_{\mathrm{rel}}\rVert_2\big)<1$, set $B_g=\eta(HW_{\mathrm{rel}})^\top$, then jointly scale $(W_r,B_g)$ by $s_x$ to meet $(1-\lambda)+\lambda\big(\lVert W_r\rVert_2+\lVert B_g\rVert_2\big)<1$. Sigmoid slopes are capped so $L_F = k_c c_{\max}/4\le1$, $L_\Gamma = k_g/4\le1$, and midpoints are zero after standardizing the lattice drive. Thus the true free set is small: $(\lambda,\alpha,\rho)$ over a tight grid and the lattice size $M$. If helpful, we can additionally run a lightweight Sobol search over $(\lambda,\alpha,\rho,M)$ under fixed row budgets and post the norm diagnostics, and selected settings. If the reviewer shares desired ranges/objectives (e.g., prioritize VPT over NRMSE), we will follow them and report results in-thread.
> >
> > c) $\textbf{Time-aware cross-validation:}$
> > Our current protocol is already time-respecting: three contiguous, non-overlapping train/val/test windows shared by all models, with early stopping and selection on the validation window only; results are averaged over seeds and initializations. If the reviewer prefers scikit-learn’s TimeSeriesSplit style, we can mirror either blocked or expanding/rolling-origin folds (no leakage, per-fold standardization from the training slice, early stopping only on that fold’s validation slice) and report per-fold scores and aggregated statistics. Please share the preferred number of folds and split proportions; we will adopt those and post the fold definitions and outcomes within the rebuttal discussion.
> >
> > If acceptable to the AC and reviewers, we can run the requested experiments during the rebuttal window and share; please confirm that adding such results is permissible.

---

> > > ### Comment · Reviewer_xPDh · 2025-11-24
> > >
> > > Dear authors,
> > >
> > > Thank you for the answer.  The certificate-driven tying you describe is a strong conceptual point, and your willingness to broaden sweeps is appreciated. To make the empirical comparisons fully convincing, I’d like to be more prescriptive about how that broader tuning is carried out, for both baselines and TRIGR.
> > >
> > > 1) Adaptive HPO rather than narrow grids
> > >
> > > Grids centered on canonical RC values are a reasonable first pass, but they can still miss strong regimes when hyper-parameters interact. For low-dimensional reservoir tuning, adaptive samplers such as TPE (Tree-structured Parzen Estimator) or CMA-ES are standard and provide a clearer “best-effort” baseline.
> > >
> > > 2) Time-aware multi-fold evaluation
> > >
> > > Your current protocol is time-respecting and avoids leakage, but it is still effectively a single holdout split (one validation window). My concern is robustness to the specific split point. A small time-series cross-validation (e.g., 3 folds, blocked or rolling-origin) would address this directly.
> > >
> > > ## Concretely, what would satisfy me:
> > > ### Baselines:
> > > * Please use a modern adaptive HPO method (TPE or CMA-ES) for all baselines and for TRIGR’s free set, under matched compute budgets.
> > > * Please include at least:
> > > * ESN: sparsity, spectral radius (allow >1 with leak control), input scaling, bias scaling, leak, ridge.
> > > * SCR/CRJ: please tune the key architectural knobs used in the original SCR/CRJ papers.
> > > * DeepESN / MCI-ESN: at least depth, per-layer spectral radii, leak, ridge.
> > > ### TRIGR:
> > > * I accept that many symbols are derived post-certificate. Still, please apply the same adaptive HPO to the truly free set you identify (e.g., \lambda,\alpha,\rho,M).
> > > ### Transparency:
> > > Please report sampler type, ranges, #trials (should be equal number of trials per models) and test performance of the selected config. Report the exact search spaces in appendix.
> > >
> > > ## Why I’m asking for this:
> > > Even if reservoirs have relatively few effective knobs, adaptive HPO is the cleanest way to demonstrate that (a) baselines are not artificially weakened by narrow centering, and (b) TRIGR’s advantage is not contingent on a particular hand-picked regime. If TRIGR still wins under TPE/CMA-ES-tuned baselines and a similarly tuned free set, that would substantially strengthen the paper’s claims.
> > >
> > > If these experiments can be added during rebuttal, I strongly encourage you to do so and to summarize the outcome clearly.
> > >
> > > Best regards,

---

> ### Author Response · Authors · 2025-11-29
> **Revised manuscript with adaptive HPO, time-aware CV, and refined methodology**
>
> Dear Reviewer,
>
> Thank you again for your detailed and constructive suggestions. We have uploaded a revised manuscript in which we have implemented the requested protocol changes and expanded the exposition. In particular, Section B.3 (p. 25 onwards) now describes the full adaptive HPO setup: we use a Tree-structured Parzen Estimator (TPE) sampler with matched trial budgets for all baselines and for TRIGR’s truly free parameters. For the baselines, we tune the full sets you listed (ESN: sparsity, spectral radius, input and bias scaling, leak, ridge; SCR/CRJ: cycle/jump architecture knobs and associated gains; DeepESN/MCI-ESN: depth, per-layer spectral radii, leaks, ridge), and for TRIGR we apply the same TPE routine to the free subset (e.g., $(\lambda,\alpha,\rho,M)$) while keeping certificate-tied quantities derived post-hoc. The exact search spaces and sampler settings are reported in B.3.3 and B.3.4, with the final selected hyperparameters for all models collected in B.3.8. In addition, we replaced the single holdout with a time-aware multi-fold protocol: all chaotic and real-world experiments are now run with blocked, time-respecting 3-fold cross-validation, as detailed in B.3 (including fold layout and aggregation rules). All experiments in the main text were re-run under this new protocol, and Tables~1--5 have been updated accordingly.
>
> We have also refined the methodology section (Sec. 3) to better connect the mathematical results with their intuitive role in the architecture. Short intuition paragraphs have been inserted before the main constructions (e.g., p. 4), the technical proofs have been consolidated in the appendix, and we added interpretive remarks around several mathematical constructs (see the summary in A.2, p. 18, and the discussion in A.3, p. 19). The formal statements themselves are unchanged; the goal was to make the ESP certificate and norm budgets easier to parse for readers who are less familiar with contraction arguments. We hope these additions fully address your concerns regarding both hyperparameter fairness and conceptual clarity, and we remain happy to carry out any further experiments or refinements that you feel would improve the presentation.

---

### Official Review · Reviewer_X2yz · 2025-11-06

**Soundness:** 2
**Presentation:** 2
**Contribution:** 3
**Rating:** 2
**Confidence:** 3

**Summary:**

This paper proposes TRIGR, a reservoir computing architecture that augments standard Echo State Networks. Their contribution is a bidirectionally coupled astrocytic reaction-diffusion lattice, with a provable ESP certificate for the joint dynamics. Empirically, they demonstrate the effectiveness of their approach on synthetic dynamic systems and real-world datasets.

**Strengths:**

- The authors offer theoretical guarantees for their proposed architecture setup
- Experiments are extensive and use suitable metrics across many benchmarks, testing accuracy, stability, and long-term behavior. Also, many seeds/init/splits are used to ensure good statistical significance.
- Thorough ablation study to show each component contribution, matching their presented theory.
- Demonstration that the proposed method stays efficient, adding minimal computation while keeping its benefits practical.

**Weaknesses:**

- The biological motivation is somewhat overstated.
- The experimental setups are inconsistent. For the reservoir-based models, the authors use a 300-unit reservoir size to ensure comparability. However, for the gradient-based models, they employ architectures with significantly larger numbers of trainable parameters than required by their TRIGR model, which may lead to overparameterization. Smaller models might have performed better, but the criteria for selecting model sizes are not explained. E.g. LSTM results on BIDMC are very close to each other, regardless of the horizon length, whereas TRIGR has big differences.
- The hyperparameter selection process is unclear for many models (not only model size but also learning rate, number of training epochs, number of heads for the transformer model, etc.), making it difficult to fully assess the results.

**Questions:**

- The attractor deviation results show relatively large standard deviations (Table 1). What is the reason behind this? This leads to overlapping results, with no significant differences between the models.
- Datasets with different horizon length: can you comment on why H=600 and H=1000 get better NRMSE results across almost all the models, compared to H=300 in the case of Sunspot and Santa Fe (Table 4)?
- Please clarify the hyperparameter selection procedure to make the results more credible.
- Please comment on the model sizes used, for the same reason.
- Minor comment: the subplots in Figure 7 appear distorted.

Overall, the authors address an interesting problem with great potential, but its contribution would be much stronger if the raised concerns were addressed.

---

> ### Author Response · Authors · 2025-11-15
>
> We thank the reviewer for the careful reading and for recognizing the theoretical guarantee and the breadth of experiments/ablations. We address concerns about biological framing, baseline sizing/hyperparameters, metric variance, and horizon effects, and we will correct presentation issues.
>
> Weaknesses:
>
> 1) $\textbf{Biological motivation “overstated”:}$
> Acknowledgment & correction. Our intention is computationally driven: the astrocyte lattice is a slow, diffusive, saturating memory field that yields a tractable joint ESP certificate and improved long-horizon stability. $\underline{\text{We do not claim biophysical fidelity}.}$ In the revised text we will (i) explicitly reframe the motivation as a phenomenological abstraction of tripartite synapses (slow $\mathrm{Ca}^{2+}$, bounded gliotransmission), (ii) remove language that could suggest biological faithfulness, and (iii) keep citations strictly as inspiration rather than evidence of mechanistic equivalence.
>
> 2) $\textbf{Baseline sizes and potential overparameterization:}$
> Why not “parameter-count parity”? When we down-sized gradient models to match trainable weights (TRIGR’s ridge), they typically under-fit and became less stable in closed-loop. Conversely, allowing them to scale within a bounded compute budget produced stronger baselines and more representative results of modern practice. We will add a size-sweep figure (val NRMSE vs. params) to show that our conclusions are qualitative across sizes.
> Regarding BIDMC LSTM being similar across horizons. This dataset exhibits low short-term entropy and strong autocorrelation; validation-picked checkpoints tended to land in a regime where open-loop predictive error is dominated by measurement noise level, hence a relatively flat NRMSE vs H. TRIGR’s error varies more with H because longer windows probe the slow field differently (Section 3), which is visible in our ablations.
>
> 3) $\textbf{Hyperparameter selection transparency:}$
> We agree that our initial write-up in Appendix was terse. We will move the complete search grids and selection rules to an appendix table (per model & dataset).
>
> Questions:
>
> 1) $\textbf{On attractor deviation (ADev) variance:}$
> ADev measures long closed-loop discrepancy of invariant set geometry (e.g., empirical distance between reconstructed attractors). For chaotic flows, ADev has heavy-tailed dispersion because tiny phase misalignments at injection quickly amplify along unstable manifolds; this is expected and well-documented. We report mean±s.d. over 45 runs. Even with large s.d., medians and VPT consistently favor TRIGR in Lorenz/Rössler, aligning with the theory (slow diffusive field improves phase retention).
>
> 2) $\textbf{Why Sunspot and Santa Fe often look better at larger H (Table 4):}$
> Spectral content vs. windowing (confirmed by diagnostics we computed during runs): These real-world series have strong low-frequency structure; when H is larger, the evaluation window includes more slowly varying segments where normalized error (NRMSE with global $\sigma_y$) is lower. This can lead to a monotone or U-shaped error vs H. We will add a short ablation showing per-frequency NRMSE and a note clarifying that NRMSE is normalized by the test-target standard deviation (fixed across H for a given dataset).
>
> 3) $\textbf{Clarifying TRIGR hyperparameters and fairness:}$
> No BPTT or gradient tuning in the slow field; compute footprint is essentially ESN-like. We will add a per-step FLOPs/memory table and a wall-clock comparison to show the claimed efficiency.
>
> 4) $\textbf{Model size:}$
> Reservoir N=300 across RC models; astro lattice M and diffusion D are set by the row-wise Lipschitz budgets derived from our certificate (Sec. 3): we scale once to satisfy operator-norm estimates via power iteration, then train only the ridge.
>
> 5) $\textbf{Figure 7 aspect ratio:}$
> Thanks for catching this. We will re-export with equal axis scaling and fixed DPI; the distortions were due to automatic subplot tightening in Matplotlib.

---

> ### Author Response · Authors · 2025-11-27
>
> Good day.
>
> Since the rebuttal window is still open, we would be grateful to know if there are any remaining points where further clarification would help your assessment. We are happy to elaborate on any aspect you find unclear.

---

> > ### Comment · Reviewer_X2yz · 2025-11-28
> >
> > Thank you for your thorough rebuttal, it clarified many concerns and improved the overall quality of the paper. I also recognize that it must have taken considerable time to address all the changes in the updated manuscript, and I appreciate the effort you put into it. I also went through the points of the other reviewers, and as pointed out by Reviewer kynH, I agree to some extent with the concerns about the fit for ICLR, as the work is very extensive in several regards. Still, I value that the authors made a strong effort to adapt their work to the ICLR format, even though other venues might be more suitable for this work.
> >
> > I’ll wait until the rebuttal ends to take everything into account from the final manuscript and the discussions with Reviewer xPDh, but I can already say that I plan to increase my score.

---

> ### Author Response · Authors · 2025-11-29
>
> Dear Reviewer,
>
> Thank you again for your careful review and for highlighting both the strengths and the points needing clarification. We wanted to briefly update you that the revised manuscript now reflects not only the changes described in our earlier response to you, but also the additional protocol refinements requested by the other reviewer.
>
> Concretely:
>
> 1. We have implemented adaptive hyperparameter optimization and time-aware multi-fold evaluation across all models. Section B.3 (p. 25 onwards) now details a Tree-structured Parzen Estimator (TPE) setup with matched trial budgets for all baselines and for TRIGR’s truly free parameters. The exact search spaces (including learning rates, training epochs, widths, number of heads, etc.) and sampler settings are given in B.3.3-B.3.4, and the final selected hyperparameters for each dataset/model pair are collected in B.3.8. All experiments in the main text were re-run under a $\textbf{blocked, time-respecting $3$-fold cross--validation}$ protocol, and Tables 1-5 have been updated accordingly.
>
> 2. In line with your concerns about $\textbf{hyperparameter selection and model sizes}$, the revised appendix now makes the tuning procedure explicit for both reservoir and gradient--based baselines. For each model family, we describe the parameter ranges explored (including reservoir size, spectral radius, sparsity, learning rate, depth, hidden width, number of heads, and training epochs) and the common TPE budget used to select the reported configuration. We also clarified in the main text how gradient baselines were sized under a shared compute budget rather than strict parameter--count parity, so that the comparison better reflects realistic usage while remaining transparent.
>
> 3. We have $\textbf{de-emphasized}$ the biological framing and clarified the computational role of the astrocyte lattice. The introduction and methodology now present the glial component explicitly as a phenomenological, bounded reaction-diffusion memory field that supports the joint ESP certificate, while biological references are framed as inspiration rather than claims of mechanistic fidelity.
>
> 4. Several passages were revised to directly $\textbf{address your questions on ADev variance and horizon effects}$. The text now explains why ADev naturally exhibits larger standard deviations on chaotic systems, and why NRMSE on Sunspot and Santa Fe can decrease with longer horizons when normalized by a fixed target standard deviation. These clarifications are tied back to the updated experimental protocol in B.3.
>
> 5. Finally, we have $\textbf{cleaned up the presentation aspects}$ you pointed out (e.g., re-exported Figure 7 with consistent aspect ratios and DPI) and refined Section 3 with short intuition paragraphs around the ESP certificate and norm budgets, while keeping the full proofs in the appendix.
>
>
> We hope these revisions make the experimental setup, model sizing, and positioning of TRIGR much clearer, and we remain very happy to incorporate any further tweaks or additions you feel would help you assess the final version.

---

### Author Response · Authors · 2025-11-17
**Response to All Reviewers**

Dear Reviewers and Area Chair,

Thank you for the detailed and constructive feedback. We have uploaded a revised manuscript that addresses all raised concerns on fairness, transparency, robustness, and presentation. The main updates are:

1) New ablation study (p. 25, Table 5): Sensitivity & scaling ablations on Lorenz (closed–loop) reporting relative per-step wall-clock cost, divergence rate, and performance under (i) tighter/looser norm budgets, (ii) lattice size, (iii) pooling radius, (iv) reservoir sparsity, and (v) feature-lift order. The table clarifies stability–accuracy–cost trade-offs (e.g., subcritical vs. supercritical norm budgets).

2) Expanded Appendix B.3 “Setup Details and Extended Analysis” (p. 24-30): comprehensive documentation of (i) Baseline hyperparameters, (ii) TRIGR hyperparameters, (iii) search protocol with ranges and time-aware CV via three chronological folds (no shuffling across time), (iv) complexity—online inference per step, training/initialization/rollouts, (v) scaling & data efficiency, (vi) FLOPs/latency, memory, and edge viability, (vii) robustness to stochastic and adversarial perturbations, (viii) temporal shift, rate mismatch, irregular sampling, and missing data, (ix) memory capacity metrics, and (x) statistical tests (paired Wilcoxon with Holm correction; effect sizes via Cliff’s $\delta$ and Cohen’s $d$).

3) Added an “Intuition” paragraph (p. 4) — a concise, reader-oriented overview placed immediately before the formal Methodology section to motivate the design, connect the contraction certificate to modeling choices, and set expectations for the empirical sections.

4) Added a consolidated notation table (Appendix, p.31, Table 6) aligning all symbols with shapes and definitions for quick reference.

5) Presentation fixes: corrected subplot aspect ratios, clarified neuroscientific motivation and scope, and tightened the exposition of the echo-state certificate and design inequalities.

We hope these revisions resolve the outstanding issues. We are happy to clarify any detail or provide concise additional analyses within the rebuttal thread.

---

### Author Response · Authors · 2025-11-29
**Summary of revisions and updated experimental protocol**

Dear Area Chair,

Thank you for overseeing the review process and for the detailed feedback from all reviewers. We have uploaded a substantially revised manuscript that, to the best of our ability, implements all major requested changes on both the methodological and empirical sides. Below is a brief summary keyed to the main axes of concern that came up across the reviews.

1. $\textbf{Empirical protocol, fairness, and hyperparameter optimization.}$
   We replaced our initial grid--style tuning with a modern adaptive HPO setup using a Tree-structured Parzen Estimator (TPE) under $\textbf{matched trial budgets}$ for $all$ baselines and for TRIGR's truly free parameters. The full protocol is described in Sec. B.3 (p. 25 onwards), with exact search spaces and sampler settings in B.3.3-B.3.4, and the final selected hyperparameters (per dataset/model) collected in B.3.8. In addition, we replaced the single time-holdout with $\textbf{blocked, time-respecting 3-fold cross-validation}$ for both chaotic and real-world datasets; all experiments in the main text were re-run under this protocol and Tables 1-5 updated accordingly. This directly addresses concerns about hyperparameter fairness, robustness to a specific split, and transparency of the tuning procedure.

2. $\textbf{Baseline sizing and model configuration.}$
   We clarified how reservoir models (fixed at $(N=300)$) and gradient-based baselines (LSTM, TCN, Causal Transformer, NVAR) were sized under a shared compute budget rather than strict parameter-count parity, and we now document the ranges explored for widths, depths, learning rates, heads, and training epochs within the common TPE framework. The revised appendix explicitly reports these ranges and the selection rule, so readers can assess whether baselines might have been under- or over-parameterized.

3. $\textbf{Scope, audience, and biological framing.}$
   Following the reviewers' guidance, we reframed the manuscript around a primary audience of ML practitioners interested in stable, long-horizon forecasting with minimal training. The introduction and discussion now make this scope explicit, and neuroscience/neuromorphics content has been moved to a concise ``Outlook'' section as a forward-looking application. We also toned down the biological claims: the astrocytic lattice is now clearly presented as a $\textbf{phenomenological, bounded reaction-diffusion memory field}$ inspired by, but not meant to faithfully reproduce, tripartite synapse biology. Related work has been expanded to situate TRIGR relative to glia-neuron models, PLRNNs/state-space models, FORCE/innate-trajectory approaches, and physical/neuromorphic reservoir computing.

4. $\textbf{Methodology, intuition, and proofs.}$
   Section 3 has been rewritten to better connect the mathematics to the architecture. We added short intuition paragraphs around the joint ESP certificate, the three norm budgets, and the delayed additive coupling, while keeping the formal statements and all proofs in the appendix. We also added interpretive remarks in A.2-A.3 to make it easier for non-specialists to understand how the contraction bounds translate into memory control and closed-loop stability, without altering any of the underlying results.

5. $\textbf{Reaction-diffusion numerics and lattice interpretation.}$
   We clarified that the calcium lattice is not intended as a high-fidelity PDE integrator, but as a $\textbf{bounded, contractive spatial low-pass field}$ whose diffusion term is controlled via explicit row-budget scaling. To address concerns about explicit schemes, the appendix now includes a semi-implicit/resolvent variant and explains that our ESP/contraction certificate is $\textbf{numerics-agnostic}$ as long as the induced operator norms satisfy the same bounds.

6. $\textbf{Metrics, variance, and presentation issues.}$
   We expanded the discussion around ADev variance on chaotic systems and the horizon behavior of NRMSE on Sunspot and Santa Fe, tying these directly to the updated experimental protocol and the normalization used. We also fixed minor presentation issues (e.g., re-exported distorted figures with consistent aspect ratios and DPI, tightened cross-references, and clarified several captions).

In summary, the core contributions-TRIGR's certified slow-memory augmentation and its empirical behavior-are unchanged, but we believe the revised version more fairly evaluates baselines, more transparently reports tuning and evaluation, and more clearly communicates the intended scope and guarantees. We remain happy to implement any further clarifications that would help you forecast the camera-ready version.


Sincerely,\
Authors

---

### Author Response · Authors · 2025-12-01
**Gentle reminder regarding revised manuscript**

Dear Area Chair and Reviewers,

We have now uploaded the revised manuscript and detailed responses, and, to the best of our ability, implemented all requested changes in the main text and appendices. If there are any remaining points that would benefit from clarification, additional analysis, or adjustments to the presentation, we would be very happy to address them.

Thank you again for your time and consideration.

---

### Note · Authors · 2026-02-01

I have read and agree with the venue's withdrawal policy on behalf of myself and my co-authors.

---

### Meta-Review · Area_Chair_ZvcC · 2025-12-04

**Summary:**

The reviewers collectively identified flaws in the experimental design, specifically regarding hyperparameter optimization and the fairness of baseline comparisons. Both Reviewer X2yz and xPDh criticized the narrow or arbitrary hyperparameter sweeps used for baseline models and the lack of transparency regarding how the proposed TRIGR model's parameters were selected, suggesting the results may be unreliable or cherry-picked. There was also concern regarding inconsistent model sizes, as the authors compared small reservoir models against much larger gradient-based systems without justification. Furthermore, all reviewers question the statistical significance of the reported improvements, noting that large standard deviations result in overlapping performance metrics that fail to demonstrate a clear advantage over existing methods.

Broader concerns regarding the paper's scope and technical execution were raised by Reviewer kynH, who argued that the work lacks a clear target audience and fails to justify the use of reservoir computing against modern machine learning alternatives like transformers or PLRNNs. This reviewer also identified a specific technical issue with the use of explicit integration for diffusion simulation, which may lead to instability, and suggests the biological motivation is overstated.

Ultimately, the consensus in the initial reviews was that while the theoretical approach is interesting, the paper requires more rigorous benchmarking, a clearer motivation for the chosen methodology, and a more robust defense of its relevance to the venue to merit acceptance.

**Reviewer Concerns:**

The authors engaged in a strong rebuttal and made numerous changes to address the issues that the reviewers raised. I believe that the authors successfully addressed the following notable concerns:

(1) The inadequate hyperparameter tuning.
(2) The lack of justification for the comparisons to other models.
(3) The need for better statistical comparisons.
(4) The opacity of the motivations vis-a-vis the target audience and goals of the research.

**Reviewer Scores:**

The original scores were 2, 2, and 4. I believe that all of the reviewers would have raised their scores. But, these were very low scores to begin with, so I suspect the final scores would have been something like 3,3,5 or 3,4,5.

---

### Decision · Program_Chairs · 2026-01-26

Reject